# Deep Neural Nets in Low Dimensions with Sign Activations are Convex Lasso Models

**Emi Zeger**
*emizeger@stanford.edu*
*Department of Electrical Engineering*
*Stanford University*

**Mert Pilanci**
*pilanci@stanford.edu*
*Department of Electrical Engineering*
*Stanford University*

**Reviewed on OpenReview:** *https://openreview.net/forum?id=weh3w6KPs6*

## Abstract

We consider neural networks with sign activations, depths ranging from 2 to an arbitrary but finite number of layers, and rectangular architectures (parallel structures with constant but arbitrary and finite width). We prove that training such neural networks with weight regularization on 1-D data is equivalent to solving convex Lasso problems with discrete, explicitly defined dictionary matrices. The Lasso dictionaries grow richer for 3-layer networks compared to 2-layers, but saturate thereafter. We show that a tree architecture overcomes this depth limitation, allowing the dictionary to expand with every layer. The Lasso model provides intuition and insight, including closed-form solution paths for 1-D data with binary, periodic labels and extensions to certain 2-D data. Numerical simulations support theory.

## 1 Introduction

In a neural network, binary activations are functions that output one of two values, often based on whether the input is above or below some threshold. We focus on sign activations, which output the sign of the input (1 if the input is nonnegative, and $-1$ otherwise.[1]) Sign-based functions are used to binarize activations in [2], [4], [5], [1], [3].

Binary activations have been investigated since some of the earliest research on neural networks [6], motivated by the all-or-nothing firing of neurons in the brain [7]. Related works include the development of Hopfield networks [8] to represent memory and analysis used to develop symbolic logic [9] from this discrete behavior. In a similar spirit, neuromorphic computing [5] represents computation with biologically-inspired neurons outputting binary spike trains [7], known as spiking neural networks. Binary neural networks are studied in [5], [10], [1], which have binary activations as well as binary weights. Binary neurons and activations [2] may be deterministic or stochastic [11], outputting binary values based on a sigmoid probability. However, the randomness of stochastic activations are challenging for hardware to implement in practice [1], [2]. All parameters and activations in this paper are deterministic.

The biological connection and simplicity of binary activations provide advantages in interpretability [7][12] and efficiency over continuous-valued activations such as ReLU. Advances in internet-of-things (IoT) technology and privacy concerns when using IoT devices are driving demand for compute on edge devices [5]. As models such as LLMs scale to billions of parameters [10], binarization represents the ultimate quantization goal for implementation on memory-constrained edge devices, where 1-bit hardware computations improve

---

[1]To keep the activation output binary, we consider $\text{sign}(0) = 1$ [1], [2]. Similar approaches are used in [3].

latency and efficiency (13), (10) Binarizing activations (and weights) in (2),(4) leads to over an order in magnitude improvement in energy efficiency, at comparable performance. Binary activations can also perform regularization similar to dropout that improves network performance (2), (1), (11). This paper focuses on only binarizing activations, and treats weights as real, continuous values.

Despite their benefits, binary activations pose a significant challenge: they are not differentiable at their transition points, and have zero gradient elsewhere. This "vanishing gradient" property can prevent standard gradient descent from learning effectively (11),(12), (3). Common training methods rely on approximations such as the Straight Through Estimator (11) or stochastic weight modeling (3). We formulate training as an exact and equivalent Lasso problem, which enjoys optimized solvers (14) and bypasses the vanishing gradient challenge without requiring randomness or approximations.

A low-dimensional data assumption enables much of our simplifications and insight. Medical and scientific images such as MRI are often assumed to have an underlying low-dimensional structure (15). Low-dimensional data assumptions are often made in modeling and analysis (16; 15; 17; 18). Time-series data can also be low-dimensional. For example, acoustic signals from a microphone over time can be interpreted as a 1-D data stream (19; 20). Coordinate-based MLPs are used in computer vision and graphics to map low-dimensional data consisting of 3-D coordinates to characteristics such as shape and color (21; 22; 23). The works (24; 25; 26; 27; 28; 29; 30; 31) study neural networks with 1-D data.

Neural networks are non-convex, which makes it challenging to train, as gradient descent methods can get stuck in local optima. In contrast, all stationary points of a convex problem are globally optimal. We show that training certain neural networks is equivalent to the well-studied, convex Lasso problem:

$$\min_{\mathbf{z}} \frac{1}{2}\|\mathbf{A}\mathbf{z} - \mathbf{y}\|_2^2 + \beta\|\mathbf{z}\|_1 \tag{1}$$

where $\mathbf{A}$ is the *dictionary matrix*, and its columns $\mathbf{A}_i$ are called *features*. The set of features is a *dictionary*. The parameter $\beta > 0$ is a fixed regularization parameter. The *solution path* for the Lasso or training problem is the map from $\beta \in (0, \infty)$ to the solution set.

The Lasso problem has several advantages in addition to global optimality guarantees via convexity. First, the $l_1$ penalty in the Lasso problem favors a sparse solution for the vector $\mathbf{z}$, making the Lasso problem interpretable as a feature selector (32). Next, the Lasso problem has efficient solvers based on proximal gradient methods and Least Angle Regression (LARS) (14). Finally, the nature of the Lasso solution set (14) and solution path (33; 34; 14) are well understood. In contrast, non-convex neural networks are largely opaque and often treated as black boxes.

## 1.1 Contributions

Our contributions include the following results. We focus on two primary classes of networks, formally defined in Section 3 and Appendix C.1: rectangular networks, which have constant width across all layers, and tree networks, which have a branching structure at every layer.

1. Training rectangular networks of arbitrary depth with sign activation on 1-D data is equivalent (Section 2.3) to solving Lasso problems with finite and fixed dictionaries (Theorem 3.4).

2. The features are data-invariant, binary[2] vectors in $\{-1, 1\}^N$ and are characterized explicitly by their switching complexity (number of times elements change value) (Theorem 3.4).

3. The Lasso problems for 2 and 3-layer rectangular networks with sign activation for certain 1-D data yield closed-form solution paths that suggest better generalization for 3-layer networks (Corollaries 4.2 and 4.4) corresponding to richer dictionaries (Corollary 3.1, Lemma 3.2).

4. At depth 3, dictionaries for rectangular networks freeze: additional layers contribute no new features (Theorem 3.4). A tree architecture can resolve this bottleneck, yielding an explicitly characterized dictionary that expands with every layer (Theorem 3.5).

---

[2]Depending on context, binary may denote values of $\pm 1$ rather than $0, 1$.

5. A similar Lasso equivalence extends to certain neural networks with sign activation trained on 2-D data in the upper half plane (Theorem 3.16).

## 1.2 Related Work

Neural networks with 1-D data are analyzed in (24; 25), but for ReLU activation and 2 layers. It is shown in (27; 28; 29; 30) that optimal 2-layer ReLU networks interpolate 1-D training data piecewise linearly. In contrast, networks (of any depth) with sign activation can yield piecewise constant interpolants (Corollary 3.10, Corollary 3.14). This paper analyzes neural networks from a convex optimization perspective. Two-layer ReLU neural networks are formulated as convex, constrained group Lasso models in (35; 28; 36). In (12), a similar approach convexifies the training problem for threshold activations with general $d$-dimensional data as a Lasso problem. In (31), 2-layer networks with sign activation trained on 1-D data are formulated as convex Lasso problems. The most relevant literature to this work are (31) and (12), and they differ from this work in the following ways.

First, while (12) studies threshold activations, this paper focuses on sign activations. As these are both binary activations, our results may inform insights about threshold activations as well. The use of sign rather than threshold activation makes the Lasso formulation simpler, for example reducing the dictionary matrix for 2-layer networks to a symmetric and invertible $N \times N$ matrix rather than a $N \times 2N$ matrix for threshold activation (31).

A (neural) *reconstruction* is a map from an optimal Lasso solution to an optimal network. Although (12) expresses deep networks as Lasso problems, the features and reconstruction are described implicitly. We focus on 1-D data to provide explicit Lasso dictionaries and reconstructions that ensure practical tractability of the Lasso problem. Furthermore, the dictionaries in (12) can require high computational complexity to create, but we show that for 1-D data and rectangular networks, the dictionaries freeze after 3 layers (Theorem 3.4), bounding the complexity of globally optimal training.

Second, while (31) only considers binary activations in networks of depth 2, this work considers arbitrarily deep networks with sign activation. Analysis of networks deeper than 3 layers in (31) is limited to absolute value activation and architectures with only one neuron per parallel unit in each layer. This work generalizes this analysis to wider 'rectangular' and 'tree' architectures of arbitrary width and depth. Moreover, while (31) only considers 1-D data, this work includes an extension to certain 2-D data. Our results show that the choice of sign activation rather than the piecewise linear activations studied in (31) distinctly changes the behavior of deep networks and corresponding Lasso features for low-dimensional data.

While our analysis follows the convexification framework for neural networks developed in prior work (31), (12), our contribution is not the general convex duality and reconstruction approach itself. Rather, we focus on sign-activated networks and exploit the low-dimensional setting to obtain explicit characterizations of the resulting Lasso dictionaries and neural reconstructions. Our characterization of features in terms of their switching complexity is a distinct approach from (12; 31). This framework enables the features to be largely data-independent, (Remark 3.7), whereas the features in (12; 31) are defined in terms of the training data. Item 1 of our contributions (Section 1.1) namely the equivalence between training and a convex Lasso problem, builds on the broader convex-training literature and generalizes the 2-layer analysis of (31) to arbitrary depth.

While (12) also treats deep networks with regards to item 1, studying deep networks through convex formulations, they focus on threshold activation and general-dimensional inputs; in contrast, we focus on 1-D data to derive explicit, tractable Lasso models. Rather than in the general convexification methodology, our main novelty is the explicit characterization of the resulting dictionaries and reconstructions for sign-activated networks on low-dimensional data. This yields the switching-complexity interpretation of features, explicit finite dictionaries, depth-saturation results for rectangular networks, growing dictionaries for tree architectures, and the 2-D extension in Section 3.4. Indeed, items 2—5 in the Contributions section are entirely distinct from both (31) and (12).

## 2 Preliminaries

### 2.1 Notation and assumptions

The number of training samples is $N$. Assume 1-D training data are distinct and ordered as $x_1 > x_2 > \cdots > x_N$. Unless otherwise stated, assume the input data is 1-D, and the activation is sign: $\sigma(x) = \begin{cases} 1 & \text{if } x \geq 0 \\ -1 & \text{else.} \end{cases}$

$\mathbf{A}_i \in \mathbb{R}^N$ is a Lasso feature, $\mathbf{z}^*$ is an optimal solution to the Lasso problem (1) and $||\mathbf{z}^*||_0$ is the number of nonzero elements in $\mathbf{z}^*$. A network that uses sign activation is a "sign network." For shorthand, a network with 1-D inputs is a "1-D network." The weights and biases of a 1-D network are **not** necessarily 1-D and may be vectors and matrices. A "sign feature" is a Lasso feature for a sign network. Let $\mathbb{R}^n$ and $\mathbb{R}^{1 \times n}$ be the set of $n$-dimensional, real-valued column and row vectors, respectively. Let $\mathbf{1}, \mathbf{0} \in \mathbb{R}^N$ be the all-ones and all-zeros vectors, respectively, and $[n] = \{1, 2, \ldots, n\}$. Finally, the indicator function of a logical statement $z$ is $\mathbf{1}\{z\}$.

### 2.2 Neural net architectures

This section defines neural net terminology and notation. For comparability and consistency, the definitions are adapted from (31; 12). The depth of a neural network is denoted as $L$. To increase flexibility and generalize to outputs beyond $\{-1, 1\}$, we train amplitude-scaled activations $\sigma_s(x) = s\sigma(x)$, where $s$ is a trainable amplitude parameter. Each amplitude scales the vector output of a corresponding neuron. Stacking these as columns into a matrix $\mathbf{Z}$, the activation is expressed in matrix form as $\sigma_{\mathbf{s}}(\mathbf{Z}) = \sigma(\mathbf{Z})\text{Diag}(\mathbf{s})$ for $\mathbf{Z} \in \mathbb{R}^{n \times m}, \mathbf{s} \in \mathbb{R}^m$. Model quantization also uses amplitude parameters to provide an additional degree of freedom that scales quantized weights and activations (10; 4; 5).

An $L$-layer *neural network* with $d$-dimensional inputs is a function $f_L(\mathbf{x}; \theta) : \mathbb{R}^{1 \times d} \rightarrow \mathbb{R}$ parameterized by $\theta$, the *parameter set*. The input $\mathbf{x}$ is a row vector, and the network extends to matrix inputs $\mathbf{X} \in \mathbb{R}^{N \times d}$ row-wise. The parameter set $\theta$ is ordered and is partitioned into ordered sets of weight and bias parameters $\theta_w$ and $\theta_b$, respectively. We first describe a typically used architecture, termed standard networks.

**Standard networks**: Let $L \geq 2$, $m_1 = d$ and $m_l \in \mathbb{N}$ be widths of layers $l = 2, \cdots, L$. A *standard neural network* is recursively defined as

$$f_L(\mathbf{x}; \theta) = \mathbf{X}^{(L)}\boldsymbol{\alpha}, \text{ where } \mathbf{X}^{(l+1)} = \sigma_{\mathbf{s}^{(l)}}\left(\mathbf{X}^{(l)}\mathbf{W}^{(l)} + \mathbf{b}^{(l)}\right), \text{ and } \mathbf{X}^{(1)} = \mathbf{x} \tag{2}$$

For each layer $l \in [L-1]$, $\mathbf{W}^{(l)} \in \mathbb{R}^{m_l \times m_{l+1}}, \mathbf{s}^{(l)} \in \mathbb{R}^{m_{l+1}}, \mathbf{b}^{(l)} \in \mathbb{R}^{1 \times m_{l+1}}$ are the weights, amplitudes and biases, respectively, and $\boldsymbol{\alpha} \in \mathbb{R}^{m_L}$ is the final layer weight. The weight and bias parameters are in $\theta_w = \{\boldsymbol{\alpha}, \mathbf{W}^{(l)}, \mathbf{s}^{(l)} : l \in [L-1]\}$ and $\theta_b = \{\mathbf{b}^{(l)} : l \in [L-1]\}$, respectively.

While convex analysis of standard networks is challenging (37), expanding to a parallel structure increases mathematical tractability, in particular for reconstructing an optimal network. This work focuses on the parallel network, an approach also taken in (38; 37; 12; 31; 39). Remark C.3 gives a conversion for a parallel to standard network. When referring to 2-layer networks, we omit the architecture as the standard and parallel architectures coincide (Appendix D). We now define the parallel network.

**Parallel networks**: Let $L \geq 2$ and $m_0 = d, m_{L-1} = 1$ and $m_l \in \mathbb{N}$ for $l \in [L] - \{L-1\}$ be the network widths[3]. A parallel network is a linear combination of $m_L$ standard networks:

$$f_L(\mathbf{x}; \theta) = \sum_{i=1}^{m_L} \hat{\mathbf{X}}^{(i,L)}\alpha_i, \text{ where } \hat{\mathbf{X}}^{(i,l+1)} = \sigma_{\mathbf{s}^{(i,l)}}\left(\hat{\mathbf{X}}^{(i,l)}\mathbf{W}^{(i,l)} + \mathbf{b}^{(i,l)}\right), \text{ and } \hat{\mathbf{X}}^{(i,1)} = \mathbf{x} \tag{3}$$

The $\hat{\mathbf{X}}^{(i,L)}$ are *parallel units*. The parameters $\mathbf{W}^{(i,l)} \in \mathbb{R}^{m_{l-1} \times m_l}$, $\mathbf{s}^{(i,l)} \in \mathbb{R}^{m_l}$, and $\mathbf{b}^{(i,l)} \in \mathbb{R}^{1 \times m_l}$ are the weights, amplitude, and biases, respectively, of layer $l \in [L-1]$ in parallel unit $i \in [m_L]$, and $\boldsymbol{\alpha} \in$

---

[3]The second dimension of $\mathbf{W}^{(i,L-1)}$ is 1 in parallel networks while the second dimension of $\mathbf{W}^{(L-1)}$ is $m_L$ in standard networks. Therefore to define $m_L$ as the length of $\boldsymbol{\alpha}$ in both standard and parallel networks, we define $m_1$ to be the first dimension of $\mathbf{W}^{(1)}$ in standard networks, while defining $m_0$ as the first dimension of $\mathbf{W}^{(i,1)}$ and $m_{L-1} = 1$ in parallel networks.

$\mathbb{R}^{m_L}$. We partition the weight and bias sets as $\theta_w = \bigcup_{i \in [m_L]} \theta_w^{(i)}$ and $\theta_b = \bigcup_{i \in [m_L]} \theta_b^{(i)}$, where $\theta_w^{(i)} = \{\alpha_i, \mathbf{s}^{(i,l)}, \mathbf{W}^{(i,l)} : l \in [L-1]\}$ and $\theta_b^{(i)} = \{\mathbf{b}^{(i,l)} : l \in [L-1]\}$.

For vector inputs $\mathbf{x}$, $\hat{\mathbf{X}}^{(i,L)}$ is a scalar in the parallel network while $\hat{\mathbf{X}}^{(L)}$ is a vector in the parallel network. To account for this difference, we let $m_1 = d$ in standard networks but set $m_0 = d$ and $m_{L-1} = 1$ in parallel networks. This ensures that $m_L$ is consistently the length of $\boldsymbol{\alpha}$, the number of final layer weights, in both architectures. With the networks defined, we can now define the training problem.

## 2.3 Training problem

We let $\theta \in \Theta$, where $\Theta$ is the *parameter space*. $\mathbf{X} \in \mathbb{R}^{N \times d}$ is a *data matrix* consisting of $N$ *training samples* $\mathbf{x}_1, \cdots \mathbf{x}_N \in \mathbb{R}^{1 \times d}$. The corresponding *target vector* is $\mathbf{y} \in \mathbb{R}^N$. All scalar functions extend to vector and matrix inputs component-wise. This paper considers neural network regression tasks with *training problem*

$$\min_{\theta \in \Theta} \frac{1}{2} \|f_L(\mathbf{X}; \theta) - \mathbf{y}\|_2^2 + \frac{\beta}{2} \|\theta_w\|^2 \tag{4}$$

where $\beta > 0$ is a fixed regularization parameter. The regularization term is $\|\theta_w\|^2 = \sum_{q \in \theta_w} \|q\|_2^2$, which penalizes the total network weight. *Training* the neural net consists of solving the training problem (4) for $\theta$. The weight set $\theta_w$ incurs regularization during training while the bias set $\theta_b$ does not. An *optimal* neural net $f_L(\mathbf{x}; \theta)$ uses parameters $\theta$ that globally minimize the training loss in problem (4). In this paper, we discuss the equivalence of the training problem and the Lasso problem. Two optimization problems are *equivalent* if they share the same optimal value, and the solution for one problem can be reconstructed from the other[4]. We begin with a prior result established in (31):

**Lemma 2.1** ((31)). *The training problem for a 1-D, 2-layer sign network is equivalent to a Lasso problem whose dictionary matrix is* $\mathbf{A} \in \mathbb{R}^{N \times N}$ *with* $\mathbf{A}_{n,i} = \sigma(x_n - x_i)$, *provided* $m_2 \geq \|\mathbf{z}^*\|_0$.

As defined in Section 2.1, $\mathbf{z}^*$ is a Lasso solution. Lemma 2.1 shows that instead of performing non-convex training, an optimal neural network can be found by solving a convex Lasso problem 4. Given an optimal Lasso solution $\mathbf{z}$, (31) shows that an optimal neural network is $f_2(\mathbf{x}; \theta) = \sum_i z_i \sigma(x - x_i)$ and the network output on the data is $f_2(\mathbf{X}; \theta) = \mathbf{A}\mathbf{z}$ with $\mathbf{A}_i = \sigma(\mathbf{X} - x_i \mathbf{1})$. Hence Lemma 2.1 expresses training as learning coefficients $z_i$ for basis neurons $\sigma(x - x_i)$ centered at the training samples $x_i$. The neurons act as basis functions or "atoms" (40). This paper generalizes Lemma 2.1 to arbitrary deep and wide rectangular networks, discussed next.

# 3 Main Results

This section describes the main results equating non-convex training to convex reformulations. We will show that sign networks can be viewed as learning binary Lasso features characterized by the number of times they switch value. We first characterize the features of shallow networks, then show that an additional layer increases the number of switches, prove that this growth saturates beyond three layers for rectangular architectures, and finally show that tree architectures overcome this limitation. Proofs in this section are deferred to Appendix E.3.

In the context of neural reconstruction developed in this and the following section, we will refer to Lasso features $\mathbf{A}_i$ as *basis signals* and their corresponding reconstructed parallel units as *basis functions*. We say the binary vector $\mathbf{h} \in \{-1, 1\}^N$ *switches* at index $1 < n \leq N$ if $h_n \neq h_{n-1}$. For $n \in \mathbb{N}$, let the *switching set* $\mathbf{H}^{(n)}$ be the set of all vectors in $\{-1, 1\}^N$ that start with 1 and switch at most $n$ times[5]. (Note the number of switches is at most $\min\{n, N\}$ times. The dependence of $\mathbf{H}^{(n)}$ on the number of data $N$ is implicit.) The next result relates the switching set to the 2-layer Lasso dictionary introduced in Lemma 2.1.

**Corollary 3.1.** *The dictionary for a 2-layer sign network with 1-D data is* $\mathbf{H}^{(1)}$.

---

[4]This is a typical definition of equivalence in optimization theory.
[5]Intuitively, it suffices to restrict the first element to 1 due to symmetry properties of the sign activation. See Appendix E and Remark E.6 for proofs.

Corollary 3.1 characterizes the Lasso dictionary for 2-layer networks as the set of binary vectors that switch at most once. The next result shows that Corollary 3.1 generalizes to 3-layer rectangular networks with richer corresponding Lasso dictionaries. Recall from Section 2.2 that a 3-layer parallel network consists of $m_3$ parallel units, each with $m_1$ neurons.

**Lemma 3.2.** *The training problem for a $3$-layer parallel network with sign activation and 1-D inputs is equivalent to a Lasso problem whose dictionary is the switching set $\mathbf{H}^{(m_1)}$, provided $m_3 \geq \|\mathbf{z}^*\|_0$.*

Lemma 3.2 shows that 3-layer parallel sign networks represent functions on 1-D data as a linear combination $\sum_i z_i \mathbf{A}_i$ of binary Lasso features $\mathbf{A}_i \in \mathbf{H}^{(m_1)}$. These features $\mathbf{A}_i$ correspond to all possible binary vector outputs of parallel units $\hat{\mathbf{X}}^{(i,L)}$ on the data $\mathbf{X}$ over different parameter values, as shown in the proof of Lemma 3.2. Lemma 3.2 characterizes these binary features by their switches. Each of the $m_1$ parallel units contributes a possible switch point to a feature. The wider the network, the more switches a feature can have, expanding the dictionary $\mathbf{H}^{(m_1)}$ and allowing more expressivity in the network. A reconstruction of an optimal network is given below in Lemma 3.12.

Corollary 3.1 shows that the features for 2-layer networks switch once, while Lemma 3.2 shows that features for 3-layer networks switch $m_1$ times. If more layers are added, do the features get richer with more switches? To focus on the impact of depth over width, we consider rectangular networks, which have the same number of neurons in each successive layer.

**Definition 3.3.** *A* rectangular network *of depth $L \geq 3$ is a parallel network (Section 2.2) with $m_1 = \cdots = m_{L-2}$.*

All 3-layer parallel networks are automatically rectangular networks since $m_1 = m_{3-2} = m_{L-2}$. The next theorem answers the above question on deeper network features in the negative.

**Theorem 3.4.** *The training problem for a rectangular network of any depth $L \geq 3$ with sign activation and 1-D data is equivalent to a Lasso problem whose dictionary is the switching set $\mathbf{H}^{(m_1)}$, provided $m_L \geq \|\mathbf{z}^*\|_0$.*

Theorem 3.4 shows that rectangular sign networks an also be interpreted as Lasso feature selectors. Moreover, for fixed width the deeper networks inherit the same dictionary as the three-layer case. The dictionary only depends on the width $m_1$ of the first layer, with features continuing to have only $m_1$ switches no matter how many layers beyond 3 the rectangular network has. Corollary 3.1 and Theorem 3.4 show that networks with sign activation are equivalent to Lasso models with libraries that expand until depth 3 and then freeze for rectangular networks.

This saturation phenomenon can be understood from the structure of sign activations. The piecewise constant, binary nature of sign activation, whose output depends only on whether the input is positive, limits the possible outputs of parallel units and makes the dictionary invariant to the specific values of the 1-D training data. This restriction causes the dictionary to stagnate. This suggests that to increase the representation power of networks, the width may need to be successively increased in each layer. While exploring width variation is a promising direction for future work, we instead focus on how network topology can influence feature diversity through an alternative tree architecture.

Like parallel networks, a *tree network* is a linear combination of parallel units, but each unit is tree-structured. While each unit of a parallel network is a standard network, each branch of a tree network unit is a parallel network. A detailed definition is given in Appendix C.1 due to space. The next result generalizes the Lasso equivalence to deeper tree networks.

**Theorem 3.5.** *Theorem 3.4 applies as is to tree networks, but with $\mathbf{H}^K$ where $K = \prod_{l=1}^{L-1} m_l$ instead of $\mathbf{H}^{m_1}$.*

Tree networks are equivalent to Lasso models as well, but with a wider set of features. Theorem 3.5 shows that unlike for rectangular sign networks, the Lasso dictionaries for tree networks with sign activation continue to expand with depth beyond 3 layers, on 1-D data. The tree network's features have up to as many switches as the product of the number of neurons in each layer.

**Remark 3.6.** *For 1-D data, $2$-layer sign networks (when written in parallel form) have features with $1 = m_0$ switches, and $3$-layer parallel sign networks have features with $m_1 = m_0 m_1$ switches, so Theorem 3.5 can be viewed as using the tree architecture to extend the product form in the number of switches.*

The branching nature of the tree network enables the combinatorial complexity of the features to increase. A tree network takes inputs at its first-layer neurons, which act as leaf nodes. Their outputs are then recursively combined through intermediate layer neurons until reaching the root (Appendix C.1). Just as parallel network features correspond to outputs of parallel units on $\mathbf{X}$, the features of a tree network correspond to outputs of the main "root branches", as shown in the proof of Theorem 3.5. The tree structure allows new combinations of neuron outputs that bypass the feature stagnation of parallel architectures.

**Remark 3.7.** *The dictionaries of 1-D rectangular and tree networks with sign activation are invariant to the training data. To train multiple neural nets with the same rectangular or tree architecture and sign activation on different 1-D datasets of the same size, the dictionary matrix $\mathbf{A}$ only needs to be constructed once.*

Moreover, Lemma 3.2 shows that the dictionary only needs to be constructed once to be used for all 3-layer or deeper rectangular sign networks with 1-D data. The Lasso problem hence provides a way to train deep networks globally optimally with fixed and limited complexity. Lemma 3.2 shows that if a 3-layer rectangular sign network is trained to global optimality on 1-D data, then a deeper network cannot achieve a better training objective value. In practice, training the non-convex program may reach a local optima, and adding more layers beyond 3 may change the loss landscape, leading to different suboptimal solutions, an area for future analysis. In terms of training time, Section 5.1 gives experiments where the non-convex training of even 2-layer networks can take significantly longer than training the corresponding Lasso problem, with the Lasso problem outperforming the non-convex training.

We now discuss how the simple dictionary $\mathbf{A}$ and the feature selection property of Lasso models makes neural networks interpretable. An *optimal fit* of the training data $\mathbf{X}$ is the output $f_L(\mathbf{X}; \theta)$ where $f_L(x; \theta)$ is an optimal network. The next result uses Theorem 3.4 to find an optimal fit in terms of a fixed set of features. As we discuss consequences of the Lasso equivalence, assume $m_L \geq ||\mathbf{z}^*||_0$, so the Lasso equivalence holds.

**Corollary 3.8.** *For a rectangular, 1-D sign network, the Lasso dictionary gives a fixed, finite set of binary features $\mathbf{A}_i$ depending only on the number of data $N$ and the width $m$ such that for any depth $L \geq 3$, any training data $\mathbf{X}, \mathbf{y}$ and regularization $\beta$, there are coefficients $z_i^*$ that give an optimal fit $f_L(\mathbf{X}; \theta) = \sum_i z_i^* \mathbf{A}_i$.*

Corollary 3.8 extends analogously to shallow networks and tree networks, although the dictionary is simply $\mathbf{H}^{(1)}$ for shallow networks and depends on the number of neurons in each layer for tree networks.

**Corollary 3.9.** *Corollary 3.8 also applies to 2-layer networks and tree networks, but with dictionaries depending only on $N$ for 2-layer networks, and on $\prod_{l=1}^{L-1} m_l$ instead of $m$ for tree networks.*

The binary features can therefore be interpreted as basis signals that can be linearly combined to fit any $N$ training points in 1-D with a wide class of networks. The coefficients $z_i^*$ are a function of the target vector $\mathbf{y}$. The next result shows that features interpreted as basis signals correspond to subsampled parallel units that are interpreted as basis functions. By definition, each parallel unit of a parallel network is implicitly a function of the input (Section 2.2). We will denote the parallel units as $f_i = \hat{\mathbf{X}}^{(i,L)}$ to make the function's dependence on the input explicit as $f_i(x)$. For unified notation, we will also denote the tree network's $i^{\text{th}}$ parallel unit as $f_i$. The $f_i$ can be interpreted as basis functions in the following sense.

**Corollary 3.10.** *For 1-D rectangular and tree networks with sign activation, each feature $\mathbf{A}_i$ corresponds to a parallel unit function $f_i$ such that there is an optimal network that can be written as $\sum_i z_i^* f_i(x)$. The $f_i$ interpolates the feature $\mathbf{A}_i$ in a piecewise constant manner as $f_i(x) = A_{n,i}$ for $x \in [x_n, x_{n-1})$.*

The parameters of the basis functions $f_i$ depend on the training data. Basis functions represent standard networks for rectangular architectures and subtrees for tree networks. The $l_1$ penalty promotes a sparse Lasso solution $\mathbf{z}^*$, resulting in a parsimonious set of basis functions $\{f_i : z_i \neq 0\}$ to represent the network. The Lasso model shows that the role of the regularization is to prune out entire parallel units from the network rather than individual weights within each unit.

We now give an explicit formula for the basis function parameters in Corollary 3.10 to reconstruct an optimal 3-layer parallel network. We first give an informal statement of the reconstruction. A 3-layer, 1-D parallel network is of the form $f_3(x; \theta) = \sum_{i=1}^{m_L} \alpha_i \sigma_{s^{(i,2)}} \left( \sigma_{s^{(i,1)}} \left( x\mathbf{W}^{(i,1)} + \mathbf{b}^{(i,1)} \right) \mathbf{W}^{(i,2)} + \mathbf{b}^{(i,2)} \right)$ where $\mathbf{W}^{(i,1)}$ and $\mathbf{b}^{(i,1)}$ are row vectors, $\mathbf{W}^{(i,2)}$ is a column vector, and $\mathbf{b}^{(i,2)}$ is a scalar. The first layer width is $m_1$, which is the length of $\mathbf{W}^{(i,1)}$ and $\mathbf{b}^{(i,1)}$. Each feature $\mathbf{A}_i$ dictates parameters for the $i^{th}$ parallel unit $\hat{\mathbf{X}}^{(i,L-1)}$.

**Lemma 3.12 (Informal)** *A Lasso solution for a 1-D, 3-layer parallel network corresponds to the following optimal parameters:* $\boldsymbol{\alpha} = \mathbf{z}^*$, $\mathbf{W}^{(i,1)} = \mathbf{1}$ *and* $\mathbf{W}^{(i,2)}$ *with elements alternating between* $\pm 1$. *The elements of* $\mathbf{b}^{(i,1)}$ *contain a subset of training data multiplied by* $-1$. *All amplitude parameters are 1. The second-layer bias* $\mathbf{b}^{(i,2)}$ *is 0 if* $m_1$ *is even and* $-1$ *otherwise.*

To formalize the reconstruction, we define a change of variables called unscaling (41). Assuming the outermost amplitude is $\mathbf{s}^{(i,L-1)} = 1$, unscaling does not change the network as a function, but rebalances the outermost weight coefficients to optimize the parameters with respect to the regularization penalty.

**Definition 3.11.** Parameter unscaling *is the transformation* $\alpha_i' = \text{sign}(\alpha_i)\sqrt{|\alpha_i|}$ *and* $s^{(i,L-1)\prime} = \sqrt{|\alpha_i|}$.

Now we give a formal reconstruction. Let $m^{(i)}$ be the number of times feature $\mathbf{A}_i$ switches, and $I_n^{(i)}$ be the index of the $n^{\text{th}}$ switch, with $I_1^{(i)} < \cdots < I_{m^{(i)}}^{(i)}$. Let $g(i,n) = I_n^{(i)}$ if $n \le m^{(i)}$ and $g(i,n) = N$ otherwise. In other words, $g(i,n)$ is the index of the $n^{\text{th}}$ existing switch in $\mathbf{A}^{(i)}$, set to the index of the last index if $n > m^{(i)}$, i.e., more than the number of switches. We consider each neuron in a 2-layer network to be a "unit."

**Lemma 3.12.** *Consider a 1-D, 2 or 3-layer parallel network and its corresponding Lasso. Optimal parameters are found from its solution* $\mathbf{z}^*$ *as follows. First, for every* $z_i^* \ne 0$, *let* $\alpha_i = z_i^*$, $\mathbf{W}_n^{(i,1)} = 1$, $\mathbf{b}_n^{(i,1)} = -x_{g(i,n)-1}$ *and* $s^{(i,1)} = 1$. *For 3-layer networks, additionally let* $\mathbf{W}_n^{(i,2)} = (-1)^{n+1}$, $\mathbf{b}^{(i,2)} = -\mathbf{1}\{m_1 \text{ odd}\}$, $s^{(i,2)} = 1$. *Second, unscale all parameters. Third, for all* $i$ *with* $z_i^* = 0$, *let all parameters in the* $i^{th}$ *unit be zero.*

Lemma 3.12 describes an explicit and efficient reconstruction. The second-layer weight contains training data $\mathbf{x}_n$ corresponding to the indices $n$ at which feature $\mathbf{A}_i$ switches. Figure 23 in the Appendix illustrates an example of reconstructing a 3-layer network as given in Lemma 3.12. Lemma 3.12 can be stated in terms of runs. A run of 1 (resp. -1) is a maximal contiguous subsequence of entries equal to 1 (resp. -1) within a feature vector. A run may have length 1. Runs are ordered from left to right by their starting index. Using runs, we give an example of the reconstruction in Lemma 3.12.

**Example (reconstruction in Lemma 3.12)** Consider a 3-layer parallel network with $m_2 = 2$ trained on $N = 3$ points $x_1, x_2, x_3 \in \mathbb{R}$. Suppose an optimal Lasso solution $\mathbf{z}$ has three non-zero indices: $z_1, z_2, z_3$ with corresponding dictionary columns $A_1 = (1,1,1)^T, A_2 = (1,1,-1)^T, A_3 = (1,-1,1)^T$. Following the reconstruction in Lemma 3.12(without parameter unscaling, for simplicity), we get $\alpha_i = z_i, \mathbf{b}^{(i,2)} = 0, \mathbf{W}^{(i,1)} = (1,1)^T, \mathbf{W}^{(i,2)} = (1,-1)^T$ for $i \in [3]$. To find $\mathbf{b}^{(i,1)}$ we identify the last indices of the first $m_2 = 2$ runs in feature $A_i$: for $A_1$ these indices are both defined as $N = 3$ since $A_1$ is entirely a run of 1s; for $A_2$ this is 2 and 3 since $A_2$ has a run of 1s along indices $\{1, 2\}$ and a run of $-1$ at index 3; and for $A_3$ this is 1 and 2 since $A_3$ has a run of 1 at index 1 and a run of $-1$ at index 2. Hence

$$\mathbf{b}^{(1,1)} = (-x_3, -x_3)^T, \mathbf{b}^{(2,1)} = (-x_2, -x_3)^T, \mathbf{b}^{(3,1)} = (-x_1, -x_2)$$

The resulting network simplifies to

$$f_L(\mathbf{x}; \theta) = z_1 + z_2 \sigma(\sigma(x - x_2) - \sigma(x - x_3)) + z_3 \sigma(\sigma(x - x_1) - \sigma(x - x_2))$$

**Remark 3.13.** *Since deeper rectangular networks have the same dictionary as for 3 layers, a reconstruction similar to Lemma 3.12 holds for them by setting deeper layer weight matrices to the identity.*

Reconstructions for tree architectures are implied by Appendix E.3 (see Lemma E.12), which shows that every feature is realizable as the output of a root branch with input $\mathbf{X}$ for some parameter configuration. Hence given a Lasso solution $\mathbf{z}^*$, there exists an optimal tree network (before unscaling) that is a linear combination of corresponding root branches with outer-layer coefficients $z_i^*$. The Lasso formulation enables insight into non-convex neural networks. For example, any globally optimized 1-D parallel or tree sign network achieves perfect interpolation as the regularization becomes minimal.

**Corollary 3.14.** *If* $m_L \ge m^*$, *then an optimal fit* $f_L(\mathbf{X}; \theta) \to \mathbf{y}$ *as* $\beta \to 0$.

The Lasso model also yields optimal performance bounds of the training loss for 2 versus 3 layer parallel networks. In the next result, the number of neurons in each layer $l$ of a 2 and 3-layer network is denoted by $m_l'$ and $m_l$, respectively.

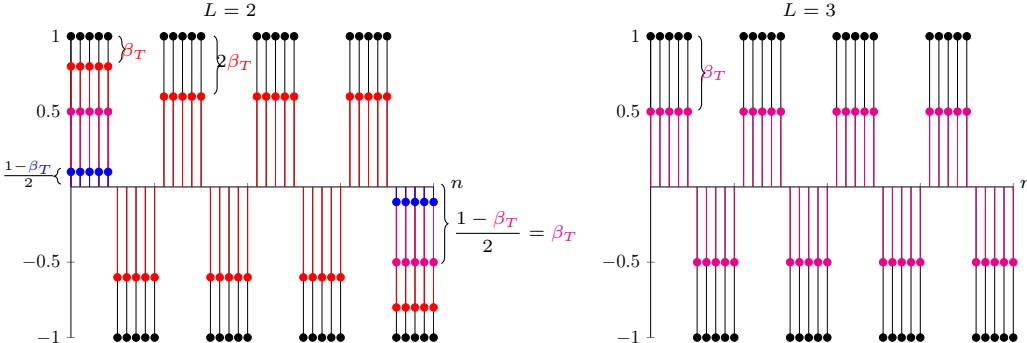

Figure 1: Each of the two figures depicts $(n, y_n)$ with black dots, where $\mathbf{y} = \mathbf{h}^{(T)}$ with $T = 10, N = 40$. Sign-activated neural net predictions are depicted as $(n, f_L(x_n; \theta))$ with blue, magenta, and red dots for $\beta_T = \frac{4}{5} \in \left[\frac{1}{2}, 1\right], \beta_T = \frac{1}{2}$, and $\beta_T = \frac{1}{5} \leq \frac{1}{2}$, respectively.

**Corollary 3.15.** *Consider a 1-D, 3-layer parallel sign network. There exists an equivalent 2-layer network with $m_2' = m_1 m_3$ neurons. Let $p_{L,\beta}^*$ be the optimal value of the training problem (4) for L layers, regularization $\beta$ and sign activation. Then, for 2-layer nets trained with $m_2' \geq m_1 m_2$ neurons, $p_{L=3,\beta}^* \leq p_{L=2,\beta}^* \leq p_{L=3,m_1\beta}^*$.*

Corollary 3.15 states that a 3-layer net can achieve lower training loss than a 2-layer net, but only while its regularization $\beta$ is at most $m_1$ times stronger. Our theory for 1-D data provides a lens to analyze higher dimensional data. We next give an extension to 2-D data on the upper half plane.

## 3.1 Example of rectangular sign networks on 2-D data

The next result extends the Lasso equivalence with dictionary matrices for rectangular 1-D networks to 2-D data on the upper half plane. A *bias-free* parallel network has no internal bias parameters, that is, $\mathbf{b}^{(i,l)} \notin \theta$ (they can be interpreted as set to 0). We denote the $n^{\text{th}}$ 2-D training point as $\mathbf{x}^{(n)}$ instead of $x_n$ to avoid confusing $x_n$ with the $n^{\text{th}}$ coordinate of $x$. Proofs are located in Appendix E.5.

**Theorem 3.16.** *The training problem for a bias-free, rectangular sign network with $L \geq 2$ layers trained on 2-D data with unique angles in $(0, \pi)$ is equivalent to a Lasso problem whose dictionary is the switching set $\mathbf{H}^{(1)}$ for $L = 2$ and $\mathbf{H}^{(m_1)}$ for $L > 2$, provided $m_L \geq \|\mathbf{z}^*\|_0$. The 2-layer dictionary can be expressed as $\mathbf{A}_{i,j} = \sigma\left(\angle\left(\mathbf{x}^{(i)} - \mathbf{x}^{(j)}\right)\right)$.*

In the 2-D setting of Theorem 3.16, the number of switches in a Lasso feature are the same as in the case for 1-D data in Corollary 3.1 and Theorem 3.4. While the data in Theorem 3.16 must have distinct angles, they can be located at any radius from the origin, distinct or not. The next result reconstructs an optimal 2-layer neural net from the Lasso problem using a dictionary defined by $\mathbf{A}_{i,j}$ in Theorem 3.16. Let $\mathbf{R}_{\frac{\pi}{2}} = \begin{pmatrix} 0 & -1 \\ 1 & 0 \end{pmatrix}$ be the counterclockwise rotation matrix by $\frac{\pi}{2}$.

**Lemma 3.17.** *When $L = 2$, the unscaled versions of $\alpha_i = z_i^*, \mathbf{s}^{(i,1)} = 1, \mathbf{W}^{(i,1)} = \mathbf{R}_{\frac{\pi}{2}}\left(\mathbf{x}^{(i)}\right)^T$, for $z_i^* \neq 0$ are optimal for the 2-D training problem in Theorem 3.16. Setting neurons to zero for $z_i^* = 0$ is optimal.*

Definition 3.11 defined unscaling. Extending Lemma 3.17 to more general data in all of $\mathbb{R}^2$ and higher dimensions is an area for future work. The next remark summarizes the expansion of libraries over depth.

**Remark 3.18.** *The dictionary for an architecture discussed in Lemma 3.2 or Theorem 3.16 is a superset of any dictionary with the same architecture but shallower depth.*

The next section uses the Lasso formulation to analyze binary, periodic data in 1-D.

# 4 Solution path for sign activation and 1-D data with binary, periodic labels

To illustrate applications of the Lasso equivalence, we study the case of a *square wave* target vector consisting of binary, periodic values, formally defined below. Binary-valued data arises in temporal sequences such as binary encodings of messages communicated digitally (42) and neuron firings in the brain (43), (7). While these real-world sequences are generally aperiodic, focusing on periodic target data allows us to derive a closed-form solution path for the Lasso problem. By the equivalence established in Section 3, the Lasso solution path maps to a sequence of globally optimal non-convex networks and their optimal performance loss as the regularization parameter $\beta$ varies, providing intuition into network behavior for more general data.

Let $T$ be a positive, even integer that divides $N$, the number of training points. A square wave with period $T$ (and amplitude 1) is a binary vector $\mathbf{h}^{(T)} \in \{-1, 1\}^N$ consisting of alternating runs of 1 and $-1$, each of length $\frac{T}{2}$, starting with 1. This corresponds to a partition of the real line into regions of alternating labels, each containing $\frac{T}{2}$ samples. The black samples in Figure 1 illustrate an example of a square wave.

In this section, assume the target vector is fixed as $\mathbf{y} = \mathbf{h}^{(T)}$ but the 1-D samples $x_1 < \cdots < x_N$ are arbitrary. There is a *critical value* $\beta_c = \max_{n \in [N]} |\mathbf{A}_n^T \mathbf{y}|$ of the regularization parameter $\beta$ such that when $\beta > \beta_c$, the all-zero solution $\mathbf{z}^* = \mathbf{0}$ is optimal in the Lasso problem (14). Let $\beta_T = \frac{\beta}{\beta_c}$ be the regularization parameter normalized with respect to this critical value. Let $k = \frac{N}{T}$ be the number of cycles in a square wave of period $T$.

First we consider the Lasso problem for a 2-layer sign network with 1-D inputs, which has dictionary as defined in Corollary 3.1. Proofs in this section are deferred to Appendix E.5.

**Theorem 4.1.** *The Lasso problem for a 2-layer sign network with 1-D inputs and target vector $\mathbf{y} = \mathbf{h}^{(T)}$ (the square wave target vector of period $T$) has critical value $\beta_c = T$ and unique solution given by*

$$z^*_{\frac{T}{2}i} = \begin{cases} 0 \qquad\quad \text{for all } i & \text{if } \beta_T \geq 1 \\ \begin{cases} \frac{1}{2}(1 - \beta_T)_+ & \text{if } i \in \{1, 2k-1\} \\ 0 & \text{else} \end{cases} & \text{if } 1 \geq \beta_T \geq \frac{1}{2} \\ \begin{cases} 1 - \frac{3}{2}\beta_T & \text{if } i \in \{1, 2k-1\} \\ (-1)^{i+1}(1 - 2\beta_T) & \text{else} \end{cases} & \text{if } \beta_T \leq \frac{1}{2}. \end{cases} \tag{5}$$

*for $i \in [2k - 1]$ and $z^*_n = 0$ at all other $n \in [N]$.*

Theorem 4.1 gives a closed-form expression for the 2-layer, square wave Lasso solution. When the regularization is strong as $\beta_T \geq 1$, the network selects the simplest solution: the all-zero function. For the moderate regularization regime $T \geq \beta \geq \frac{T}{2}$, the optimal solution $\mathbf{z}^*$ is supported on only two indices, specifically $z_i^*$ is nonzero only at the first index and the $\frac{T}{2}(2k - 1) = (N - \frac{T}{2})^{\text{th}}$ index. Under moderate regularization, the network focuses on representing the cycles defining the boundary of the data. As the regularization weakens ($\beta \leq \frac{T}{2}$), the solution remains sparse, with non-zero $z_i^*$ occurring at intervals of $\frac{T}{2}$ indices. The network may be interpreted as learning the key transition points when the square wave changes value. The next result gives the optimal network corresponding to the Lasso solution of Theorem 4.1. Assume the data is indexed in descending order as $x_i > x_j$ if $i < j$ (Section 2.1).

**Corollary 4.2.** *For a target vector $\mathbf{y} = \mathbf{h}^{(T)}$, an optimal 1-D, 2-layer sign network takes the form*

$$f_2(x; \theta) = 0, \qquad\qquad\qquad\qquad\qquad\qquad\qquad\qquad\qquad \text{if } \beta_T \geq 1$$

$$f_2(x; \theta) = \begin{cases} -(1 - \beta_T) & \text{if } x < x_{N - \frac{T}{2}} \\ 0 & \text{if } x_{N - \frac{T}{2}} \leq x < x_{\frac{T}{2}} \\ 1 - \beta_T & \text{if } x \geq x_{\frac{T}{2}} \end{cases} \qquad\qquad\qquad \text{if } \frac{1}{2} \leq \beta_T \leq 1$$

$$f_2(x; \theta) = \begin{cases} -(1 - \beta_T) & \text{if } x < x_{N - \frac{T}{2}} \\ (-1)^i(1 - 2\beta_T) & \text{if } x_{\frac{T}{2}(i+1)} \leq x < x_{\frac{T}{2}i}, \quad i \in [2k - 2] \\ 1 - \beta_T & \text{if } x \geq x_{\frac{T}{2}} \end{cases} \qquad \text{if } \beta_T \leq \frac{1}{2}$$

While Theorem 4.1 gives the Lasso solution, Corollary 4.2 gives an explicit, reconstructed optimal neural network function at all inputs $x$. The left panel of Figure 1 depicts an example of training data $(x_n, y_n)$ in black and overlayed with various reconstructed optimal fits $f_2(\mathbf{X}; \theta)$ in different colors. By Corollary 3.10, the reconstructed network interpolates the optimal fit in a piecewise constant manner. The resulting fit depends on the three regimes for $\beta$ in Corollary 4.2.

In the high regularization regime ($\beta > T$), an optimal neural net is the constant zero function, which is the mean of the $\pm 1$ target values in $\mathbf{h}^{(T)}$. As $\beta$ decreases into the moderate regime ($\frac{T}{2} \leq \beta \leq T$), the network vanishes on the central segment $[x_{N-\frac{T}{2}}, x_{\frac{T}{2}}]$ but maintains a non-zero magnitude outside this region. The blue samples in Figure 1 illustrate this fit. The magnitude $\frac{1-\beta_T}{2}$ of the outer segments increases as $\beta$ decreases, reaching $\frac{1}{2}$ at the transition point $\beta_T = \frac{1}{2}$. The magenta samples in Figure 1 depict this transitional fit. In the weak regularization regime ($\beta \leq \frac{T}{2}$), the optimal fit $f_2(\mathbf{X}; \theta)$ takes the form of a square wave of period $T$ over the central segment $[x_{N-\frac{T}{2}}, x_{\frac{T}{2}}]$. The regularization damps the square wave's amplitude, producing a fit of height $1 - 2\frac{\beta}{T}$, as shown by the red samples in Figure 1. This fit converges to $\mathbf{y}$ as $\beta \to 0$.

As established by Theorem 4.1 and Corollary 4.2, and visualized in Figure 1, the solution path of the 2-layer Lasso evolves continuously from the zero function to a perfect fit as the regularization strength $\beta$ decreases. We now investigate how the Lasso solution path changes for 3 layers. By Lemma 3.2 and Corollary 3.1, while 2-layer features possess only 1 switch, 3-layer features contain $m_1$ switches, enabling the network to capture more complex, global properties of the data. The next result demonstrates this behavior for the square wave. Denote $(x)_+ = \max\{0, x\}$, and recall that $m_3$ is the number of parallel units in a 3-layer network.

**Theorem 4.3.** *Consider a Lasso problem for a 1-D, 3-layer, parallel sign network with $m_3 \geq 2k - 1$ and target vector $\mathbf{y} = \mathbf{h}^{(T)}$. The critical value is $\beta_c = N$. There exists an index $i$ such that the corresponding feature is $\mathbf{A}_i = -\mathbf{h}^{(T)}$, with a unique optimal solution given by $z_i^* = -(1 - \beta_T)_+$ and $z_n^* = 0$ for all $n \neq i$.*

Theorem 4.3 gives an exact, 1-sparse, closed-form Lasso solution for a 3-layer parallel network with at least twice the number of neurons as periods $k$ in $\mathbf{y}$. The condition $m_3 \geq 2k - 1$ ensures that the 3-layer dictionary is sufficiently richer than its 2-layer counterpart, enabling the 3-layer network to model the entire target vector with a single feature. Accordingly, only the entry $z_i^*$ corresponding to the feature $-\mathbf{h}^{(T)}$ is non-zero. When $\beta$ exceeds the critical value $N$, $z_i^* = (1 - \beta_T)_+ = 0$ and the all-zero solution is optimal. The following result specifies the corresponding optimal 3-layer network as a function of $\beta$.

**Corollary 4.4.** *Let $x_0 = \infty$. For target vector $\mathbf{y} = \mathbf{h}^{(T)}$, an optimal 1-D, 3-layer, parallel sign network with $m_3 \geq 2k - 1$ is given by*

$$f_3(x; \theta) = \begin{cases} (1 - \beta_T)_+ (-1)^{(i-1)} & \text{if } x_{\frac{T}{2}i} \leq x < x_{\frac{T}{2}(i-1)}, \quad i = 1, \cdots, 2k - 1 \\ -(1 - \beta_T)_+ & \text{if } x < x_{N-\frac{T}{2}} \end{cases}$$

The $x_0 = \infty$ notation ensures that for $i = 1$, $f_3(x; \theta) = (1 - \beta_T)_+$ for all $x > x_{\frac{T}{2}}$. Corollary 4.4 gives an explicit, optimal 3-layer parallel network with sufficient width. When $\beta > N$, the constant zero function is an optimal network. For $\beta < N$, an optimal fit is a damped square wave $f_3(\mathbf{X}; \theta) = (1 - \beta_T)_+ \mathbf{h}^{(T)}$, as depicted by the magenta samples in the right panel of Figure 1. Since only one parallel unit is active in this network, the network can also be expressed as a standard (non-parallel) network.

As illustrated in Figure 1, Corollary 4.2 and Corollary 4.4 reveal distinct ways in which 2 and 3-layer parallel sign networks respond to regularization when fitting periodic targets. suggest that as the regularization increases, 2-layer sign networks focus on preserving the boundary points of the data (first and last $T$ points) to closely match the square wave target vector, and reduces the magnitude of the rest of the network. Therefore the network may generalize well if there is noise that corrupts the middle of the data, and the edges of the data are more important. In contrast, if the data is noiseless or noise occurs uniformly over the data, the 3-layer parallel sign network will generalize well as its amplitude reduces uniformly over $x$, and may demonstrate higher robustness against fitting local artifacts. The next section explores numerical experiments.

## 5 Numerical Results

This section describes experiments related to our theory for convex equivalence and solution sets. Due to space, additional experiment details and plots are deferred to Appendix B.

### 5.1 Simulations for Section 3

We compare training with the non-convex problem versus the convex Lasso on 1-D data for 2-layer sign networks. The training and testing data for the $x$ values each consist of $N = 1000$ points uniformly randomly sampled between $-10$ and $10$. A 2-layer network with $N$ neurons and sign activation is initialized with random weights and planted in the data, i.e., the network's outputs on the training data $x_n$ generate the target values $y_n$. The planted network ensures that training a neural network with at least $N$ neurons in each layer to exactly fit the data is possible, and moreover optimal if $\beta = 0$. The experiment investigates how well the planted network's fit can be recovered by training a 2-layer network from random initializations using the same number of neurons ($m_2 = N$) with $\beta = 0$. Setting $\beta = 0$ establishes a known optimal training loss of 0 and also allows evaluation of the Lasso model's utility in non-regularized settings that may be common in practice. Letting $m_2 = N$ also ensures equivalence between the training problem and corresponding Lasso. We train 2-layer Lasso models with the dictionary described in Corollary 3.1 and reconstruct networks as described in Lemma 3.12.

The first row of Figure 2 plots loss curves starting from 20 different initializations for the non-convex training. Figure 2 shows that the convex Lasso network reaches the optimal training loss of 0 and outperforms the non-convex trained networks in both training and testing error. We extend our non-convex training experiments to 3 and 4-layer standard networks with $m_l = N$ neurons in each layer and compare them against the original, 2-layer convex training. While standard and parallel architectures coincide for 2 layers, they diverge as depth increases. Because our theory guarantees that training the Lasso model reaches global optimality for a parallel network, we use these simulations to investigate how the Lasso model performs relative to standard networks that may be typically used in applications. To establish a conservative comparison, we evaluate the deeper non-convex training against the simple 2-layer convex network. Since 2-layer models are a subnetwork of deeper models, the optimal training objective of the 2-layer Lasso model upper bounds that of deeper architectures.

The second and third rows of Figure 2 repeat the first row for depth 3 and 4, respectively and their plots (especially their zoomed-in versions on the right) show that the 2-layer convex training still outperforms the deeper non-convex training in training error and typically in testing error as well. The non-convex training uses a straight-through-estimator (STE) (11) to handle the sign activation. The convex training avoids activation approximations and hyperparameter tuning, and finds a globally optimal solution to the training problem, leading to its improved performance.

We expect the convex Lasso training of 2-layer networks to be equivalent to global optimization of the non-convex 2-layer networks, which are shown in the first row of Figure 2. In the second and third row of Figure 2 we do not expect the Lasso for 2-layer networks to be equivalent to the non-convex training of 3 and 4-layer networks, respectively. Rather, since deeper networks have no lower expressivity than shallower networks, we expect a globally optimal fit of non-convex 3 and 4-layer networks to be no worse than global optimization of 2-layer networks, which is equivalent to that of Lasso for 2-layer networks. Hence if the non-convex training in the second and third row of Figure 2 were able to reach a global optimum, we would expect its training loss to be no larger than that of the 2-layer Lasso model.

Consistent with this intuition, the gap in Figure 2 between the convex 2-layer network and deeper non-convex training narrows with depth, as expected for deeper models improving upon shallower models. However, the gap is still positive: the non-convex training of 3 and 4-layer networks still underperforms 2-layer Lasso training. Hence the narrowing gap in Figure 2 between the convex 2-layer network and deeper non-convex training is qualitatively consistent with the greater expressivity of deeper networks. This gap is neither predicted by nor contradicts our theory, which compares Lasso and non-convex training for networks of the same depth. Future directions can directly compare non-convex training with convex training of networks of the same depth. Nonetheless, in Figure 2 the optimization for the non-convex networks appears substantially

suboptimal relative to the Lasso optimum such that even increasing depth to 3 or 4 layers does not outperform the global optima found by 2-layer Lasso.

For the 2-layer experiment in the first row of Figure 2, the theory (Lemma 2.1) predicts exact equivalence between the Lasso problem and global optimization of the neural network. Therefore any observed gap is most likely due to the non-convex training not reaching a global optimum. Possible causes include the following. First, the non-convex model finds suboptimal solutions while the convex model globally optimizes. Second, the use of sign activation can make gradient descent challenging. This is because sign activation is discontinuous and piecewise constant, with zero gradient at all but one point where the gradient does not even exist. Third, related to the second reason, the non-convex training approximates backpropagation of the sign activation using STE (11). Also while we varied the learning rate and initialization, the experiment only tried a limited, finite set of these hyperparameters and they, along with optimization technique (e.g., Adam vs SGD) may require further tuning for the non-convex network. In the comparisons between 2-layer convex versus deeper non-convex training, where the training objectives differ, the above factors also contribute to the gap and its direction, with the 2-layer convex training having better performance.

Therefore since the non-convex training is suboptimal, the gap between the convex and non-convex training itself does not definitively prove that the Lasso model finds a global optimum, only that it finds a better solution. However, as described in the first paragraph of Section 5.1, since $\beta = 0$ the optimal training loss is known to be 0, which the Lasso model appears to closely approach. This suggests the Lasso model may globally optimize the training loss, which supports the theory. Our theory only guarantees that the Lasso model optimizes the training loss, and does not guarantee optimality of the test loss. Experimental verification of the theory could be strengthened through additional datasets, larger hyperparameter searches, and more variation in network depths and widths.

## 5.2 2-D Experiments

We train 2-layer sign networks on a 2-D spiral dataset shown in Figure 3, where red and green training data correspond to labels 1 and $-1$, respectively. Networks are trained with regression and $\beta = 0$. We compare non-convex training versus convex training, which uses the Lasso model in Theorem 3.16 and reconstructs a network using Lemma 3.12. The non-convex training uses STE (11) and Adam with a learning rate of $10^{-3}$ for 500 epochs. Nonnegative outputs of the network on the training data are interpreted as label 1 and colored in red in Figure 8 while negative outputs are in green representing $-1$. Figure 8 shows that binary classification using the convex training achieves higher training accuracy than the non-convex training, which supports Theorem 3.16. The convex training allows global optimization while the non-convex training can get stuck in local minima. The next section compares 2 and 3-layer Lasso models for neural networks.

## 5.3 Simulations for Section 4

This section verifies our solution path theory for binary periodic data in Theorem 4.1 and Theorem 4.3 by solving the Lasso problem for 2 and 3-layer parallel sign networks on sample 1-D training data with square wave target vector $\mathbf{y} = \mathbf{h}^{(T)}$. The 1-D training samples $x_n$ are chosen randomly from a uniform distribution on $[-100, 100]$. 2-layer and 3-layer parallel sign networks are trained using the convex Lasso problem. The target vector $\mathbf{y}$ for the 2-layer network is a square wave with $N = 40$ points, consisting of $k = 4$ periods and $T = 10$ points in each period. The target vector $\mathbf{y}$ for the 3-layer network is a square wave with $N = 30$ points, consisting of $k = 3$ periods and $T = 10$ points in each period. The bottom row of Figure 4 graphs the training data $(x_n, y_n)$ as stem plots. The figure plots positive target values $y_n = 1$ in blue, and negative target values $y_n = -1$ in red.

To evaluate the solution path, we repeatedly train the networks across a range of $\beta_T$ regularization values. The top row of Figure 4 displays these results as heatmaps, where each row represents the network fit for a specific $\beta_T$. As indicated by the colorbar, darker shades represent higher magnitudes of the network output, and white indicates a value of 0. The blue and red shaded regions in each row indicate positive and negative outputs, respectively. For each $\beta_T \in (0, \beta_c)$, the 3-layer net (top right panel) maintains a uniform magnitude over all $x$, as predicted by Corollary 4.4. However, the 2-layer net is spatially biased toward a stronger prediction on the first and last intervals, as consistent with Corollary 4.2. For this noiseless training data, the

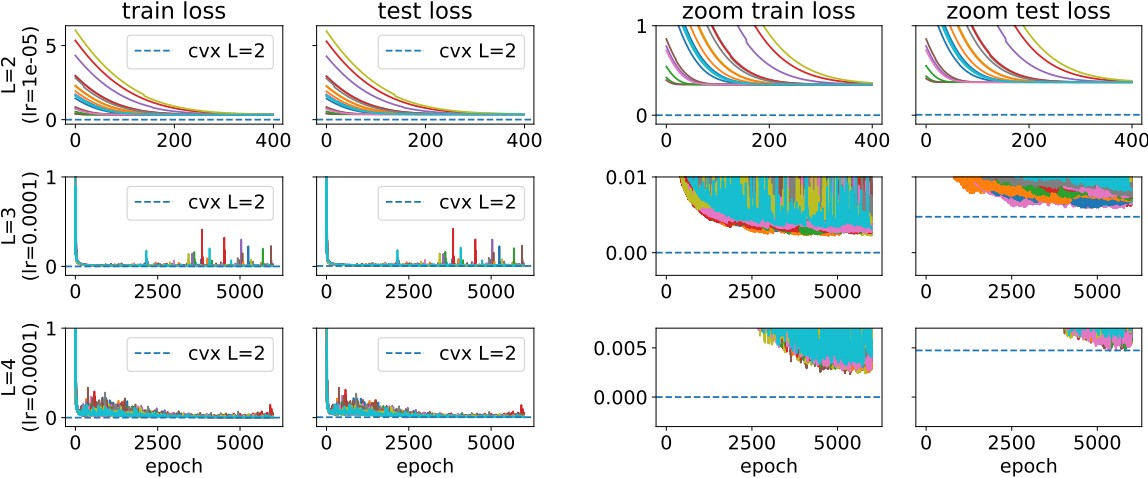

Figure 2: Neural network performance on random 1-D data. Solid curves: each curve is the non-convex training loss for a different random initialization (20 runs total). Since all runs are shown individually, the figure visualizes the full empirical distribution of outcomes rather than an aggregate statistic. Dashed line: convex loss.

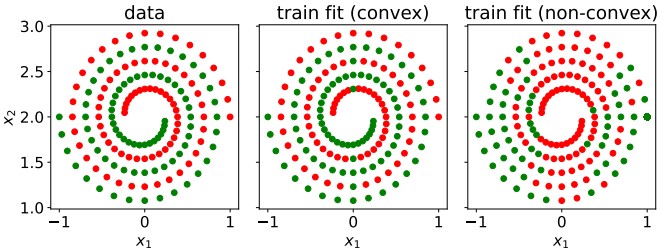

Figure 3: Binary classification of 2-D data by networks with convex and non-convex training. Binary labels are indicated by red and green colors.

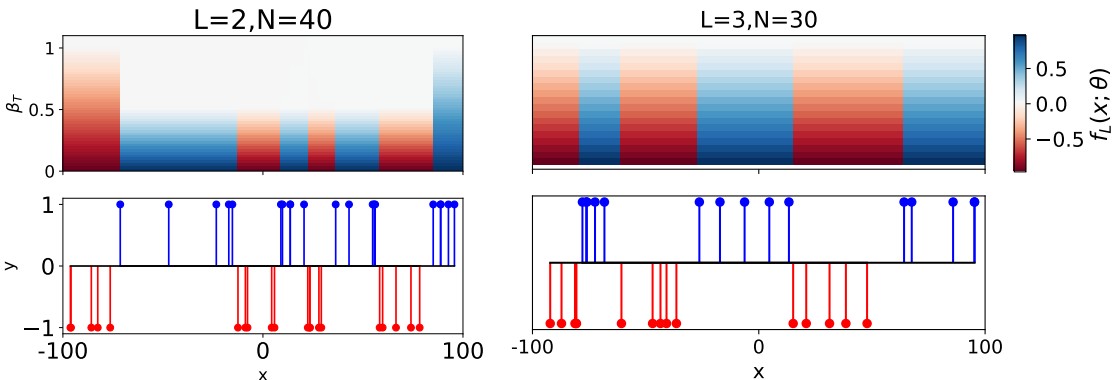

Figure 4: The bottom figures plot training data $(x_n, y_n)$. The top figures plot sign network predictions by color for each $x$, as parameterized by $\beta$ on the vertical axis.

uniform scaling of the 3-layer network suggests a better fit and generalization. The consistent solution path of the 3-layer network also makes tuning $\beta$ more straightforward.

Consider a binary classifier that outputs a label 1 if $f_L(x; \theta) > 0$, $-1$ if $f_L(x; \theta) < 0$, and 'undecided' if $f_L(x; \theta) = 0$. In the top row of Figure 4, the red, blue and white shaded regions represent decision regions classified as $-1$, 1, and "undecided," respectively. Training points with target value $y_n = 1$ (blue samples) in the bottom row consistently align with blue or white decision regions in the top row, while points with $y_n = -1$ (red samples) align with red or white regions. In particular, the 2-layer and the 3-layer networks appear to correctly classify all training data for $0 < \beta_T < \frac{\beta_c}{2}$ and $0 < \beta < \beta_c$, respectively. For $\beta_c/2 < \beta_T < \beta_c$, the 2-layer net correctly classifies the data in the first and last half-period, but is undecided on the interior region.

## 6 Conclusion

Our results show that deep rectangular and tree neural networks with sign activation trained on 1-D data with weight regularization can be recast as convex Lasso models with explicit dictionary matrices. This can provide insight into their solution path and applications to certain 2-D data. Appendix C provides additional discussion including uniqueness and complexity, due to space. Limitations of this work include the assumption of 1-D or certain 2-D data, the rectangular and tree architectures, and the sign activation. Our analysis provides a foundation for studying more general architectures and dimensions.

## Acknowledgements

This work was supported in part by the National Science Foundation (NSF) CAREER Award under Grant CCF-2236829, in part by the National Institutes of Health under Grant 1R01AG08950901A1, in part by the Office of Naval Research under Grant N00014-24-1-2164, in part by the Defense Advanced Research Projects Agency under Grant HR00112490441, and in part by the National Science Foundation Graduate Research Fellowship under Grant No. DGE-1656518.

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

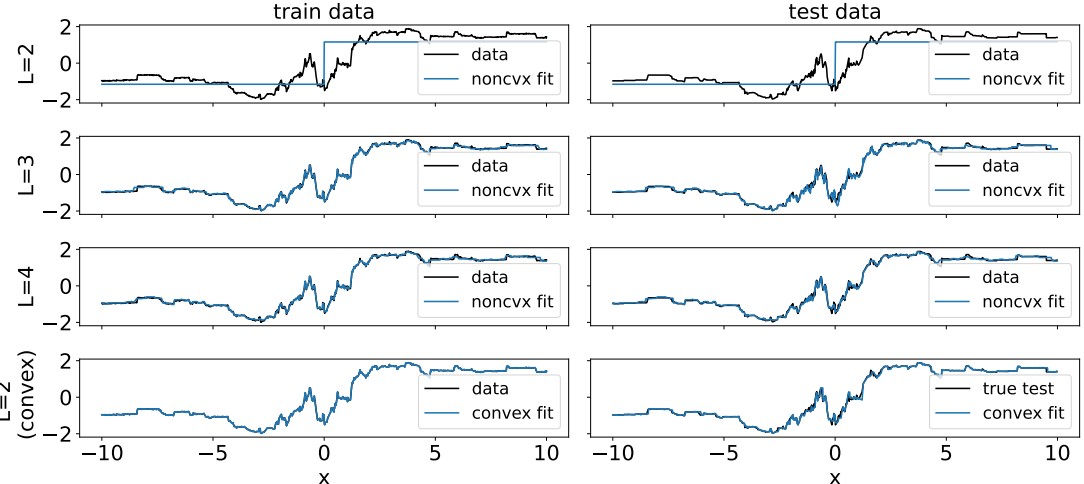

Figure 5: Blue: best network fit (out of 20 initializations) when trained with non-convex problem in first 3 rows, and with convex problem in the bottom row. Black: true data from planted network.

## Appendix

## A Comparison to (12)

(12) describes the reconstruction of an optimal network from a Lasso solution implicitly. Even for 2 layers and the special case where the data 'shatters,' i.e., the dictionary consists of all binary vectors of a fixed length, resulting in a simplified Lasso problem, the reconstruction invokes Cartheodory's theorem to help show the existence of optimal weights rather than an explicit construction. Because of the large, implicit number of features, (12) randomly subsamples features in their experiments. While this still performs better than non-convex training, it results in an approximation of a globally optimal solution. Our results for rectangular networks limit the feature complexity (Remark 3.7Lemma 3.2) while finding a global optimum. The implicit feature approach in (12) also results in ambiguity over multiple reconstructed parameters that can result in the same network, so (12) uses heuristics such as training support vector machines in its experiments to choose parameters with better generalization. In contrast, we provide explicit, straightforward features and reconstruction formulas for rectangular networks (Lemma 3.12 for 3 layers).

## B Experiment details for Section 5

### B.1 Additional plots for Section 5.1

Out of the 20 initializations for each network depth and selected learning rate, Figure 5 plots the corresponding fit (blue) for the network with the lowest training mean squared error (MSE). Figure 5 shows that 2-layer convex network fits the training data more accurately than 2-layer and deeper non-convex networks.

The 2-layer non-convex training was repeated with 10 different random datasets, shown in Figure 6. In all such datasets, the non-convex training failed to accurately learn the data. This demonstrates challenges with non-convex training and STE approximations (11) for sign activation that the convex training bypasses. It also suggests that depth may improve training for non-convex methods and STE, perhaps by smoothening the loss landscape with increased parameters, preventing gradient descent from getting stuck in particularly suboptimal local minima.

Figure 7 plots the training time for each non-convex training for the best learning rate, for depths $2, 3, 4$. Although the 2-layer non-convex training and convex training have comparable training times, the 2-layer network shows poor performance as shown in Figure 5. While the 3 and 4-layer non-convex networks fit the

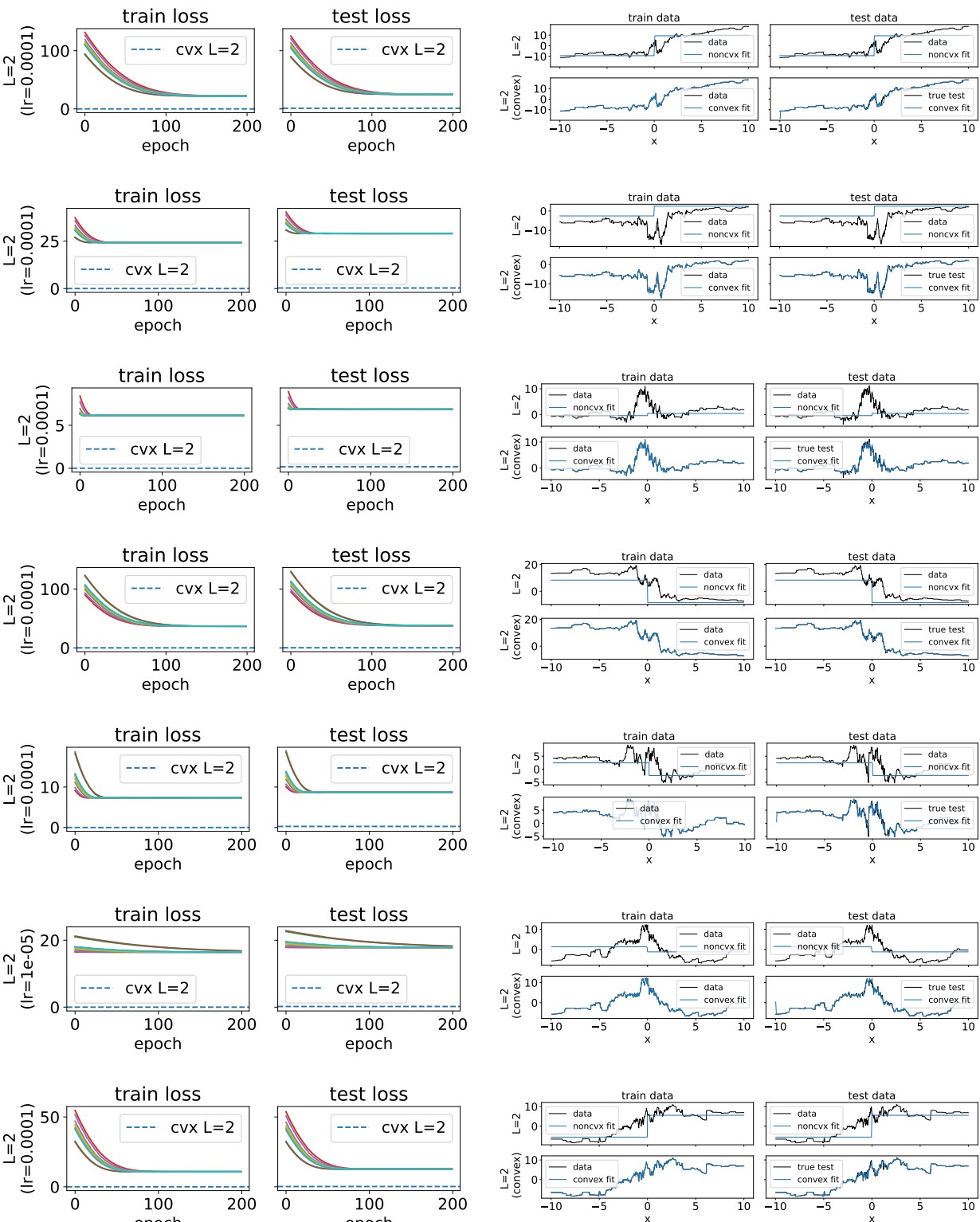

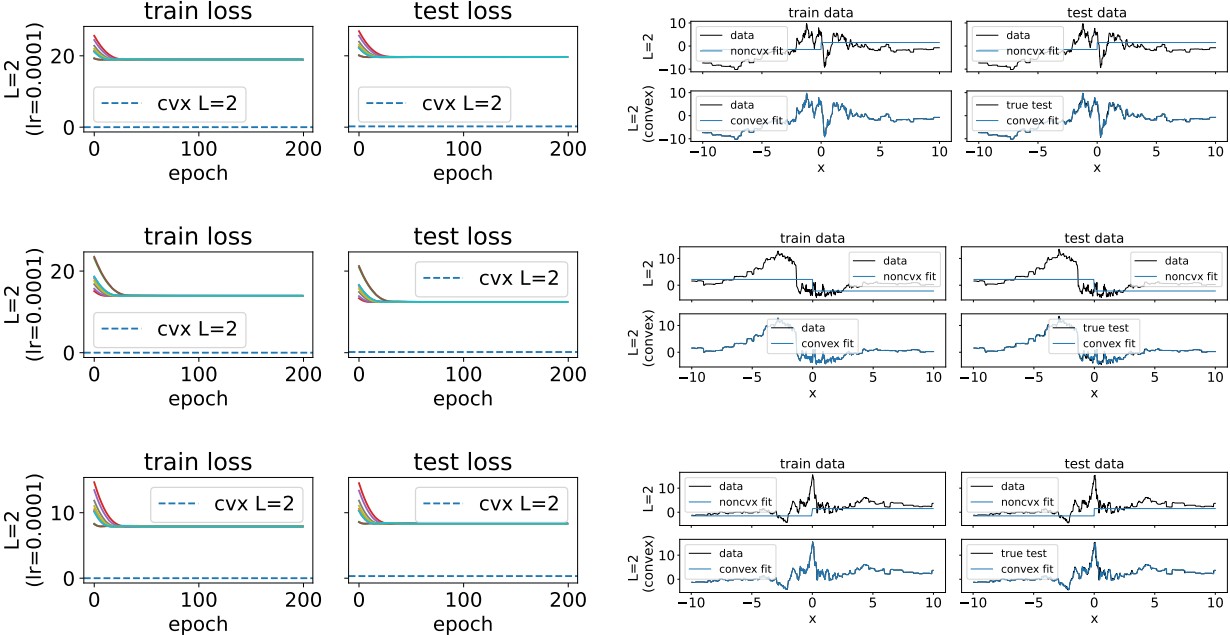

Figure 6: Continued from page above: 2-layer sign networks trained on different data.

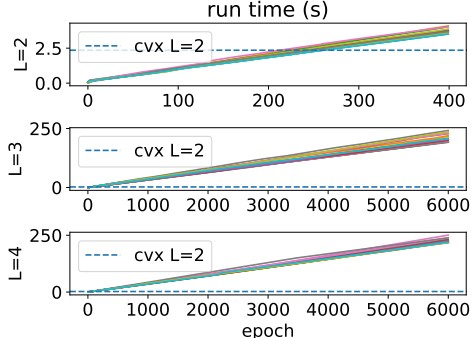

Figure 7: Training time of 1-D experiments in Figure 2 for non-convex training with 20 initializations in solid curves, and convex training in the dashed curve.

data significantly better than 2-layer networks, their training until convergence takes significantly longer than the convex Lasso, and still the Lasso outperforms them in training error as shown in Figure 2.

## B.2    Details for Section 5.1

The 2-layer Lasso problem uses dictionary as described in Corollary 3.1. While (31) finds the dictionary for 2-layer networks with sign activation, (31) lacks any experiments with them, convex or non-convex, of any depth, nor discuss solution paths.

The non-convex training uses 1000 epochs. The non-convex training is repeated for 20 different random weight initializations using the Adam optimizer and MSE (mean squared error) loss, for each of the learning rates $10^{-3}, 10^{-4}, 10^{-5}, 10^{-6}$. The first row of Figure 2 plots the 20 loss curves corresponding to the learning rate with the lowest training MSE across initializations. This optimal learning rate turned out to be $10^{-4}$ for all networks. Out of the 20 initializations shown in Figure 2, the first row of Figure 5 plots the corresponding fit (blue) for the network with the lowest training MSE.

The Lasso problem's training performance and training time do not depend on epochs, so Figure 2 shows the Lasso model's final error by a dashed blue horizontal line and Figure 7 plots the total training time for the Lasso problem as a horizontal line. The true data in plotted in black in Figure 5 for comparison with the fit.

## B.3    Details for Section 5.3:

The 2-layer Lasso uses a dictionary $\mathbf{H}^{(1)}$ corresponding to $N = 40$ neurons, as prescribed by Corollary 3.1. The 3-layer network uses a dictionary $\mathbf{H}^{(m_1)}$ corresponding to $m_1 = 2k - 1$ neurons by Lemma 3.2. We sample 50 values of $\beta_T$ for the 2-layer network and and 20 points for the 3-layer network.

## B.4    2-D Experiments

The number of non-zero $z_i^*$ in the solution found for the Lasso problem is used as the number of neurons for the non-convex training to ensure equivalence between the network and the Lasso model. The non-convex training is repeated for learning rates $10^{-2}, 10^{-3}, 10^{-4}$ and 20 initializations for each rate. Figure 9 shows the 20 loss curves for each learning rate in the non-convex training. The learning rate with the lowest training loss out of all initializations was $10^{-3}$ and its network fit is plotted in Figure 3 and Figure 8. The convex training loss from solving the Lasso model with cvxpy does not depend on epochs and is plotted as a horizontal dashed line in Figure 9. The convex training loss is lower than that of the non-convex training as consistent with Theorem 3.16.

The convex training misclassifies only a few points, while the non-convex training misclassifies entire sectors of spirals. The plots in Figure 3 are a subset of those in Figure 8, which shows the test performance in addition to the training performance. As with the training fits in Figure 3, nonnegative outputs of the network on the test data in Figure 8 are interpreted as label 1 and colored in red while negative outputs are interpreted as $-1$ and colored in green. The test data results shown in the third column of Figure 8 reveal the reasons for the training misclassification pattern in the non-convex training. Due to the sign activation, the output of the first layer $\sigma(\mathbf{x}\mathbf{w})$ without bias parameters depends only on the direction of $\mathbf{x}$ and is invariant its magnitude. Hence the decision regions consist of sectors radiating from the origin, as shown in the third column. The convex training achieves higher training accuracy because it is able to more finely position the decision regions to align with the training data. This experiment also illustrates the limitations of not including bias parameters and demonstrates their importance especially for low-dimensional data. The Lasso formulation enables exact convex analysis of the network and provides a globally optimal solution that illustrates how the network may optimally fit the training data even when its architecture constrains the structure of its decision regions. A direction for future work is to analyze sign networks with higher dimensional inputs and bias parameters.

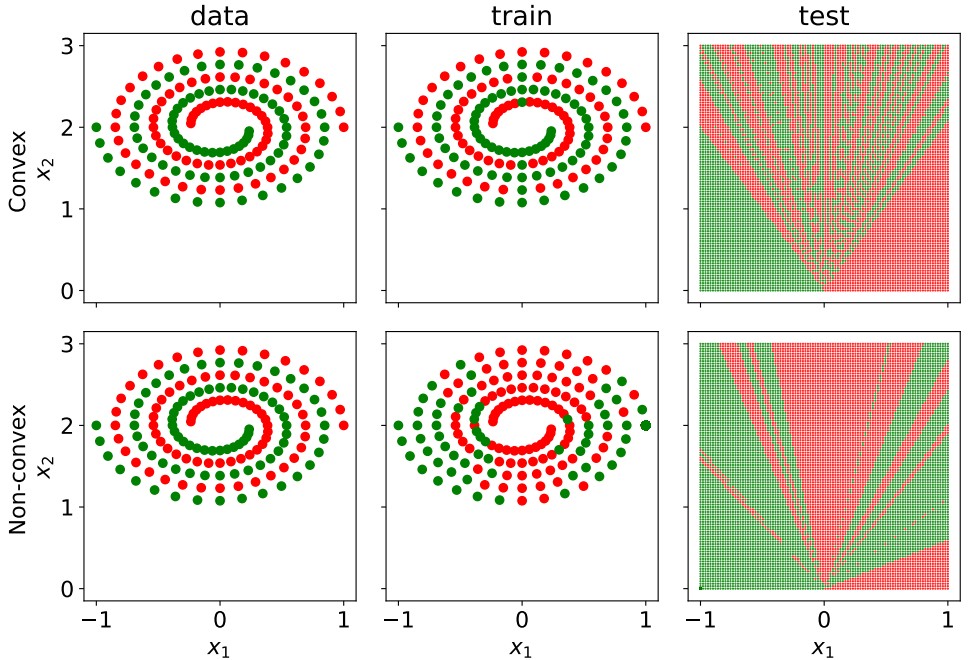

Figure 8: 2-layer networks trained on a 2-D spiral dataset. The convex training demonstrates better training performance.

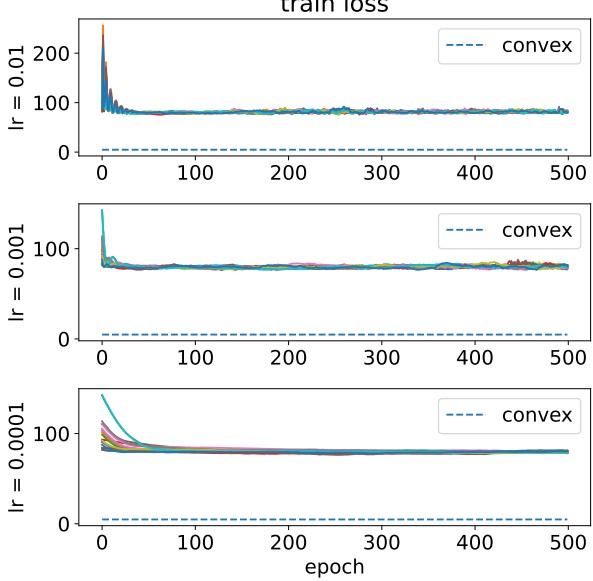

Figure 9: Solid colors show training loss curves of sign networks with non-convex training on the 2-D spiral dataset in Figure 8, over 20 initializations with different learning rates $(10^{-2}, 10^{-3}, 10^{-4})$. The dashed line is the training loss found by the convex Lasso. The learning rate with the lowest training loss was $10^{-3}$ and its corresponding fit is shown in Figure 8.

### B.5 Structured and unstructured feature sampling

As Lasso dictionaries grow with network widths and dataset sizes (Appendix C), feature sampling can reduce computational cost and make the Lasso problem more tractable in practice. Subsampling features approximates the Lasso problem, so there is a trade-off between increasing the number of sampled features to improve accuracy versus increasing the training complexity, explored in (44), (45). We consider two kinds of feature sampling.

*Structured sampling* selects vectors from the switching set $\mathbf{H}^{(m_1)}$ in Theorem 3.4 for parallel networks. While our explicit characterization of the Lasso dictionary in terms of a switching set holds for univariate data, the proof of Theorem 3.4 shows that the Lasso equivalence extends to multivariate data, with the dictionary given by all possible parallel unit outputs $\hat{\mathbf{X}}^{(i,L)}$ on $\mathbf{X}$. This perspective is analyzed in (12) for parallel networks with threshold activation. *Unstructured sampling* samples weights and biases from a Gaussian distribution and then generates parallel units as features, also studied in (12). We record the parameters used to generate the features, enabling reconstruction of an optimal network from a Lasso solution $\mathbf{z}^*$. For a parallel network these features are $\hat{\mathbf{X}}^{(i,L)}$ and a network is reconstructed as $f_L(\mathbf{x}; \theta) = \sum_{i=1}^{m_L} \hat{\mathbf{X}}^{(i,L)} z_i^*$ 3, where $z_i^*$ correspond to optimal $\alpha_i$. Alternative strategies to robustly recover parameters can be used as well (12). Unstructured sampling is related to similar Gaussian feature sampling approaches analyzed for standard 2-layer ReLU networks in (44), (45).

Similarly, unstructured sampling for tree networks generates parallel units $\sigma(\mathbf{X}^{(i)}\mathbf{w}^{(i)} + b^{(i)})$ (Appendix C.1) where amplitudes are 1 and weight and bias parameters are randomly sampled from a $\mathcal{N}(0,1)$ distribution. These parallel units correspond to features, as seen in the proof of Theorem 3.5. The sampled parameters are recorded to reconstruct a network $f_L(\mathbf{x}; \theta) = \mathbf{X}^{(\emptyset)} = \sum_i z_i^* \sigma(\mathbf{X}^{(i)}\mathbf{w}^{(i)} + b^{(i)})$ 6 from a Lasso solution $\mathbf{z}^*$ whose elements correspond to optimal parameters for $\alpha^{(i)}$.

Structured and unstructured feature sampling are experimentally explored in Appendix B.6, Appendix B.7. In the experiments, non-convex training with sign activations and STE (11) often converged to degenerate or suboptimal solutions (frequently collapsing to near-constant outputs), consistent with the optimization difficulties discussed in Section 5.1 and also seen in Figure 6. Therefore, we focus on the convex Lasso formulation, which by Theorem 3.4, Theorem 3.5 characterizes global optima and enables controlled study of network properties. Experiments in Appendix B.6, Appendix B.7 focus on networks trained with Lasso to investigate solution path, generalization, and robustness properties of globally optimal networks. These experiments investigate properties of the convex models such as network reconstruction and effects of feature sampling.

### B.6 Parallel networks on ECG data: sampling, reconstruction, and multi-dimensional variations

Experiments are performed on ECG time-series data from the MIT-BIH Arrhythmia Database (46). The goal is to autoregressively predict the ECG value $e_{t+1}$ at time $t+1$ from the ECG values $e_t, e_{t-1}, \cdots, e_{t-d}$ at times $t, t-1, \cdots, t-d$. In other words, the labels are $y_t = e_{t+1} \in \mathbb{R}$ and the training samples are of the form $x_t = (e_t, \cdots, e_{t-d}) \in \mathbb{R}^d$. The training data is generated by the first $N = 500 - d$ ECG values, and the test data consists of the next 500 ECG values. In all plots of this subsection, the top panel (which is the only panel for the 2-layer case in Figure 10) shows both the training fit and test fit. The training fit is plotted over the first $500 - d$ time steps and the test fit is plotted for the remaining time steps. The regularization parameter is $\beta = 10^{-6}$.

Parallel networks with sign activation and depths 2 and 3 are trained using the convex Lasso problem. The 2-layer parallel network uses the full $N \times N$ Lasso dictionary from Lemma 2.1, Corollary 3.1 and reconstructs a network using Lemma 3.12. The 2-layer convex fit is shown in Figure 10. The 3-layer networks use a subset of the dictionary in Theorem 3.4 with 1000 sampled features for computational tractability, generated via structured sampling when $d = 1$ and unstructured sampling for $d > 1$ (Appendix B.5.) The bottom panels in all figures in this section except Figure 10 show training and test error as a function of a hyperparameter such as $m_1$ or $d$. There are 10 initializations for each value of the hyperparameter, and the mean is plotted as solid dots while error bars indicate one standard deviation from the mean.

While the 2-layer network uses the full dictionary and the number of non-zero elements in a Lasso solution

gives an optimal number of neurons (Lemma 3.12), for 3-layer networks which subsample the dictionary, the optimal number of neurons $m_1$ is less well-defined. Figure 12 uses structured feature sampling and explores the effect of width $m_1$ when the number of sampled features is fixed. The bottom panels in Figure 12 show a decrease in training error as width increases. This is consistent with width increasing feature complexity, allowing the network to be more expressible and better fit the data. The test data also initially decreases, but it experiences a smaller change with width, and exhibits fluctuations. This may be due to larger widths increasing the size and complexity of the dictionary, such that sampling 1000 features becomes insufficiently representative of the dictionary and the network overfits with excessively complex features. Nonetheless, in both training and test MSE, the variance appears to decrease with width, suggesting improved training robustness and stability. The fit seen in the top panel also shows a closer fit to the data with less variance as the width increases.

When using structured sampling, we assume that all vectors in $\mathbf{H}^{(m_1)}$ are valid features, or parallel unit outputs. This requires the assumption that data points are all distinct. Because the values given in the ECG dataset were quantized, they were not all unique. So a small amount of random noise $\epsilon \sim \mathcal{N}(0, \sigma^2)$ where $\sigma = 10^{-6}$ was added to the ECG data in Figure 12 before generating the training and test $\mathbf{X}$ and $\mathbf{y}$ datasets. The noise was added before dataset generation to simulate inherent instrumentation or measurement noise in the entire dataset. The noise was only added for the structured feature sampling experiments in Figure 12. Figure 13 shows the same experiment without any noise added. The reduction in test loss and variance in the middle panel is less visible in Figure 13 than in Figure 12 and the corresponding fits in the top panels appear less accurate. However, Figure 13 shows that the convex training can still reasonably learn the noiseless ECG data. We note that "noiseless" quantized ECG data may still have implicit discretization artifacts or quantization noise due to finite measurement resolution. Adding a small amount of Gaussian noise can help smooth these effects.

Figure 11 repeats the experiment in Figure 13 but with unstructured feature sampling. Similarly to the structured feature sampling case, the training error decreases with width as expected. The fit in the top panel is better with unstructured sampling compared to structured sampling in Figure 13, Figure 12, but the test error does not noticeably decrease with width. Theoretical analysis of differences in behavior between structured and unstructured sampling is a direction for future work. While structured sampling from our theory-defined dictionary $\mathbf{H}^{(m_1)}$ performed worse, our theory offers theoretical understanding of the full explicit dictionary. The structured sampling occurs as follows. First, the number of switches, $k$ is selected uniformly from $[m_1]$. Then $k$ random switch locations are uniformly selected to generate a feature. This process is independently repeated 1000 times to generate 1000 random features. However, this does not produce a uniform distribution over features, since the number of features with $k$ switches varies with $k$. Better structured sampling techniques may improve their performance.

In all plots except Figure 14, $d = 1$. Figure 14 illustrates the network fit with unstructured sampling as the autoregression lag $d$ increases. As $d$ increases, the network fit of the training data improves, as seen in its decreasing training MSE of the bottom panels in Figure 14. This is because the network can use more ECG values from the past to predict the next ECG value. However, the training loss plateaus, showing diminishing returns as the lag grows larger. In Figure 14, the test loss is less monotonic and has high variance for $d = 5, 7$. The fit in the top panel also shows a gap between training and test performance. This may be because using larger $d$ makes the model more complex, resulting in overfitting. Another reason may be that as $d$ increases, the dictionary grows in size and complexity while the number of sampled features remains fixed (1000), so representative features may be missed, leading to overfitting to more complex features. While the top panel shows high variance in network fits, with some being poorer predictors of the data, it also shows fits that are more accurate as well, indicating that initialization can have a large impact. Reducing the variance and improving performance is an area for future work.

## B.7   4-layer tree networks on sinusoidal data: regularization path and effect of noise

We train 4-layer tree networks with sign activation using the convex Lasso to fit 1-D sinusoidal data. A 4-layer network is the minimum depth for a tree network to have a distinct architecture from standard or parallel networks. We set $m_2 = m_3 = 2$ so that features have up to $K = 2^2 = 4$ switches in Theorem 3.5

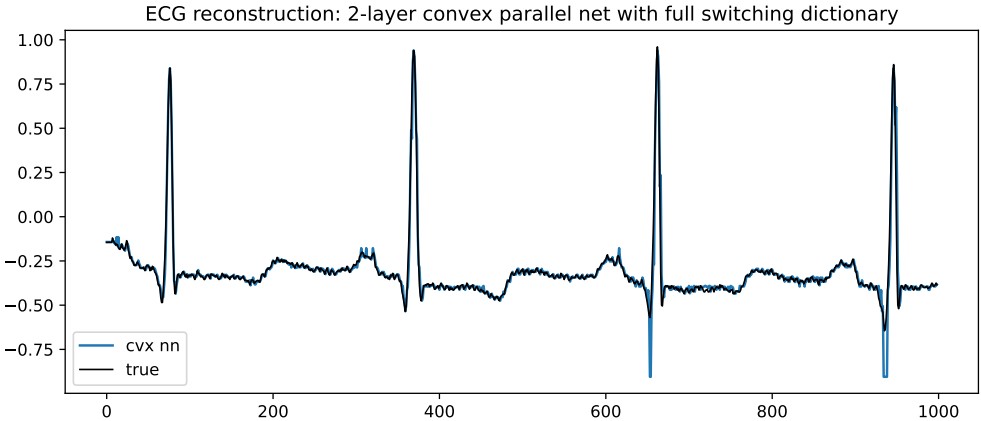

Figure 10: 2-layer neural network trained with full Lasso dictionary.

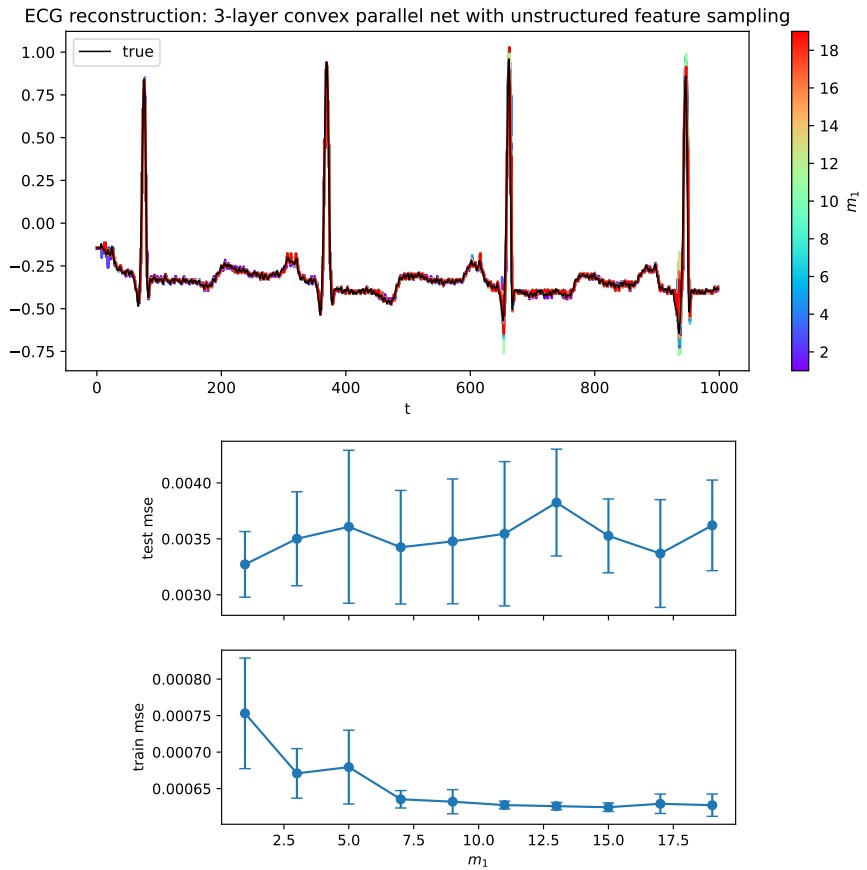

Figure 11: Convex Lasso training with unstructured feature sampling versus width $m_1$.

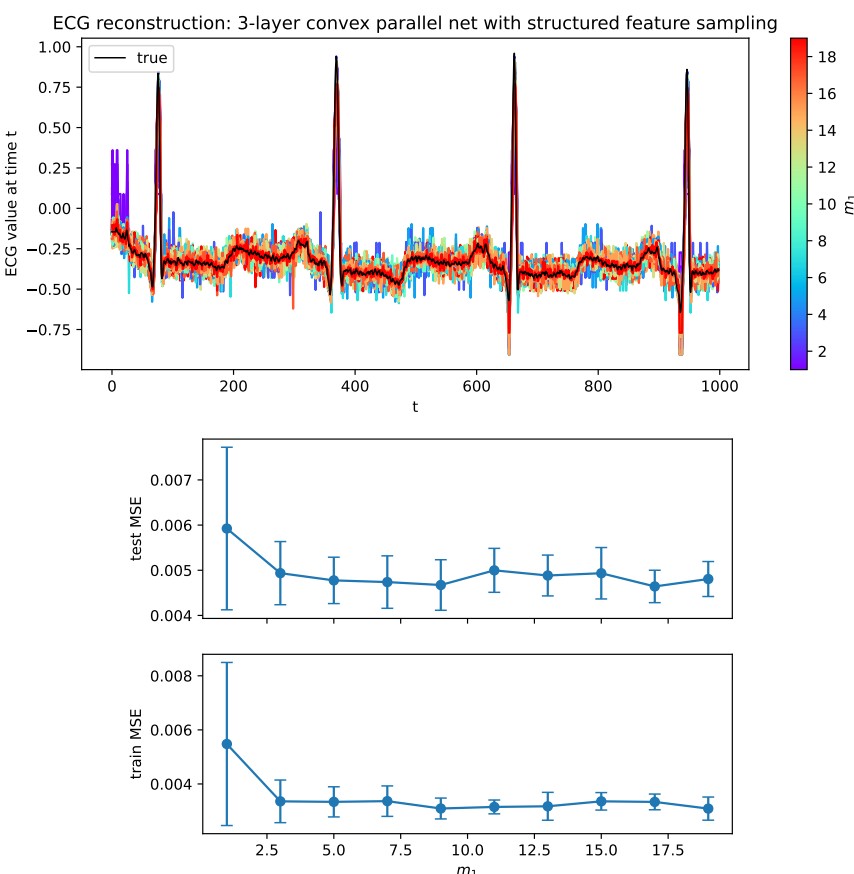

Figure 12: Convex Lasso training with noisy data and structured feature sampling versus width.

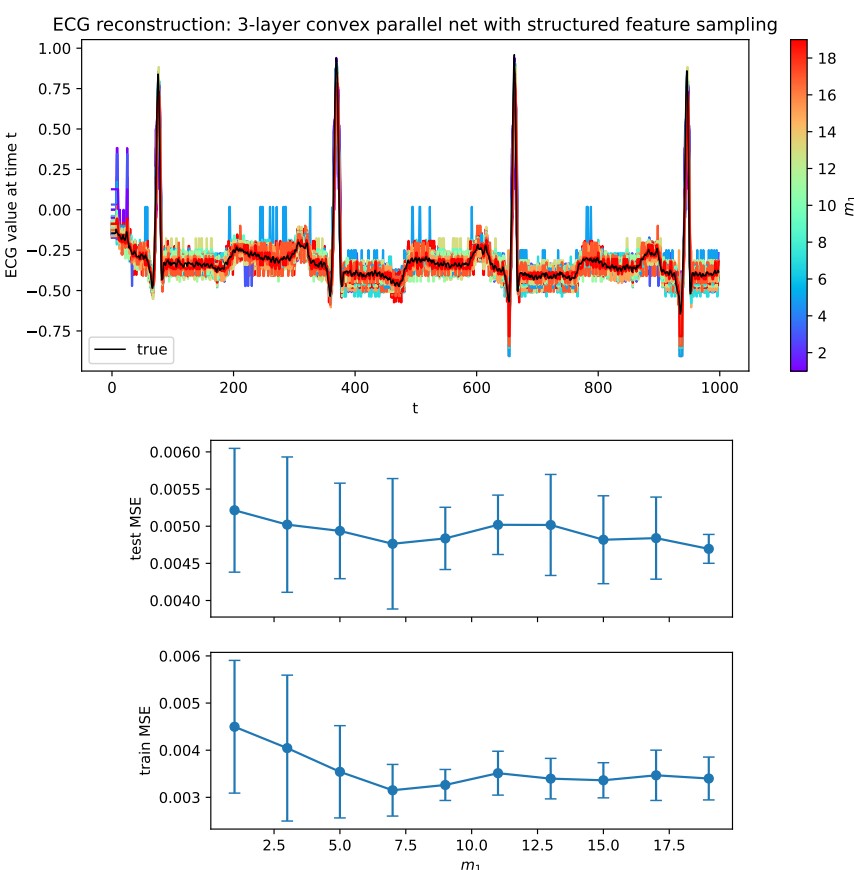

Figure 13: Convex Lasso training with noiseless and structured feature sampling versus width.

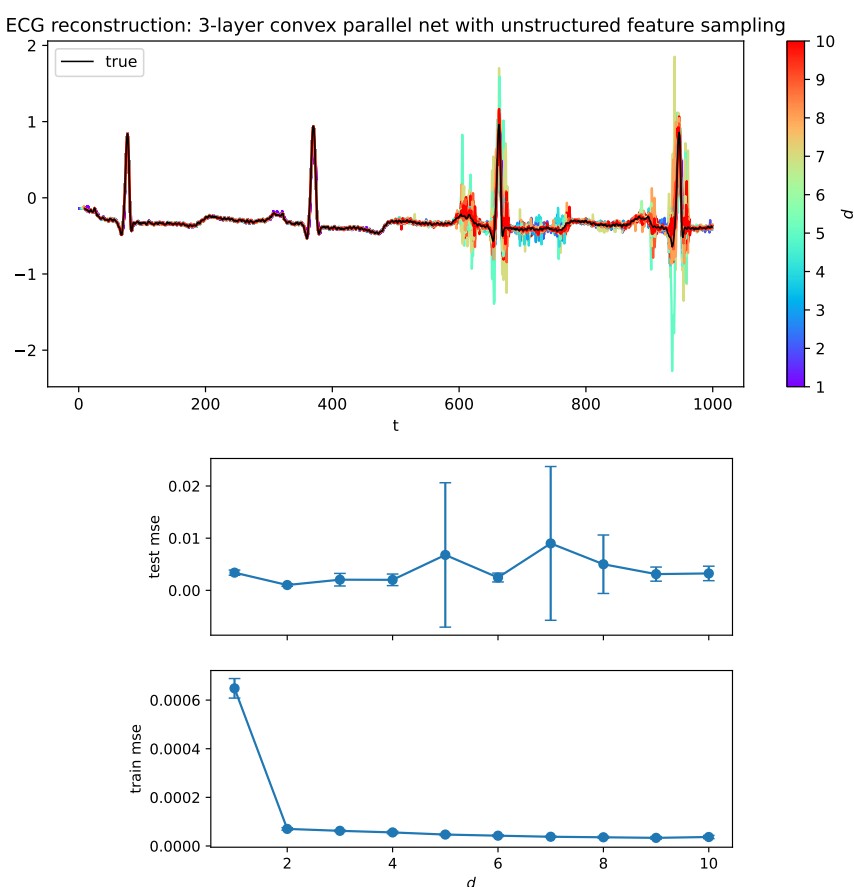

Figure 14: Convex Lasso training with $m_1 = 10$ and unstructured feature sampling versus autoregression lag $d$.

(Remark C.1). The $x$ data consists of $N = 50$ evenly spaced points in $[-2, 2]$. As features can have up to 4 switches over a vector of length 50, the number of possible features is large (Appendix C). To demonstrate a small but computationally tractable example, we keep $m_2, m_3$ and the number of layers small and use a subset of features by random sampling.

Figure 15 shows the impact of sampling on performance. As the number of features sampled increases, the train and test error decreases as seen in the left panel, and the fit improves as seen in the right panel. In the right panel, the bottom plot shows the fit evaluated only at the $N$ training points, with linear interpolation between them, while the top plot shows the fit over all 1000 test points, highlighting regions where the reconstruction deviates from the true signal.

Figure 16 shows the regularization path, or how the error varies with the regularization parameter $\beta$. The training error increases with $\beta$ since the training objective is monotone increasing with $\beta$. The test error in the top left panel follows an expected U-shaped curve. For large $\beta$, the neural network underfits the data, leading to large test error. For sufficiently low $\beta$, the network overfits the data, also leading to large test error. In Figure 16 Gaussian noise with variance $\sigma^2 = 0.1^2$ is added to the sinusoid to study overfitting in the presence of noise.

Figure 17 further investigates the effect of unstructured noise. As the noise variance $\sigma^2$ increases, the network performance typically degrades, seen in the fits in the right panel and error in the left panel. We note the training error initially is non-monotone for small $\sigma$. This may be due to random occurrences of the sampled features fitting the noisy data well, and is an area for future investigation.

While Figure 16, Figure 17 study effects of Gaussian noise, Figure 18 explores the effect of structured noise in the form of an additive perturbation or phase shift given by $\phi \sin(3x)$ where $\phi \in \mathbb{R}$. The bottom-right panel shows the shifted sinusoid for different values of $\phi$ and the bottom-top panel shows the corresponding network fit when trained on that data. The left panel shows that the as the shift amplitude $\phi$ increases, the training and test error increases as expected. However as seen in the MSE values and the fit in the top right panel, for moderate values of $\phi$, the network still approximates the true non-shifted sinusoid. Moreover, the test error in the top left panel increases minimally at a slower rate for small $\phi$, suggesting robustness. Analyzing this apparent convexity in the test error from a theoretical perspective is a direction for future work. Figure 15, Figure 16, and Figure 18 show results for a single sampling of features. In Figure 19, the experiment in Figure 15 is repeated for 50 different random feature subsets. The left panel illustrates the distribution in error and the right panel shows the fit of all networks trained.

In Figure 16, Figure 18, Figure 19, 1000 features were sampled. Evenly spaced samples in $x$ were used for simplicity to focus on investigating the network's ability to learn a given function (sinusoid) and the effects of noise to the $y$ values, feature sampling and regularization parameter in an ideal case without being affected by the randomness in the $x$ distribution. Future work includes investigating the effect of randomly selected $x$ values and an uneven distribution of data points along $x$.

### B.8   Comparison with non-binary MLP

This section compares the convex Lasso for 2-layer sign networks with standard non-convex training of 2-layer ReLU networks on ECG data in the same autoregression setting as Appendix B.6. The ReLU networks use the same number of neurons as the number of non-zero $z_i^*$ in the Lasso solution. Adam with a learning rate of $10^{-3}$, 1000 epochs, and 10 initializations are used for the non-convex training. The non-convex loss over epochs is shown in Figure 22. The network fits of the 10 initializations are individually plotted along with the convex training fit in Figure 21. The fits are similar that visually distinguishing them is challenging. Figure 20 shows a box plot of the training and test loss. While the convex sign network achieves lower training loss, its test loss is higher than that of the standard MLP. Reasons include a different activation function between the two networks, as well as optimality guarantee differences between convex versus non-convex training. While the sign network may have higher test error over ReLU, the 2-layer convex sign training still outperforms the 2-layer non-convex sign training in Section 5.1. While ReLU may enjoy somewhat better test performance over sign activation, their performance in this experiment is comparable, as can be seen visually in the overlapping fits in Figure 21. Moreover, sign activation has benefits such as being compatible with

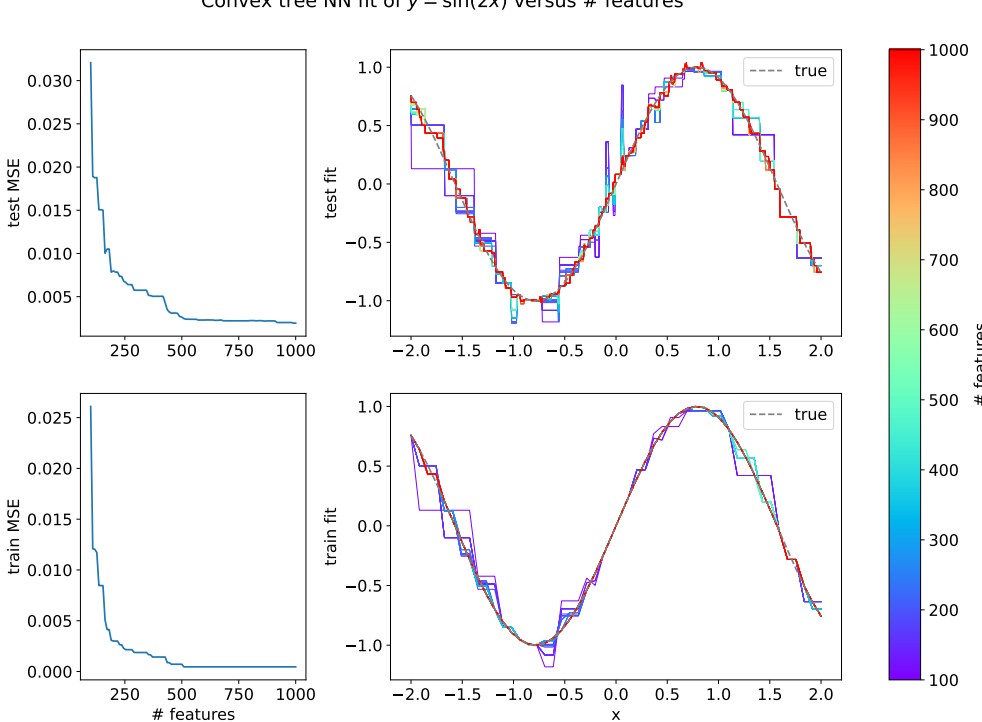

Figure 15: Tree network fit versus number of subsampled features.

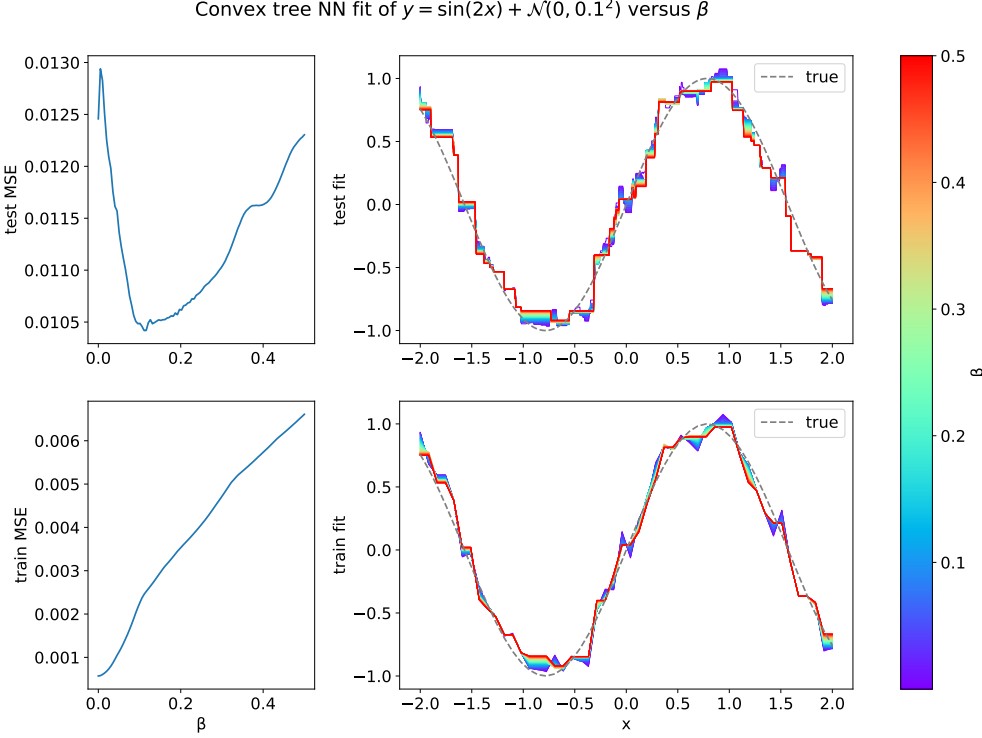

Figure 16: Regularization path of tree network with subsampled Lasso.

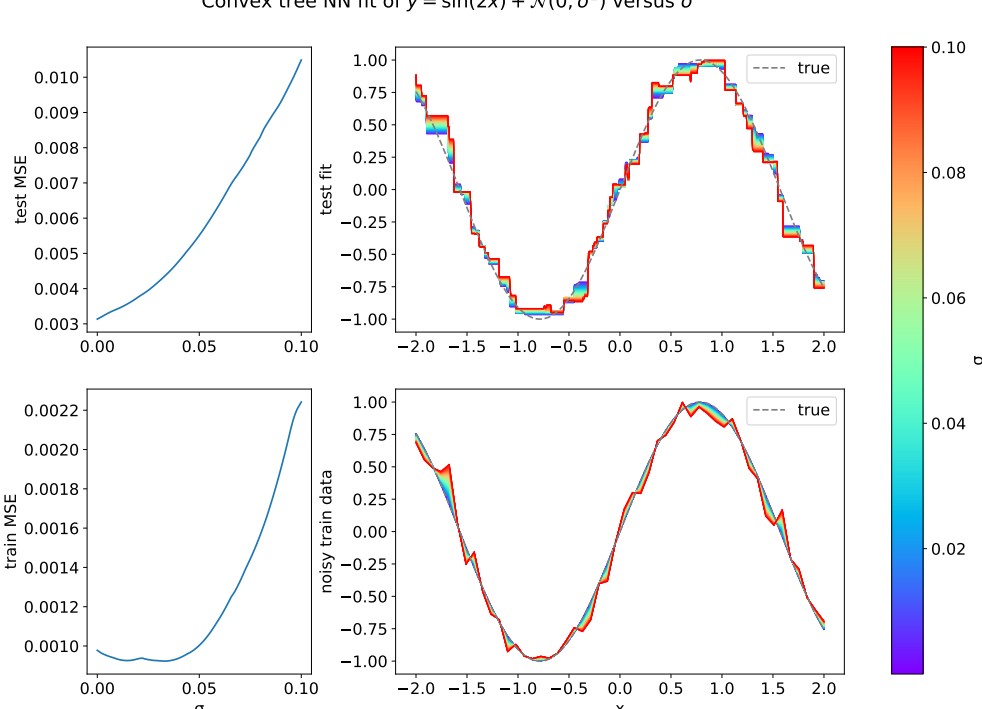

Figure 17: Tree network fit versus Gaussian noise.

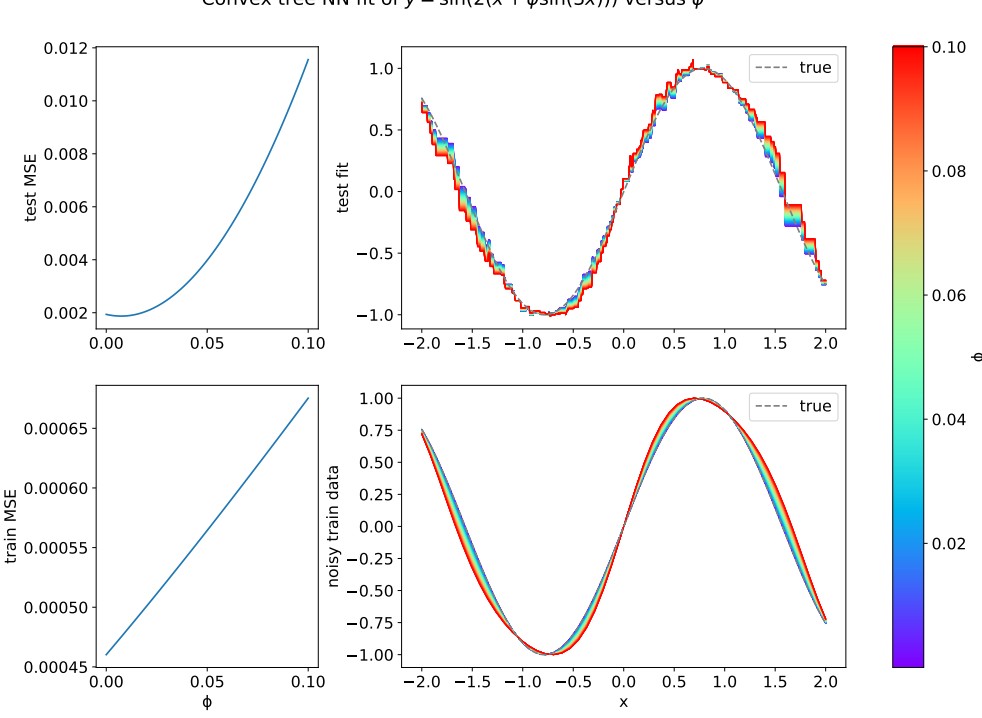

Figure 18: Tree network fit versus phase error in sinusoidal data.

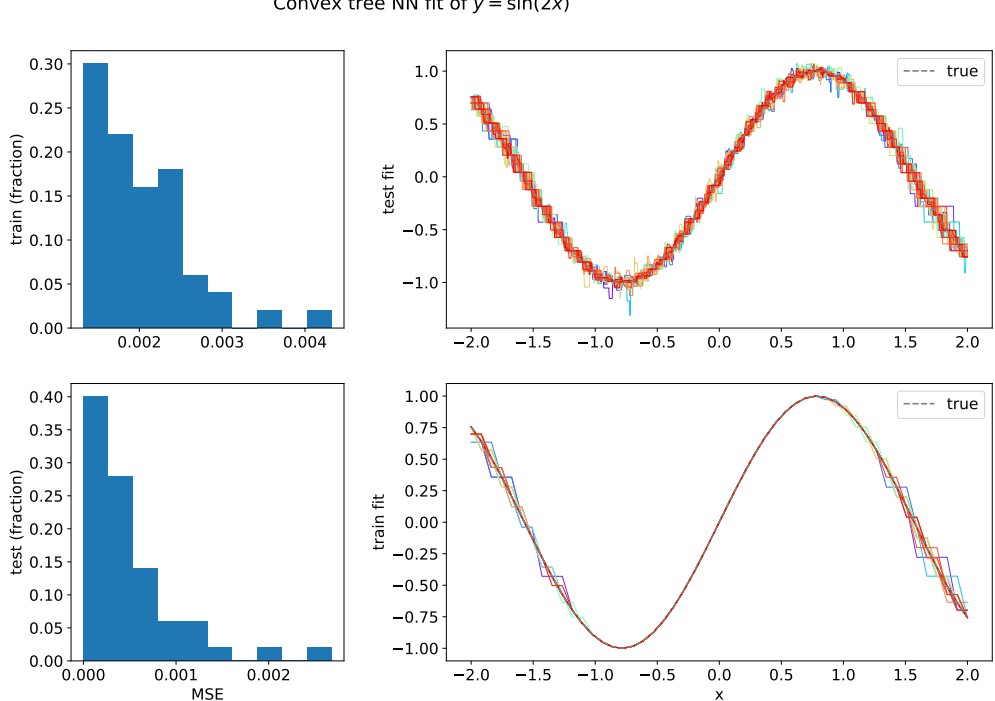

Figure 19: Distribution of tree network performance over different feature subsamplings.

quantization and efficient computation. The best choice of activation therefore depends on the computational and performance needs of the setting.

## C   Discussion

**Uniqueness**: In general, the Lasso problem may have multiple solutions, which can be exactly characterized using the equicorrelation set (47), (14), (31). The generalization capability of different globally optimal solutions to Lasso problems for other architectures is explored in (47), (31) and analysis with respect to networks in this paper is an area of future work. Even when the Lasso solution is unique (such as the closed-form solution in Theorem 4.1), the reconstructed network is only one of the multiple globally optimal solutions to the non-convex training problem. A class of equivalent optimal networks can be generated by splitting, permuting, and merging neurons (47), (48). In certain architectures and under certain assumptions, the solution set and stationary points of the non-convex training problem can be found from performing these operations on the network reconstructed from Lasso problem and its subsampled version, respectively (47), (48). Analysis comparing the properties of the network corresponding to the Lasso solution versus all other solutions to the non-convex problem remains an area for future work.

**Complexity**: The complexity of solving the Lasso problem is $O(N^2 F)$, where $F$ is the number of features (14). The number of features for a 1-D, 2-layer sign network is $|\mathbf{H}^{(1)}| = N$. The number of features for a 1-D, rectangular sign network with at least 3 layers is $|\mathbf{H}^{(m_1)}| = \sum_{k=0}^{m_1} \binom{N-1}{k}$, since there are $N-1$ indices $2, \cdots, N$ at which a vector $\mathbf{h} \in \mathbf{H}^{(m_1)}$ can switch, and since $h_1 = 1$, this determines the rest of the elements of $\mathbf{h}$ through consecutive runs of 1 and $-1$. Similarly for a tree network, this becomes $|\mathbf{H}^M| = \sum_{k=0}^{M} \binom{N-1}{k}$ where $M = \prod_{l=1}^{L-1}$.

The tree network is designed to expand the dictionary, which increases the expressivity of the network. On the other hand, in practice, training the tree network, and even the rectangular network for a large number of neurons and data, can potentially result in a large number of features. Nonetheless, the convexity provides analytical tractability and intuition, and we believe the value of the Lasso formulation, in addition to practical

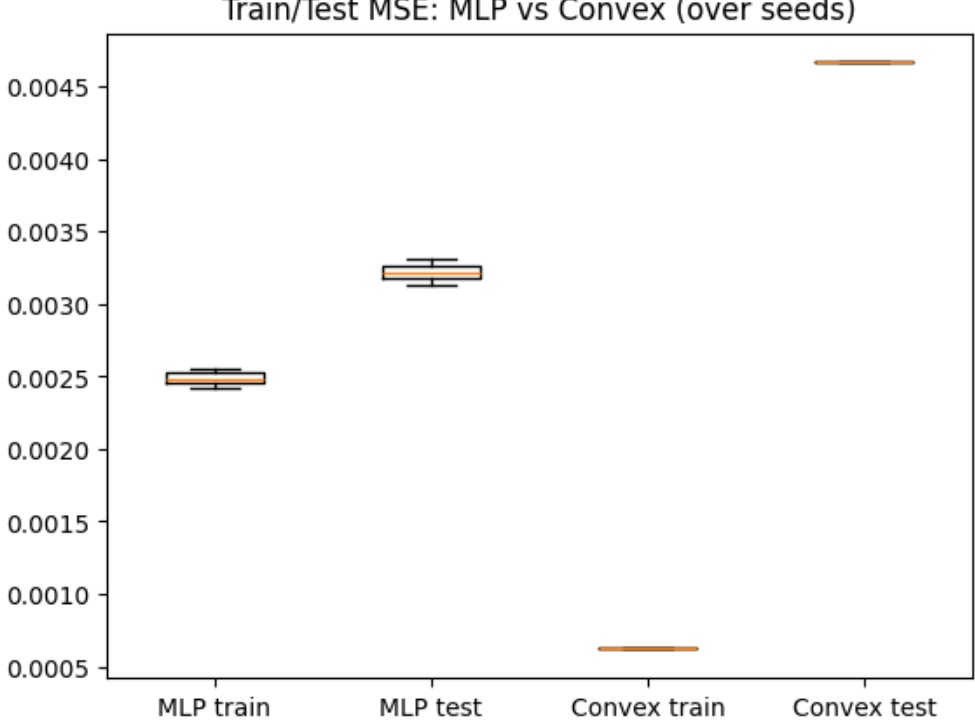

Figure 20: Losses of 2-layer non-convex ReLU training and convex 2-layer sign network training.

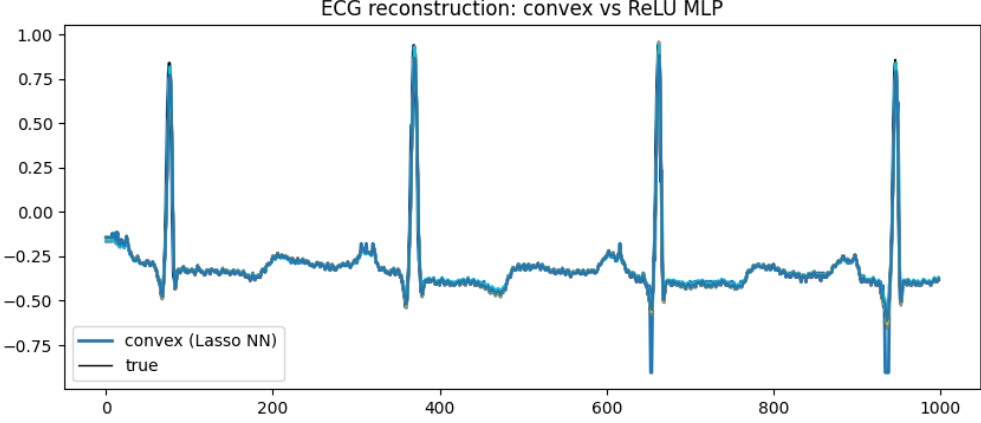

Figure 21: ECG fit of 2-layer non-convex ReLU training and convex 2-layer sign network training. The curves that are neither blue nor black are from the non-convex training, but they largely overlap with the true data and convex training.

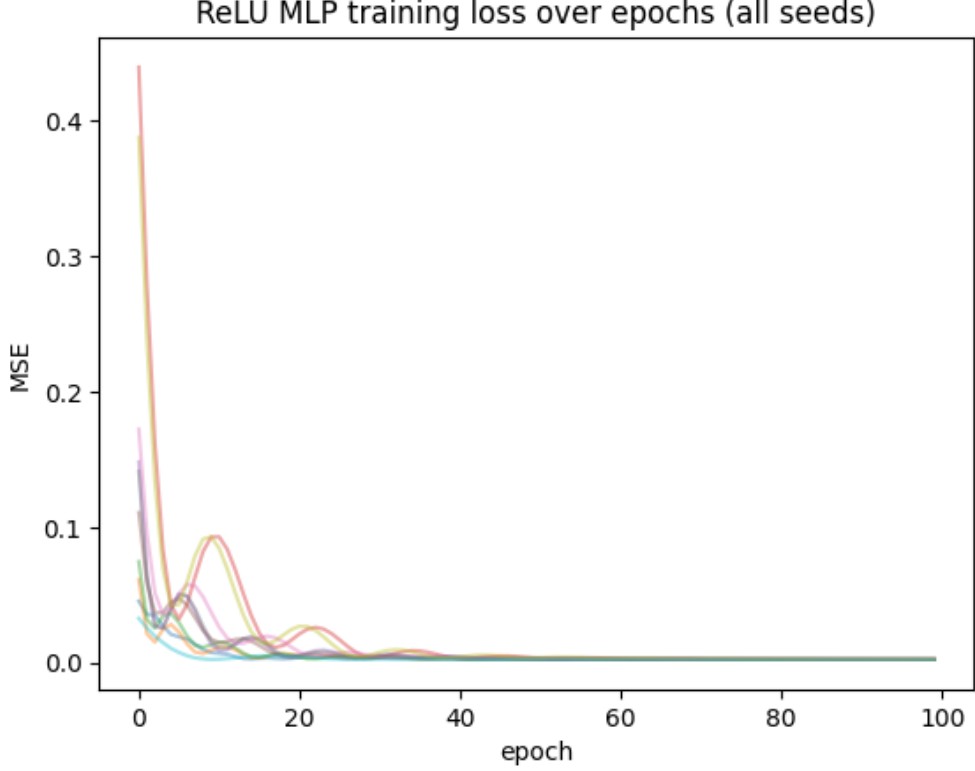

Figure 22: Loss of 2-layer non-convex ReLU training on ECG data.

use, is providing a transparent model, whereas neural networks are typically treated as a black box. In practice, features can be subsampled as well (45; 48; 49; 50).

Another property that mitigates the complexity is that the features can be re-used for different training data. The characterization of the dictionaries in terms of switching sets shows that the Lasso problems depend only on the architecture and the number of neurons rather than the data. In contrast, the Lasso dictionaries for deep neural nets in previous work (12), (31) use dictionaries that are functions of the training data. Remark 3.7 formally states this observation. The invariance of the sign dictionary to the data enables the solution path results in Section 4 to apply to arbitrary $x_n \in \mathbb{R}$. Finally, as discussed in Remark E.6, the symmetric properties of the sign activation allow the potential complexity of the dictionary to be halved, by only including features whose first element is 1.

**Comparison with piecewise linear activations (31)**: For 2-layer networks, the inner parameters in Lemma 3.12 are consistent with the reconstruction in (31). Like ReLU dictionaries (for 'symmetrized' networks) in (31), Theorem 3.4 shows that sign dictionaries freeze after 3 layers, even for rectangular networks, which are a wider version of symmetrized networks in (31). (The deep narrow and symmetrized networks in (31) are special cases of rectangular networks when each unit has width 1 and 2, respectively.) This may suggest the limitations of network representation for low-dimensional data and certain architectures. Basis functions for deep networks with piecewise linear activations in (31) change slope at training data, as well as certain non-training data points (in particular, reflections of training data) for deep narrow networks with absolute value activation. However, sign basis functions only change value at training data, as shown in Corollary 3.10. Whereas ReLU features in (31) are piecewise linear even for the limited deep narrow network, sign features are piecewise constant (Corollary 3.10). The rigidity of sign features may limit its expressibility (51).

**Network parameter simplification**: Corollary 4.2 and Corollary 4.4 give optimal neural network functions. Similar to Corollary 3.10, the optimal weight and bias parameters are not explicit, as the network expressions

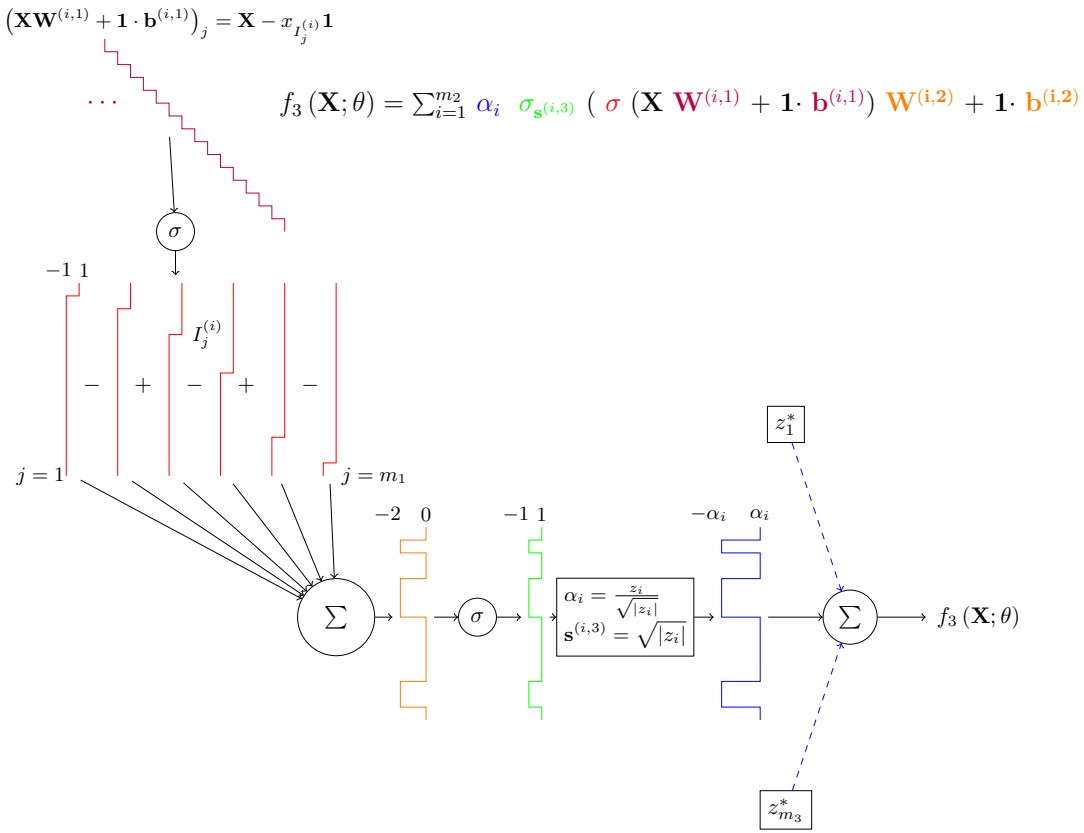

Figure 23: Output of an optimal 3-layer neural net with sign activation reconstructed from a Lasso solution $\mathbf{z}^*$ using Lemma 3.12. The pulse colors correspond to network operations. The alternating $+, -$ represent $\mathbf{W}^{(i,2)} = (1, -1, 1, -1, \cdots)$. The red and green pulses illustrate 2 and 3-layer basis functions, respectively (Lemma 3.2), while the other colors represent multiplication by weights and amplitudes.

have been simplified. However, the optimal parameters can be reconstructed from the Lasso solution following the procedure in Lemma 3.12.

**General loss function**: Our results generalize to a wide class of convex loss functions $\mathcal{L}_{\mathbf{y}} : \mathbb{R}^N \to \mathbb{R}$ with corresponding training problem $\min_{\theta \in \Theta} \mathcal{L}_{\mathbf{y}}(f_L(\mathbf{X}; \theta)) + \frac{\beta}{2}\|\theta_w\|^2$. The proofs of the main results show that an equivalent Lasso problem is $\min_{\mathbf{z}} \mathcal{L}_{\mathbf{y}}(\mathbf{A}\mathbf{z} - \mathbf{y}) + \beta\|\mathbf{z}\|_1$, generalizing the Lasso problem 1.

**Extensions and limitations**: The training complexity poses a potential limitation for wide or tree networks and is addressed in Section 3. Future work will study extensions to variable width neural networks and higher dimensional data.

The 1-D results may be applied to sufficiently structured or low rank data in higher dimensions. Extension to higher dimensional data may be approached through geometric algebra (52). Another future direction is extending the analysis of binary periodic data in Section 4 to broader classes of signals. One approach may involve leveraging wavelet analysis; as square waves can be expressed as sums of Haar wavelets (53), our results may generalize to signals representable in that basis.

## C.1 Tree network definition

Let $L \geq 3, m_2, \cdots, m_L \in \mathbb{N}$. Given $l \in \{0, \cdots, L-2\}$, let $\mathbf{u}$ be an $l$-tuple where if $l = 0$, we denote $\mathbf{u} = \emptyset$ and otherwise, $\mathbf{u} = (u_1, \cdots, u_l)$ such that $u_i \in [m_{L-i}]$ for $i \in [l]$. For an integer $a$, denote $\mathbf{u} \oplus a$ as the concatenation $(u_1, \cdots, u_l, a)$. For $l \in [L-1]$, and $\mathbf{u}$ of length $l$, let $\alpha^{(\mathbf{u})}, s^{(\mathbf{u})}, b^{(\mathbf{u})}, \mathbf{w}^{(\mathbf{u})} \in \mathbb{R}$, except let

$\mathbf{w}^{(u_1,\cdots,u_{L-1})} \in \mathbb{R}^d$. For all $\mathbf{u}$ of length $L-1$, let $\mathbf{X}^{(u_1,\cdots,u_{L-1})} = \mathbf{x} \in \mathbb{R}^{1\times d}$. For $\mathbf{u}$ of length $l \in \{0,\cdots,L-2\}$, let $\mathbf{X}^{(\mathbf{u})} \in \mathbb{R}$ be defined by

$$\mathbf{X}^{(\mathbf{u})} = \sum_{i=1}^{m_{L-l}} \sigma_{s^{(\mathbf{u}\oplus i)}} \left( \mathbf{X}^{(\mathbf{u}\oplus i)}\mathbf{w}^{(\mathbf{u}\oplus i)} + b^{(\mathbf{u}\oplus i)} \right) \alpha^{(\mathbf{u}\oplus i)}. \tag{6}$$

A *tree neural network* is $f_L(\mathbf{x};\theta) = \mathbf{X}^{(\emptyset)}$. Visualizing the neural network as a tree, $\mathbf{X}^{(\emptyset)}$ is the "root," $\mathbf{u} = (u_1,\cdots u_l)$ specifies the path from the root at level $0$ to the $u_l{}^{\text{th}}$ node (or neuron) at level $l$, $\mathbf{X}^{(\mathbf{u})}$ represents a subtree at this node, and (6) specifies how this subtree is built from its child nodes $\mathbf{X}^{(\mathbf{u}\oplus i)}$. The leaves of the tree are all copies of $\mathbf{X}^{(u_1,\cdots,u_{L-1})} = \mathbf{X}$. Let $\mathcal{U} = \prod_{l=0}^{L-2}[m_{L-l}]$. The weight and bias parameter sets are $\theta_w^{(i)} = \left\{ \alpha^{(\mathbf{u})}, s^{(\mathbf{u})}, \mathbf{w}^{(\mathbf{u})} : \mathbf{u} \in \mathcal{U}, u_1 = i \right\}, \theta_b^{(i)} = \left\{ b^{(\mathbf{u})} : \mathbf{u} \in \mathcal{U}, u_1 = i \right\}$. For tree networks, let $\boldsymbol{\alpha} = \left( \alpha^{(1)}, \cdots, \alpha^{(m_L)} \right) \in \mathbb{R}^{m_L}$.

For $L = 2$, we let $\mathbf{u} = $ and hence $(\mathbf{u} \oplus i) = i$ so a tree network is $f_2(\mathbf{x};\theta) = \sum_{i=1}^{m_2} \sigma_{s^{(i)}} \left( \mathbf{x}\mathbf{w}^{(i)} + b^{(i)} \right) \alpha^{(i)}$, which is the same structure as a 2-layer standard network. For $L = 3$, a tree network is $f_3(\mathbf{x};\theta) = \sum_{j=1}^{m_3} \sigma_{s^{(j)}} \left( \mathbf{X}^{(j)}\mathbf{w}^{(j)} + b^{(j)} \right) \alpha^{(i)} = \sum_{j=1}^{m_2} \left( \sigma_{s^{(j)}} \left( \sum_{i=1}^{m_2} \sigma_{s^{(j,i)}} \left( \mathbf{x}\mathbf{w}^{(j,i)} + b^{(j,i)} \right) \alpha^{(j,i)} \right) \mathbf{w}^{(i)} + b^{(i)} \right) \alpha^{(i)}$ which is equivalent to a 3-layer parallel network where $\mathbf{W}^{(i,1)} = \left( \mathbf{w}^{(j,1)}, \cdots, \mathbf{w}^{(j,m_2)} \right), \mathbf{b}^{(i,1)} = \left( \mathbf{b}^{(j,1)}, \cdots, \mathbf{b}^{(j,m_2)} \right), \mathbf{W}^{(i,2)} = \mathbf{w}^{(i)}(\alpha^{(j,1)}, \cdots, \alpha^{(j,m_2)})^T, \mathbf{b}^{(i,2)} = b^{(i)}, \alpha_i = \alpha^{(i)}$ and $m_1$ as defined for parallel networks is set to $m_2$ as defined for tree networks.

Indeed, every tree network has as its base a 2-layer network, where its *innermost parameters* $\mathbf{w}^{(\mathbf{u})}, b^{(\mathbf{u})}, \alpha^{(\mathbf{u})}$ (with $\mathbf{u}$ of maximum length $L-1$) correspond to the 2-layer network parameters (written in parallel architecture form as shown in Appendix D) $\mathbf{W}^{(i,1)}, \mathbf{b}^{(i,1)}, \alpha_i$ respectively, for $i \in [m_1]$.

**Remark C.1.** *In parallel networks, we define $m_{L-1} = 1$. However in tree networks, we define $m_1 = 1$ when we write that tree features have $\mathbf{H}^K$ switches where $K = \prod_{l=1}^{L-1} m_l$. This appears in Theorem 3.5.*

Each $\sigma_{s^{(i)}} \left( \mathbf{X}^{(i)}\mathbf{w}^{(i)} + b^{(i)} \right)$ for $i \in [m_L]$, where $\mathbf{u} = \emptyset$ is considered a parallel unit.

The following remark, which defines the training problem minimization, is adapted from (31). The remark in (31) discusses only parallel networks, but we adapt it for tree networks as well.

**Remark C.2.** *Plugging equation 3 into its own expression for the next layer shows that in a parallel network, for $l \in [L-2]$,*

$$\hat{\mathbf{X}}^{(i,l+2)} = \sigma_{\mathbf{s}^{(i,l+1)}} \left( \sigma \left( \hat{\mathbf{X}}^{(i,l)}\mathbf{W}^{(i,l+1)} + \mathbf{b}^{(i,l)} \right) \mathbf{s}^{(i,l)}\mathbf{W}^{(i,l+1)} + \mathbf{b}^{(i,l+1)} \right).$$

*Similarly, plugging in equation 6 into itself shows that in a tree network, for $0 \le l \le L-3$,*

$$\mathbf{X}^{(\mathbf{u})} = \sum_{i=1}^{m_{L-l}} \alpha^{(\mathbf{u}\oplus i)} \sigma_{s^{(\mathbf{u}\oplus i)}} \left( b^{(\mathbf{u}\oplus i)} + \sum_{j=1}^{m_{L-l-1}} \alpha^{(\mathbf{u}\oplus i\oplus j)} \sigma \left( \mathbf{X}^{(\mathbf{u}\oplus i\oplus j)}\mathbf{w}^{(\mathbf{u}\oplus i\oplus j)} + b^{(\mathbf{u}\oplus i\oplus j)} \right) s^{(\mathbf{u}\oplus i\oplus j)}\mathbf{w}^{(\mathbf{u}\oplus i)} \right).$$

*The* inner parameters *of a parallel network are $\mathbf{s}^{(i,l)}$ for $l \le L-2$ and $\mathbf{W}^{(i,l)}$ for $l \le L-1$. In a tree network, they are $\left( \mathbf{s}^{\mathbf{u}\oplus 1}, \cdots \mathbf{s}^{\mathbf{u}\oplus m_{L-l}} \right), \left( \alpha^{\mathbf{u}\oplus 1}, \cdots \alpha^{\mathbf{u}\oplus m_{L-l}} \right)$ for $\mathbf{u}$ of positive length, and $\left( \mathbf{w}^{\mathbf{u}\oplus 1}, \cdots \mathbf{w}^{\mathbf{u}\oplus m_{L-l}} \right)$ for $\mathbf{u}$ of any length. The network $f_L(\mathbf{X};\theta)$ is invariant to the value of the inner parameters, so they would be driven to $0$ by weight regularization. We define the minimum value in equation 4 as an infimum which is approached as the norms of the inner parameters approach $0$. Therefore the inner parameters are not regularized (the effective depth is $2$), and we optimize for their directions rather than their magnitudes.*

## Proofs

**Remark C.3** (Parallel to standard architecture conversion). *Let $\mathbf{W}^{(1)} = \left[ \mathbf{W}^{(1,1)} \cdots \mathbf{W}^{(m_1,1)} \right]$. For $l \ge 1$, let $\mathbf{b}^{(l)} = \left( \mathbf{b}^{(1,l)} \cdots \mathbf{b}^{(m_l,l)} \right)$. For $l > 1$, let $\mathbf{W}^{(l)} = \text{blockdiag} \left( \mathbf{W}^{(1,l)} \cdots \mathbf{W}^{(m_l,l)} \right)$. And let $\boldsymbol{\alpha}, \xi$ be the same in the standard network as the parallel one.*

## D  2-layer networks

A 2-layer network can either be written 1) as a standard network $f_2(\mathbf{x}; \theta) = \mathbf{X}^{(2)}\boldsymbol{\alpha}$ with $L = 2, m_1 = d, m_2 \in \mathbb{N}$ and $\mathbf{X}^{(2)} = \sigma\left(\mathbf{X}\mathbf{W}^{(1)} + \mathbf{b}^{(1)}\right)$ for $\mathbf{W}^{(1)} \in \mathbb{R}^{m_1 \times m_2}, \mathbf{b}^{(1)} \in \mathbb{R}^{1 \times m_2}$; or 2) as a parallel network $f_2(\mathbf{x}; \theta) = \sum_{i=1}^{m_2} \hat{\mathbf{X}}^{(i,2)}\alpha_i$ with $L = 2, m_0 = d, m_1 = 1, m_2 \in \mathbb{N}$ and $\hat{\mathbf{X}}^{(i,2)} = \sigma\left(\hat{\mathbf{X}}^{(i,1)}\mathbf{W}^{(i,1)} + \mathbf{b}^{(i,1)}\right)$ where $\mathbf{W}^{(i,1)} \in \mathbb{R}^{m_0 \times m_1}, \mathbf{b}^{(i,1)} \in \mathbb{R}^{1 \times m_1}$. Note that in both the standard and parallel form, $m_2$ denotes the number of neurons.

## Parallel and tree networks with data dimension $d \geq 1$

The proofs for parallel networks in this section follow similar approaches as (31) and (12). We assume the activation is sign and the networks are of parallel or tree architecture. The training problem (4) can be written as

$$\min_{\theta \in \Theta} \mathcal{L}_{\mathbf{y}}\left(f_L\left(\mathbf{X}; \theta\right)\right) + \frac{\beta}{2} \sum_{i=1}^{m_L} \sum_{\theta^{(i,l)} \in \theta_w^{(i)}} \|\theta^{(i,l)}\|_2^2$$

where $\theta^{(i,l)}$ are parameters, for example $\mathbf{W}^{(i,l)}$, $s^{(i,l)}$ or $\alpha_i$ for parallel networks.

**Definition D.1.** *An* amplitude-distilled *network has amplitude parameters only in the last layer:*

$$f_L(\mathbf{x}; \theta) = \sum_{i=1}^{m_L} \hat{\mathbf{X}}^{(i,L)}\alpha_i s^{(i,L-1)}$$

$$\hat{\mathbf{X}}^{(i,l+1)} = \sigma\left(\hat{\mathbf{X}}^{(i,l)}\mathbf{W}^{(i,l)} + \mathbf{b}^{(i,l)}\right) \; for \; l \in [L-1]; \quad \hat{\mathbf{X}}^{(i,1)} = \mathbf{x} \tag{7}$$

$$\theta_w^{(i)} = \left\{\alpha_i, s^{(i,L-1)}\right\}, \theta_b^{(i)} = \left\{\mathbf{b}^{(i,l)}, \mathbf{W}^{(i,l)} : l \in [L-1]\right\} \; for \; i \in [m_L]$$

*for a parallel network, and*

$$f_L(\mathbf{x}; \theta) = \sum_{i=1}^{m_L} \sigma\left(\mathbf{X}^{(i)} + b^{(i)}\mathbf{1}\right)\alpha_i s^{(i)}$$

$$\mathbf{X}^{(\mathbf{u})} = \begin{cases} \sum_{i=1}^{m_{L-l}} \alpha^{(\mathbf{u}\oplus i)}\sigma\left(\mathbf{X}^{(\mathbf{u}\oplus i)} + b^{(\mathbf{u}\oplus i)}\right) & if \; 1 \leq l \leq L-3 \\ \sum_{i=1}^{m_{L-l}} \alpha^{(\mathbf{u}\oplus i)}\sigma\left(\mathbf{X}\mathbf{w}^{(\mathbf{u}\oplus i)} + b^{(\mathbf{u}\oplus i)}\right) & if \; l = L-2 \end{cases} \tag{8}$$

$$\theta_w^{(i)} = \left\{\alpha^{(i)}, \mathbf{s}^{(i)}\right\} \theta_b^{(i)} = \left\{\alpha^{(\mathbf{u})}, b^{(\mathbf{u})} : \mathbf{u} \in \mathcal{U}, u_1 = i\right\} \; for \; i \in [m_L]$$

*where $\mathbf{u}$ has positive length for $\alpha^{(\mathbf{u})} \in \theta_b^{(i)}$, for a tree network.*

**Lemma D.2.** *The amplitude-distilled parallel and tree networks are equivalent to the corresponding original neural networks.*

*Proof.* By Remark C.2, it makes sense to limit the regularization penalty to the weights in the final layers $L-1$ and $L$ instead of applying to all layers. For parallel networks, amplitude parameters $\mathbf{s}^{(i,l)}$ can be omitted for layers $l \leq L-2$ through a change of variables $\mathbf{W}^{(i,l)'} = s^{(i,l-1)}\mathbf{W}^{(i,l)}$ for $2 \leq l \leq L-1$. Similarly for tree networks, for $\mathbf{u}$ of length $1 \leq l \leq L-2$ and $i \in [m_{L-l-1}]$, let $\alpha^{(\mathbf{u}\oplus i)'} = \alpha^{(\mathbf{u}\oplus i)} s^{(\mathbf{u}\oplus i)}\mathbf{w}^{(\mathbf{u})}$ (note $\mathbf{w}^{(\mathbf{u})} \in \mathbb{R}$) to drop the parameters $\mathbf{w}^{(\mathbf{u})}$ and $s^{(\mathbf{u}\oplus i)}$. $\square$

Henceforth, networks are assumed to be in amplitude-distilled form. By Lemma D.2, the training problem's regularization term becomes

$$\frac{\beta}{2} \sum_{i=1}^{m_L} \left(|\theta^{(i,L-1)}|^2 + |\theta^{(i,L)}|^2\right) \tag{9}$$

where $\theta^{(i,L)}=\alpha_i$ and $\theta^{(i,L-1)}=s^{(i,L-1)}$ in parallel networks and $\theta^{(i,L)}=\alpha^{(i)}$ and $\theta^{(i,L-1)}=s^{(i)}$ in tree networks. The definitions for the outputs of all networks extend row-wise when the input is $\mathbf{X} \in \mathbb{R}^{N \times d}$ instead of $\mathbf{x} \in \mathbb{R}^{1 \times d}$. For notational uniformity, we denote $\alpha^{(i)}$ for tree networks as $\alpha_i$, the same notation as used in parallel networks.

**Lemma D.3.** *Let $\tilde{\mathbf{X}}^{(i)} \in \mathbb{R}^N$ be $\hat{\mathbf{X}}^{(i,L)}$ for a parallel network or $\sigma\left(\mathbf{X}^{(i)} + b^{(i)}\mathbf{1}\right)$ for a tree network, where the input is $\mathbf{X} \in \mathbb{R}^{N \times d}$. The training problem is equivalent to the* rescaled *problem*

$$\min_{\theta \in \Theta : |\theta^{(i,L-1)}|=1} \mathcal{L}_{\mathbf{y}}\left(\sum_{i=1}^{m_L} \alpha_i \tilde{\mathbf{X}}^{(i)}\right) + \beta \sum_{i=1}^{m_L} |\alpha_i| \tag{10}$$

*Proof.* Applying the AM-GM inequality in equation 9 shows that the optimal value of

$$\min_{\theta \in \Theta} \mathcal{L}_y\left(f_L\left(\mathbf{X}; \theta\right)\right) + \beta \sum_{i=1}^{m_L} |\theta^{(i,L-1)}||\theta^{(i,L)}| \tag{11}$$

is a lower bound on the optimal value of the training problem. Now, the problem

$$\min_{\theta \in \Theta : |\theta^{(i,L-1)}|=1} \mathcal{L}_y\left(f_L\left(\mathbf{X}; \theta\right)\right) + \beta \sum_{i=1}^{m_L} |\theta^{(i,L-1)}||\theta^{(i,L)}| \tag{12}$$

is an upper bound on the value in (11). Given optimal $\left\{\theta^{(i,l)}\right\}$ in equation 11, rescaling parameters as $\theta^{(i,L-1)'}=\theta^{(i,L-1)}/|\theta^{(i,L-1)}|$ and $\theta^{(i,L)'}=|\theta^{(i,L-1)}|\theta^{(i,L)}$, and similarly rescaling bias parameters makes feasible parameters that achieve the same objective in problem (12). Therefore problems (12) and (11) are equivalent. Given optimal $\left\{\theta^{(i,l)}\right\}$ in equation 12, rescaling parameters as $\theta^{(i,L-1)'} = \text{sign}(\theta^{(i,L-1)})\sqrt{|\theta^{(i,L)}|}$, $\theta^{(i,L)'} = \text{sign}(\theta^{(i,L)})\sqrt{|\theta^{(i,L)}|}$ (and rescaling bias parameters) achieves the same objective in the training problem, which is thus equivalent to problem (12). Applying the constraint in problem (12) and absorbing the sign of $\theta^{(i,L-1)}$ into $\theta^{(i,L)} = \alpha_i$ gives problem (10). $\qquad\square$

For simplicity, denote $\tilde{\mathbf{X}} = \tilde{\mathbf{X}}^{(1)}$, which is $\hat{\mathbf{X}}^{(1,L)}$ for a parallel network and $\sigma(\mathbf{X}^{(1)} + b^{(1)}\mathbf{1})$ for a tree network. The convex conjugate of $f$ is $f^*(\mathbf{x}) = \max_{\mathbf{x}}\left\{\mathbf{z}^T\mathbf{x} - f(\mathbf{x})\right\}$ (54).

**Lemma D.4.** *A lower bound on the rescaled training problem is the dual problem*

$$\max_{\lambda \in \mathbb{R}^N} -\mathcal{L}_{\mathbf{y}}^*(\lambda) \quad s.t. \quad \max_{\theta \in \Theta} \left|\lambda^T \tilde{\mathbf{X}}\right| \leq \beta, \tag{13}$$

*Proof.* Rewrite the rescaled problem (10) as

$$\min_{\theta \in \Theta} \mathcal{L}_{\mathbf{y}}(\mathbf{z}) + \beta||\boldsymbol{\alpha}||_1, \quad s.t. \quad \mathbf{z} = \sum_{i=1}^{m_L} \alpha_i \tilde{\mathbf{X}}^{(i)}. \tag{14}$$

The Lagrangian of problem (14) with dual variable $\lambda \in \mathbb{R}^N$ is $L\left(\lambda, \theta\right) = \mathcal{L}_{\mathbf{y}}(\mathbf{z}) + \beta||\boldsymbol{\alpha}||_1 - \lambda^T\mathbf{z} + \sum_{i=1}^{m_L} \lambda^T \tilde{\mathbf{X}}^{(i)} \alpha_i$.

Minimizing the Lagrangian over $\boldsymbol{\alpha}$ and then $\mathbf{z}$ using Fenchel duality (54) gives the dual of problem (14) as

$$\max_{\lambda \in \mathbb{R}^N} -\mathcal{L}_{\mathbf{y}}^*(\lambda) \quad s.t. \quad \max_{\theta \in \Theta} \left|\lambda^T \tilde{\mathbf{X}}^{(i)}\right| \leq \beta \text{ for each } i \in [m_L]. \tag{15}$$

Problem (15) has $m_L$ constraints, but we can see that they are all equivalent since each network unit $\tilde{\mathbf{X}}^{(i)}$ is of the same architecture form. Therefore problem (15) can be simplified to (13). $\qquad\square$

# E   Main Results

## E.1   Parallel networks

In this section, assume the architecture is a parallel network. Denote $\hat{\mathbf{X}}^{(l)} = \hat{\mathbf{X}}^{(1,l)}$; then the dual problem 13 is

$$\max_{\lambda \in \mathbb{R}^N} - \mathcal{L}_{\mathbf{y}}^*(\lambda) \quad \text{s.t.} \quad \max_{\theta \in \Theta} \left| \lambda^T \hat{\mathbf{X}}^{(L)} \right| \le \beta. \tag{16}$$

For a set of vectors $S$, let $[S]$ be a matrix whose set of columns is $S$.

**Definition E.1.** *Define the* hyperplane arrangement set *for a matrix* $\mathbf{Z} \in \mathbb{R}^{N \times m}$ *as*

$$\mathcal{H}(\mathbf{Z}) := \{\sigma(\mathbf{Z}\mathbf{w} + b\mathbf{1}) : \mathbf{w} \in \mathbb{R}^m, b \in \mathbb{R}\} \subset \{-1, 1\}^N. \tag{17}$$

*Let $S_0$ be the set of columns of $\mathbf{X}$. Let $\{S_l\}_{l=1}^{L-1}$ be a tuple of sets satisfying $S_l \subset \mathcal{H}([S_{l-1}])$ and $|S_l| = m_l$. Let $A_{Lpar}(\mathbf{X})$ be the union of all possible sets $S_{L-1}$.*

$S_{L-1}$ contains one vector since $m_{L-1} = 1$ in a parallel network. For example, if $L = 3$ and the data is 1-D, then $S_0 = \{\mathbf{X}\}$ and $\mathcal{H}(\mathbf{X}) = \{\sigma(\mathbf{X}w + b\mathbf{1}) : w, b \in \mathbb{R}\}$, so $S_1 \subset \mathcal{H}(\mathbf{X})$ is a set of $|S_1| = m_1$ vectors of the form $\sigma(\mathbf{X}w + b\mathbf{1})$, and so $[S_1]$ is a $N \times m_1$ matrix of the form $\sigma(\mathbf{X}\mathbf{W} + \mathbf{1}\mathbf{b}^T)$ and $S_2 \subset \mathcal{H}([S_1])$ contains one $N \times 1$ vector of the form $\sigma(\sigma(\mathbf{X}\mathbf{W} + \mathbf{1}\mathbf{b}^T)\mathbf{w} + b\mathbf{1})$. The hyperplane arrangement set $\mathcal{H}(\mathbf{Z})$ enumerates all possible binary, linear classifications of $\{\mathbf{z}_i\}_{i=1}^N$ and contains at most $2\sum_{k=0}^{r-1}\binom{N-1}{k} \le 2r\left(\frac{e(n-1)}{r}\right)^r \le 2^N$ arrangements where $r := \text{rank}(\mathbf{Z}) \le \min\{N, m\}$ (55; 56).

**Lemma E.2.** *The lower bound problem 16 is equivalent to*

$$\max_{\lambda} - \mathcal{L}_{\mathbf{y}}^*(\lambda), \quad \text{s.t.} \max_{\mathbf{h} \in A_{Lpar}(\mathbf{X})} |\lambda^T \mathbf{h}| \le \beta. \tag{18}$$

*Proof.* By the compositional structure of a parallel network (7), for every layer $l \in [L]$, its output $\hat{\mathbf{X}}^{(l)} = [S_{l-1}]$ for some hyperplane arrangement set $S_{l-1} \subset \mathcal{H}\left(\hat{\mathbf{X}}^{(l-1)}\right)$. Recursing over $l \in [L]$ gives $\left\{\hat{\mathbf{X}}^{(L)} : \theta \in \Theta\right\} = A_{Lpar}(\mathbf{X})$. So, the constraints of problems (16) and (18) are the same. $\square$

**Lemma E.3.** *If $\mathbf{h} \in \mathcal{H}(\mathbf{Z})$, then $-\mathbf{h} \in \mathcal{H}(\mathbf{Z})$.*

*Proof.* Let $\mathbf{h} \in \mathcal{H}(\mathbf{Z})$. Then $\mathbf{h} = \sigma(\mathbf{q})$ where $\mathbf{q} = \mathbf{Z}\mathbf{w} + b\mathbf{1}$ for some $\mathbf{w} \in \mathbb{R}^{m_l}, b \in \mathbb{R}$. Note $q_n = b + \sum_{i=1}^{m_l} z_{n,i}w_i$. Let $\tilde{b} = \max_{n \in [N]}\{\sum_{i=1}^{m_l} z_{n,i}w_i : \sum_{i=1}^{m_l} z_{n,i}w_i < -b\}$. Let $\tilde{\mathbf{w}} = -\mathbf{w}$. Let $\tilde{\mathbf{q}} = \mathbf{Z}\tilde{\mathbf{w}} + \tilde{b}\mathbf{1}$. Then $\tilde{q}_n = \tilde{b} - \sum_{i=1}^{m_l} z_{n,i}w_i$. If $q_n \le 0$, we have $\sum_{i=1}^{m_l} z_{n,i}w_i < -b$ and so $\tilde{q}_n \ge 0$ by definition of $\tilde{b}$. Now, since $\tilde{b} < -b$, we have $\tilde{q}_n < -b - \sum_{i=1}^{m_l} z_{n,i}w_i = -q_n$. So if $q_n \ge 0$, then $\tilde{q}_n < 0$. Therefore $-\mathbf{h} = -\sigma(\mathbf{q}) = \sigma(\tilde{\mathbf{q}}) \in \mathcal{H}(\mathbf{Z})$. $\square$

In the proof above, we denote the $(n, i)^{\text{th}}$ element of $\mathbf{Z}$ as $z_{n,i}$ and the $n^{\text{th}}$ element of vector $\mathbf{q}$ as $q_n$ to emphasize that they are scalars. Note that we cannot simply set $\tilde{b} = -b$ so that $\tilde{q} = -\tilde{q}$, because when $q_n = 0$, $\sigma(q_n) = 1$, not 0.

**Lemma E.4.** *There exist $S_l'$ for $l \in [L-1]$ all of whose vectors start with 1 such that $\mathcal{H}([S_l]) = \mathcal{H}([S_l'])$ and $S_l' \subset \mathcal{H}([S_{l-1}'])$ and $|S_l'| = m_l$.*

*Proof.* Let $l \in [L-1]$. Let $\mathbf{Z} = [S_l]$. For every $\mathbf{z}_i$, let $\mathbf{z}_i^+ = -\mathbf{z}_i$ if $\mathbf{z}_i = -1$ and $\mathbf{z}_i^+ = \mathbf{z}_i$ otherwise. By Lemma E.3, since $\mathbf{z}_i^+ = \pm\mathbf{z}_i$, there exists $S_l' = \{\mathbf{z}_i^+ : i \in [m_l]\}$. Let $\mathbf{h} \in \mathcal{H}([S_l])$. For some $\mathbf{w} \in \mathbb{R}^{m_l}, b \in \mathbb{R}$, we have $\mathbf{h} = \sigma(\mathbf{Z}\mathbf{w} + b\mathbf{1})$. Let $w_i^+ = w_i$ if $\mathbf{z}_i = \mathbf{z}_i^+$ and $w_i^+ = -w_i$ otherwise. Then $\mathbf{Z}\mathbf{w} = \sum_{i=1}^{m_{l+1}} w_i \mathbf{z}_i = \sum_{i=1}^{m_{l+1}} w_i^+ \mathbf{z}_i^+ = \mathbf{Z}^+ \mathbf{w}^+$. So $\mathcal{H}([S_l]) \subset \mathcal{H}([S_l'])$. Conversely, an analogous argument shows that $\mathcal{H}([S_l']) \subset \mathcal{H}([S_l])$. So $\mathcal{H}([S_l]) = \mathcal{H}([S_l'])$.

Let $l \in \{0, \cdots, L-2\}$. It suffices to show $S_{l+1}' \subset \mathcal{H}([S_l'])$. Let $\mathbf{z}' \in S_{l+1}'$. Then $\mathbf{z}' = \pm\mathbf{z}$ for some $\mathbf{z} \in S_{l+1} \subset \mathcal{H}([S_l]) = \mathcal{H}([S_l'])$. such that $z_1 = 1$. So $\mathbf{z}$ or $-\mathbf{z}$ is in $\mathcal{H}([S_l'])$. By Lemma E.3, $\mathbf{z} \in \mathcal{H}([S_l'])$. So $S_{l+1}' \subset \mathcal{H}([S_l'])$. $\square$

By Lemma E.3 with $\mathbf{Z} = [S_{l-1}]$, for every set $S_l$, there is a set $S'_l$ consisting of the all the vectors in $S_l$ scaled by $-1$. In particular, for all $S_{L-1} = \{\mathbf{h}\}$, there is another set $S'_{L-1} = \{\mathbf{h}'\}$ and so for every $\mathbf{h} \in A_{Lpar}(\mathbf{X})$, there is a $-\mathbf{h} \in A_{Lpar}(\mathbf{X})$. Therefore we can partition $A = A^+_{L_{par}} \cup A^-_{L_{par}}$ where $A^+_{L_{par}} = \{\mathbf{h} \in A_{Lpar} : h_1 = 1\}$ and $A^-_{L_{par}} = \{\mathbf{h} \in A_{Lpar} : h_1 = -1\} = \{-\mathbf{h} : \mathbf{h} \in A^+_{L_{par}}\}$. This definition is used in the next result.

**Lemma E.5.** *Let $\tilde{\mathcal{H}}$ be $\mathcal{H}$ with the additional requirement that all vectors in $\mathcal{H}(\mathbf{Z})$ start with 1, and $\tilde{A}_{L_{par}}$ be the corresponding $A_{Lpar}$ as defined in Definition E.1. Then $\tilde{A}_{L_{par}} = A^+_{L_{par}}$.*

*Proof.* By definition of $\tilde{\mathcal{H}}$, $\tilde{A}_{L_{par}} \subset A^+_{L_{par}}$. Let $\mathbf{h} \in A^+_{L_{par}}$. Then $h_1 = 1$ and there is a sequence of sets $S_1, \cdots, S_{L-1}$ such that $S_l \subset \mathcal{H}([S_{l-1}])$, $|S_l| = m_l$ and $S_{L-1} = \{\mathbf{h}\}$. By Lemma E.4, for each $l \in [L-1]$, there is some $S'_l$ all of whose vectors start with 1 such that $\mathcal{H}([S'_l]) = \mathcal{H}([S_l])$ and $S'_l \subset \mathcal{H}([S'_{l-1}])$. So if $\tilde{\mathcal{H}}$ is used instead of $\mathcal{H}$ then all vectors in $A^+_{L_{par}}$ are still recovered. □

**Remark E.6.** *Henceforth, add the additional requirement to $\mathcal{H}(\mathbf{Z})$ that its vectors start with 1. Since $|\lambda^T \mathbf{h}| = |\lambda^T(-\mathbf{h})|$, Lemma E.2 remains the same if $A_{Lpar}(\mathbf{X})$ is restricted to its vectors starting at 1. Therefore by Lemma E.5, the result Lemma E.2 still holds.*

**Lemma E.7.** *Let $\mathbf{A} = [A_{Lpar}(\mathbf{X})]$. The lower bound problem 18 is equivalent to*

$$\min_{\mathbf{z}} \quad \mathcal{L}_{\mathbf{y}}(\mathbf{A}\mathbf{z}) + \beta ||\mathbf{z}||_1. \tag{19}$$

*Proof.* Problem (18) is the dual of (19), and since the problems are convex with feasible regions that have nonempty interior, by Slater's condition, strong duality holds (54). □

The set $A_{L,par}$ consists of all possible sign patterns at the final layer of a parallel neural net, up to multiplying by $-1$.

**Lemma E.8.** *Let $\mathbf{A}$ be defined as in Lemma E.7. Let $\mathbf{z}$ be a solution to problem (19). Suppose $m_L \geq ||\mathbf{z}||_0$. There is a parallel neural network satisfying $f_L(\mathbf{X}; \theta) = \mathbf{A}\mathbf{z}$ which achieves the same objective in the rescaled training problem as $\mathbf{z}$ does in (19).*

*Proof.* By construction of $A_{Lpar}$ and the proof of Lemma E.2, for every $\mathbf{A}_i \in A_{Lpar}(\mathbf{X})$, there are tuples $\{\mathbf{W}^{(i,l)}\}^{L-1}_{l=1}, \{\mathbf{b}^{(i,l)}\}^{L-1}_{l=1}$ that, when plugged into the weights and biases for a parallel network, give $\hat{\mathbf{X}}^{(i,L)} = \mathbf{A}_i$. Let $\mathcal{I} = \{i : z_i \neq 0\}$. For $i \in \mathcal{I}$, set $\alpha_i = z_i$. This gives a neural net $f_L(\mathbf{X}; \theta) = \sum_{i \in \mathcal{I}} \alpha_i \hat{\mathbf{X}}^{(i,L)} = \mathbf{A}\mathbf{z}$ with $|\mathcal{I}| \leq m_L$. □

**Proposition E.9.** *For $L$-layer parallel networks with sign activation, the Lasso problem 19 and the original training problem are equivalent.*

*Proof.* By Lemma E.7, the Lasso problem is a lower bound for the training problem. By the reconstruction in Lemma E.8 (see Remark E.21), the lower bound is met with equality. □

## E.2 Tree networks

Now we assume the architecture is a tree network. We use the hyperplane arrangement set $\mathcal{H}$ (17) and define a hyperplane arrangement set $A_{Ltree}$ analogous to $A_{Lpar}$ that enumerates outputs of the tree network. For a matrix $\mathbf{Z}$ and set of column indices $S$, let $\mathbf{Z}_S$ be a matrix consisting of columns of $\mathbf{Z}$ indexed by $S$.

**Definition E.10.** *Define a matrix-to-matrix operator*

$$J^{(m)}(\mathbf{Z}) := \left[ \bigcup_{|S|=m} \mathcal{H}(\mathbf{Z}_S) \right]. \tag{20}$$

*For $L = 2$, let $A_{Ltree}(\mathbf{X}) = \mathcal{H}(\mathbf{X})$ and for $L \geq 2$, let $A_{Ltree}(\mathbf{X})$ be the set of columns in $J^{(m_{L-1})} \circ \cdots \circ J^{(m_2)}([\mathcal{H}(\mathbf{X})])$.*

The columns of $J^{(m)}(\mathbf{Z})$ are all hyperplane arrangement patterns of $m$ columns of $\mathbf{Z}$.

**Lemma E.11.** *For $L \geq 3$, the lower bound problem 13 for tree networks is equivalent to*

$$\max_{z \in \mathbb{R}^N} - \mathcal{L}_{\mathbf{y}}^*(\lambda), \quad \text{s.t.} \quad \max_{\mathbf{h} \in A_{L\,tree}(\mathbf{X})} |\lambda^T \mathbf{h}| \leq \beta. \tag{21}$$

*Proof.* By an argument similar to Remark E.6, it suffices to assume that all vectors in $\mathcal{H}$ start with 1. Let $\mathbf{u}$ be a tuple such that $u_1 = 1$. First suppose $\mathbf{u}$ has length $L - 2$. For all nodes $i$, note that $\left\{ \sigma \left( \mathbf{X}\mathbf{w}^{(\mathbf{u}+i)} + b^{(\mathbf{u}+i)}\mathbf{1} \right) : \mathbf{w}^{(\mathbf{u}+i)} \in \mathbb{R}^d, b^{(\mathbf{u}+i)} \in \mathbb{R} \right\} = \mathcal{H}(\mathbf{X})$ independently of any other sibling nodes $j \neq i$. So every $\mathbf{X}^{(\mathbf{u})}$ is the linear combination of $m_2$ columns in $\mathcal{H}(\mathbf{X})$, with the choice of columns independent of other $\mathbf{u}$ of the same length. Next, for all $\mathbf{u}$ of length $L-3$, the set of all possible $\sigma \left( \mathbf{X}^{(\mathbf{u}+i)} + b^{(\mathbf{u}+i)}\mathbf{1} \right)$ is $J^{(m_2)}(\mathcal{H}(\mathbf{X}))$. Repeating this for decreasing lengths of $\mathbf{u}$ until $\mathbf{u}$ has length 1 gives $\tilde{\mathbf{X}} = \sigma \left( \mathbf{X}^{(i)} + b^{(i)}\mathbf{1} \right) = A_{L\text{tree}}(\mathbf{X})$. $\square$

**Lemma E.12.** *Lemma E.8 analogously holds for the tree network: there is a neural net satisfying $f_L(\mathbf{X}; \theta) = \mathbf{A}\mathbf{z}$.*

*Proof.* The proof is analogous to that of Lemma E.8. By construction of $A_{Ltree}$ and the proof of Lemma E.11, for every $\mathbf{A}_i \in A_{Ltree}(\mathbf{X})$, there are tuples $\{\mathbf{w}^{(\mathbf{u})}, b^{(\mathbf{u})}\} : u_i \in [m_{L-1}]\}$ of each $\mathbf{u}$ of length $l \in \{0, \cdots, L-2\}$ that, when plugged into the weights and biases for a tree network, give $\sigma_{s^{(i)}} \left( \mathbf{X}^{(i)}\mathbf{w}^{(i)} + b^{(i)} \right) = \mathbf{A}_i$. Let $\mathcal{I} = \{i : z_i \neq 0\}$. For $i \in \mathcal{I}$, set $\alpha_i = z_i$. This gives a neural net $f_L(\mathbf{X}; \theta) = \sum_{i \in \mathcal{I}} \alpha_i \sigma_{s^{(i)}} \left( \mathbf{X}^{(i)}\mathbf{w}^{(i)} + b^{(i)} \right) = \mathbf{A}\mathbf{z}$ with $|\mathcal{I}| \leq m_L$. $\square$

*Proof of Corollary 3.10.* The proof of Lemma E.8, Lemma E.12 gives an optimal network $\sum_i z_i^* f_i(x)$ with $f_i(x_n) = A_{n,i}$. Now we show the piecewise constancy of $f_i$. As discussed in Lemma 3.12 and Remark 3.13, the innermost linear term (before unscaling) of the reconstructed $i^{\text{th}}$ unit in a 1-D, $L$-layer parallel network is a vector with $k^{th}$ element of the form $\mathbf{W}_k^{(i,1)}x + \mathbf{b}_k^{(i,1)} = x - x_{g(i,k)-1}$ and so for $x \in [x_n, x_{n-1})$ this value either remains nonnegative or nonpositive for all $k$, and hence passing it through an activation does not change the first neuron's output. This innermost term is the inner linear term of a 2-layer network.

As shown in Appendix C.1, there is a one-to-one correspondence between the innermost parameters of a tree network and a 2-layer network, so the piecewise continuity property also applies to tree networks. Formally we argue as follows. By construction of $A_{Ltree}$ and the proof of Lemma E.11, for every $\mathbf{A}_i \in A_{Ltree}(\mathbf{X})$, there are tuples $\{\mathbf{w}^{(\mathbf{u})}, b^{(\mathbf{u})}\} : u_i \in [m_{L-i}]\}$ of each $\mathbf{u}$ of length $l \in \{0, \cdots, L-2\}$ that, when plugged into the weights and biases for a tree network, give the parallel unit $\sigma_{s^{(i)}} \left( \mathbf{X}^{(i)}\mathbf{w}^{(i)} + b^{(i)} \right) = \mathbf{A}_i$, which is a function of the innermost layer vector $[\sigma_{s^{(\mathbf{u}\oplus 1)}} \left( \mathbf{X}\mathbf{w}^{(\mathbf{u}\oplus 1)} + b^{(\mathbf{u}\oplus 1)} \right), \cdots \sigma_{s^{(\mathbf{u}\oplus m_2)}} \left( \mathbf{X}\mathbf{w}^{(\mathbf{u}\oplus m_2)} + b^{(\mathbf{u}\oplus m_2)} \right)]$ where $\mathbf{u}$ is of length $L - 2$. By Lemma E.15, the innermost layer vector is in $A_{2tree} = \mathbf{H}^{(1)}$ and can be expressed in the form $s^{(\mathbf{u}\oplus i)} = \mathbf{w}^{(\mathbf{u}\oplus i)} = 1$ for all $i$ and $b^{(\mathbf{u}\oplus i)} \in \{-x_1, \cdots, -x_N\}$. $\square$

### E.3 1-D data

In this section, we assume the data is 1-D. We will refer to a switching set defined in Section 3. We call the set of indices a vector switches at the *switching set*.

**Lemma E.13.** *Let $m_1, m_2 \in \mathbb{N}, k \in [m_1 m_2]$. A vector $\mathbf{h} \in \{-1, 1\}^N$ that starts with 1 and switches $k$ times can be written as the sum of at most $m_2$ vectors in $\{-1, 1\}^N$ that each switch at most $m_1$ times, and $\mathbf{1}$.*

*Proof.* Let $I$ be the switching set of $\mathbf{h}$. Let $Q = \left\lceil \frac{k}{m_1} \right\rceil \leq m_2$. For $q \in [Q]$, let $\mathbf{h}^{(q)}$ be a vector in $\{-1, 1\}^N$ that starts with $(-1)^{q+1}$ and whose switching set is $I_q = \{i \in I : i = q \mod m_2\}$. Each $\mathbf{h}^{(q)}$ switches at most $|I_q| \leq m_1$ times. Let $\mathbf{s} = \sum_{q=1}^Q \mathbf{h}^{(q)}$. Since $I_1, \cdots, I_Q$ partition the switching set $I$ of $h$, the sequences $s$ and $h$ switch at exactly the same indices. Note $s_1 = \mathbf{1}\{Q \text{ odd}\} \in \{h_1, h_1 - 1\}$. For $i > 1$,

$$s_i = \begin{cases} s_{i-1} + 2 & \text{if } h_{i-1}^{(q)} = -1, h_i^{(q)} = 1 \text{ for some } q \\ s_{i-1} - 2 & \text{if } h_{i-1}^{(q)} = 1, h_i^{(q)} = -1 \text{ for some } q \\ s_{i-1} & \text{else.} \end{cases}$$

So $\mathbf{s}$ is a vector in $\{0, -2\}^N$ or $\{-1, 1\}^N$ with switch index $I$. Thus $\mathbf{s}$ is $\mathbf{h}$ or $\mathbf{h} - \mathbf{1}$. $\qquad\square$

**Lemma E.14.** *Let $p, m \in \mathbf{N}$. Let $\mathbf{z} \in \{-1, 1\}^N$ with at most $pm$ switches. There is an integer $n \leq m$, $\mathbf{w} \in \{-1, 1\}^n$, and a $N \times n$ matrix $\mathbf{H}$ with columns in $\mathbf{H}^{(p)}$ such that $\mathbf{z} = \sigma(\mathbf{Hw})$.*

*Proof.* For $x \in \{-1, 1\}, \sigma(x) = \sigma(x - 1)$. Apply Lemma E.13 with $m_2 = p, m_1 = m$. Let $w_q = \text{sign}(h_1^{(q)})$. Let $\mathbf{w}$ contain $w_q$ as elements and $\mathbf{H}$ contain $w_q \mathbf{h}^{(q)}$ as columns. $\qquad\square$

The next result states that the hyperplane arrangement set $\mathcal{H}$ (17) consists of all vectors in $\{-1, 1\}^N$ that switch at most once, ie the columns of $\mathbf{H}^{(1)}$ as defined in Section 3.

**Lemma E.15.** $\mathcal{H}(\mathbf{X}) = \mathbf{H}^{(1)}$.

*Proof.* First, $\mathbf{1} = \sigma(\mathbf{0}) = \sigma(\mathbf{X} \cdot \mathbf{0}) \in \mathcal{H}(\mathbf{X})$. Next, let $\mathbf{h} \in \mathcal{H}(\mathbf{X}) - \{\mathbf{1}\} \in \{-1, 1\}^N$. By definition of $\mathcal{H}(\mathbf{X})$, there exists $w, b \in \mathbb{R}$ such that $\mathbf{h} = \mathbf{X}w + b\mathbf{1}$. By Remark E.6, $h_1 = 1$. Let $i$ be the first index at which $\mathbf{h}$ switches. So $x_i w + b < 0 \leq x_{i-1} w + b$, which implies $x_i w < x_{i-1} w$. As assumed in Section 2.1, for all $j > i$, $x_j < x_i$ so $h_j = \sigma(x_j w + b) \leq \sigma(x_i w + b) = \sigma(h_i) = -1$, so $h_j = -1$. So $\mathbf{h}$ switches at most once.

Now, let $\mathbf{h} \in \{-1, 1\}^N$ with $h_1 = 1$. Suppose $\mathbf{h}$ switches once, at index $i \in \{2, \cdots, N\}$. In particular, $h_i = -1$. Let $w = 1, b = -x_{i-1}$. Then at $j < i$, $x_j w + b = x_j - x_{i-1} \geq 0$ so $h_j = 1 = \sigma(x_j w + b)$. And for $j \geq i$, $x_j w + b = x_j - x_{i-1} < 0$ so $h_j = -1 = \sigma(x_j w + b)$. So $\mathbf{h} \in \mathcal{H}(\mathbf{X})$. $\qquad\square$

In a similar notation as $\hat{\mathbf{X}}^{(l)} = \mathbf{X}^{(1,l)}$ from Appendix E.1, denote $\hat{\mathbf{W}}^{(l)} = \mathbf{W}^{(1,l)}, \hat{\mathbf{b}}^{(l)} = \mathbf{b}^{(1,l)}$. Then $\hat{\mathbf{X}}^{(l)} = \sigma(\hat{\mathbf{X}}^{(l-1)} \hat{\mathbf{W}}^{(l-1)} + \mathbf{1} \cdot \hat{\mathbf{b}}^{(l-1)})$ with $\hat{\mathbf{X}}^{(1)} = \mathbf{X}$ (7).

**Proposition E.16.** *The set $A_{L=3,par}$ contains all columns of $\mathbf{H}^{(m_1)}$.*

*Proof.* Note $A_{L=3,par} = \bigcup_{\hat{\mathbf{X}}^{(2)}} \mathcal{H}\left(\hat{\mathbf{X}}^{(2)}\right)$, where the union is taken over $\hat{\mathbf{X}}^{(2)}$ formed from all possible $\hat{\mathbf{W}}^{(1)}, \hat{\mathbf{b}}^{(1)}$. From Lemma E.15, any possible column in $\hat{\mathbf{X}}^{(2)}$ switches at most once. Apply Lemma E.14 with $p = 1, m = m_1$ (so that $mp = m_1$ switches), to any column $\mathbf{z}$ of $\hat{\mathbf{X}}^{(2)}$. $\qquad\square$

The remaining results of this section apply for networks of arbitrary depth $L \geq 2$. We refer to the rectangular network defined in Section 2.2.

**Proposition E.17.** *For a rectangular network of any depth $L \geq 2$, $A_{L,par}(\mathbf{X}) \subset \mathbf{H}^{(m_1)}$.*

*Proof.* Let $\hat{\mathbf{W}}^{(1)} \in \mathbb{R}^{1 \times m_1}$. As assumed in Section 2.1, the data are ordered. So, for any $w, b \in \mathbb{R}$, $\sigma(\mathbf{X}w + b\mathbf{1}) \in \{-1, 1\}^N$ has at most 1 switch. Then $\hat{\mathbf{X}}^{(2)} = \sigma\left(\mathbf{X}\hat{\mathbf{W}}^{(1)} + \mathbf{1} \cdot \hat{\mathbf{b}}^{(1)}\right)$ has $m_1$ columns each with at most one switch. Therefore $\hat{\mathbf{X}}^{(2)}$ has at most $m_1 + 1$ unique rows. Let $R$ be the set of the smallest index of each unique row. We claim that for all layers $l \in \{2, \cdots, L\}$, the rows of $\hat{\mathbf{X}}^{(l)}$ are constant at indices in $[N] - R$, that is, for all $i \in [N] - R$, the $i^{\text{th}}$ and $(i-1)^{\text{th}}$ rows of $\hat{\mathbf{X}}^{(l)}$ are the same. We prove our claim by induction. The base case for $l = 2$ already holds.

Suppose our claim holds for $l \in \{2, \cdots, L-1\}$. Let $\hat{\mathbf{W}}^{(l)} \in \mathbb{R}^{(m_{l-1} \times m_l)}$. The rows of $\hat{\mathbf{X}}^{(l)}$ are constant at indices in $[N] - R$, so for any $\mathbf{w} \in \mathbb{R}^{m_l}, b \in \mathbb{R}$, the elements of the vector $\hat{\mathbf{X}}^{(l)}\mathbf{w} + b\mathbf{1}$ are constant at indices in $[N] - R$. This held for any $\mathbf{w} \in \mathbb{R}^{m_l}$, so the rows of $\hat{\mathbf{X}}^{(l+1)} = \hat{\mathbf{X}}^{(l)}\hat{\mathbf{W}}^{(l)} + \mathbf{1} \cdot \hat{\mathbf{b}}^{(l)}$ are again constant at indices in $[N] - R$. By induction, our claim holds for all $l \in [L]$. So $\hat{\mathbf{X}}^{(L)}$ has at most $|R| \leq m_1 + 1$ unique rows and hence has columns that each switch at most $m_1$ times. $\qquad\square$

The following provides a converse.

**Proposition E.18.** *For a rectangular network of any depth $L \geq 2$, $A_{L,par}(\mathbf{X}) \supset \mathbf{H}^{(m_1)}$.*

*Proof.* Let $\mathbf{z} \in \{-1,1\}^N$ with at most $\min\{N-1, m_1\}$ switches. By Proposition E.16, there exists a feasible $\hat{\mathbf{X}}^{(2)} = \sigma\left(\mathbf{X}\hat{\mathbf{W}}^{(1)} + \mathbf{1} \cdot \hat{\mathbf{b}}^{(1)}\right) \in \mathbb{R}^{N \times m}$ and $\hat{\mathbf{W}}^{(2)} \in \mathbb{R}^{m \times m}$ such that $\mathbf{z}$ is a column of $\hat{\mathbf{X}}^{(3)} = \sigma\left(\hat{\mathbf{X}}^{(2)}\hat{\mathbf{W}}^{(2)} + \mathbf{1} \cdot \hat{\mathbf{b}}^{(1)}\right)$. Now for every $l \in \{3, \ldots, L-1\}$ we can set $\hat{\mathbf{W}}^{(l)} = \mathbf{I}_m$ to be the $m \times m$ identity matrix and $\hat{\mathbf{b}}^{(l)} = \mathbf{0}$ so that $\hat{\mathbf{X}}^{(l+1)} = \sigma\left(\hat{\mathbf{X}}^{(l)}\hat{\mathbf{W}}^{(l)}\right) = \hat{\mathbf{X}}^{(l)}$. Then $\hat{\mathbf{X}}^{(L)} = \hat{\mathbf{X}}^{(3)}$ contains $\mathbf{z}$ as a column. Therefore $\mathbf{z} \in A_{L,par}(\mathbf{X})$. $\qquad\square$

**Lemma E.19.** *Let $L \geq 2$ and $K = \prod_{l=1}^{L-1} m_l$. Then $A_{L\,tree}(\mathbf{X}) = \mathbf{H}^{(K)}$.*

*Proof.* For $l \in [L]$, let $A_l = A_{l\,tree}(\mathbf{X})$. Let $p_k = \prod_{l=1}^{k-1} m_l$. We claim for all $l \in \{2, \cdots, L\}$, $A_l$ consists of all columns in $\mathbf{H}^{(p_l)}$. We prove our claim by induction on $l$. The base case when $l = 2$ holds by Lemma E.15. Now suppose Lemma E.19 holds when $l = k \in \{2, \cdots, L-1\}$. Observe $A_k \subset A_{k+1}$, so if $p_k \geq N-1$, then Lemma E.19 holds for $l = k+1$. So suppose $p_k < N-1$. Then $A_k$ contains all vectors in $\{-1,1\}^N$ with at most $p_k$ switches.

Let $\mathbf{h} \in A_{k+1}$. The set $A_{k+1}$ contains all columns in $J^{(m_k)}([A_k])$. So, there exist $\mathbf{w} \in \mathbb{R}^{m_k}, b \in \mathbb{R}$ and a submatrix $\mathbf{Z} = [A_k]_S$ where $|S| = m_k$ and $\mathbf{h} = \sigma(\mathbf{Z}\mathbf{w} + b)$. Each of the at most $m_k$ columns of $\mathbf{Z}$ has at most $p_k$ switches, so the $N$ rows of $\mathbf{Z}$ change at most $m_k p_k = p_{k+1}$ times. So $\mathbf{Z}\mathbf{w} + b$ changes value, and hence $\mathbf{h} = \sigma(\mathbf{Z}\mathbf{w} + b)$ switches, at most $p_{k+1}$ times. Conversely, by Lemma E.14 with $p = p_k, m = m_k$, the set $A_{k+1}$ of all columns in $J^{(m_k)}([A_k])$ contains all vectors with at most $p_k m_k = p_{k+1}$ switches. So our claim holds for $l = k+1$. By induction, it holds for $l = L$ layers. $\qquad\square$

## E.4 Proofs of results in Section 3

*Proof of Corollary 3.1.* The result follows from the definition of $\mathbf{A}$, the assumption that the data is ordered and that the output of sign activation is $\sigma$ is $\pm 1$ for sign activation. $\qquad\square$

*Proof of Lemma 3.2.* Apply Proposition E.9. Then apply Proposition E.16 and Proposition E.17. $\qquad\square$

*Proof of Theorem 3.4.* Apply Proposition E.9. Then apply Proposition E.17 and Proposition E.18. $\qquad\square$

*Proof of Theorem 3.5.* By Lemma E.12, the training problem is equivalent to Lasso lower bound with dictionary $A_{L,tree}$ given by Lemma E.19. $\qquad\square$

*Proof of Corollary 3.8, Corollary 3.9, Corollary 3.8 .* Follows from Lemma 3.2, Lemma E.8 and Lemma E.12. $\qquad\square$

The following definition adds parameter unscaling for tree networks to Definition E.20.

**Definition E.20.** Parameter unscaling *is the change of variables $\alpha_i' = \text{sign}(\alpha_i)\sqrt{|\alpha_i|}$ and $s^{(i,L-1)'} = \sqrt{|\alpha_i|}$ for parallel nets. For tree nets, it is $\alpha^{(i)'} = \text{sign}(\alpha^{(i)})\sqrt{|\alpha^{(i)}|}$ and $s^{(i)'} = \sqrt{|\alpha^{(i)}|}$.*

**Remark E.21.** *By Remark C.2, the inner weights can be unregularized. So, reconstructed parameters (as defined in Lemma 3.12) that are unscaled according to Definition E.20 achieve the same objective in the training problem as the optimal value of the rescaled problem.*

*Proof of Lemma 3.12.* A reconstruction for 2-layer networks is given in (31). Since 2-layer Lasso features switch at most only once, $g(i,n)$ is the index of that switch, and so it can be seen that the reconstruction in (31) is equivalent to that given in Lemma 3.12. So we now focus on 3-layer networks. By Lemma 3.2 and Lemma D.3, it suffices to show the parameters without unscaling achieve the same objective in the rescaled problem (10) as Lasso. Note $|\mathcal{I}| \leq m_2$ and by Lemma 3.2, $m^{(i)} \leq m_1$ so the weight matrices are feasible. Let $S_n^{(i)}$ be the number of times $\mathbf{A}_i$ switches until index $n$. Since $x_1 > \cdots > x_N$, we get $\hat{\mathbf{X}}_{n,j}^{(i,2)} = \sigma\left(\mathbf{X}\mathbf{W}^{(i,1)} + \mathbf{1} \cdot \mathbf{b}^{(i,1)}\right)_{n,j} = \sigma\left(x_n - x_{\min\{I_j^{(i)}, m^{(i)}\}-1}\right) = -\sigma\left(S_n^{(i)} - j\right)$. So

$$\left(\hat{\mathbf{X}}^{(i,2)}\mathbf{W}^{(i,2)}\right)_n = \sum_{j=1}^{S_n^{(i)}}(-1)(-1)^{j+1} + \sum_{j=S_n^{(i)}+1}^{m_1}(1)(-1)^{j+1} = -\sum_{j=1}^{S_n^{(i)}}(-1)^{j+1} + (-1)^{S_n^{(i)}}\sum_{j=1}^{m_1-S_n^{(i)}}(-1)^{j+1}$$

$$= -\mathbf{1}\{S_n^{(i)}\text{odd}\} + (-1)^{S_n^{(i)}}\mathbf{1}\{m_1 - S_n^{(i)} \text{ odd}\} = \begin{cases} -\mathbf{1}\{S_n^{(i)}\text{odd}\} + (-1)^{S_n^{(i)}}\mathbf{1}\{S_n^{(i)} \text{ odd}\} \text{ if } m_1 \text{ even} \\ -\mathbf{1}\{S_n^{(i)}\text{odd}\} + (-1)^{S_n^{(i)}}\mathbf{1}\{S_n^{(i)} \text{ even}\} \text{ if } m_1 \text{ odd} \end{cases}$$

$$= \begin{cases} -\mathbf{1}\{S_n^{(i)}\text{odd}\} - \mathbf{1}\{S_n^{(i)} \text{ odd}\} \text{ if } m_1 \text{ even} \\ -\mathbf{1}\{S_n^{(i)}\text{odd}\} + \mathbf{1}\{S_n^{(i)} \text{ even}\} \text{ if } m_1 \text{ odd} \end{cases} = \mathbf{1}\{m_1 \text{ odd}\} - 2 \cdot \mathbf{1}\{S_n^{(i)}\text{odd}\}.$$

So $\hat{\mathbf{X}}_n^{(i,3)} = \sigma\left(\hat{\mathbf{X}}^{(i,2)}\mathbf{W}^{(i,2)} + \mathbf{b}^{(i,2)}\mathbf{1}\right)_n = \sigma\left(-2 \cdot \mathbf{1}\left\{S_n^{(i)} \text{ odd}\right\}\right) = (-1)^{S_n^{(i)}} = \mathbf{A}_{n,i}$. So, $f_3(\mathbf{X};\theta) = \sum_{i\in\mathcal{I}}\alpha_i\hat{\mathbf{X}}^{(i,3)} = \mathbf{A}\mathbf{z}$. And, $||\boldsymbol{\alpha}||_1 = ||\mathbf{z}_I^*||_1 = ||\mathbf{z}^*||_1$. So the rescaled problem and Lasso achieve the same objective. □

*Proof of Corollary 3.15 .* By Remark 3.18, $p_{L=3,\beta}^* \leq p_{L=2,\beta}^*$. Let $\theta^{(L)}$ and $\boldsymbol{\alpha}^{(L)}$ denote $\theta$ and $\boldsymbol{\alpha}$ for a $L$-layer net. Since the training and rescaled problems have the same optimal value, to show $p_{L=2,\beta}^* \leq p_{L=3,m_1\beta}^*$, it suffices to show for any optimal $\theta^{(3)}$, there is $\theta^{(2)}$ with $f_2\left(\mathbf{X};\theta^{(2)}\right) = f_3\left(\mathbf{X};\theta^{(3)}\right)$ and $\left\|\boldsymbol{\alpha}^{(2)}\right\|_1 \leq m_1\left\|\boldsymbol{\alpha}^{(3)}\right\|_1$. Let $\mathbf{z}^*$ be optimal in the 3-layer Lasso problem (1). Let $m_3^* = ||\mathbf{z}^*||_0$ and let $\mathbf{z} \in \mathbb{R}^{m_3^*}$ be the subvector of nonzero elements of $\mathbf{z}^*$. Let $m = m_1 m_3^*$. By Lemma 3.12 and its proof, there are $\mathbf{W}^{(i,1)} \in \mathbb{R}^{1\times m_1}, \mathbf{b}^{(i,1)} \in \mathbb{R}^{1\times m_1}, \mathbf{W}^{(i,2)} \in \{1,-1,0\}^{m_1}, \mathbf{b}^{(i,2)} \in \mathbb{R}$ such that $\hat{\mathbf{X}}^{(i,2)}\mathbf{W}^{(i,2)} + \mathbf{b}^{(i,2)}\mathbf{1} \in \{-2,0\}^N$ and $f_3\left(\mathbf{X};\theta\right) =$

$$\sum_{n=1}^{m_3^*} z_i\sigma\left(\hat{\mathbf{X}}^{(i,2)}\mathbf{W}^{(i,2)} + \mathbf{1} \cdot \mathbf{b}^{(i,2)}\right) = \sum_{i=1}^{m_2^*} z_i^*\left(\hat{\mathbf{X}}^{(i,2)}\mathbf{W}^{(i,2)} + \mathbf{1} \cdot \mathbf{b}^{(i,2)} + \mathbf{1}\right) =$$

$$\sum_{i=1}^{m_3^*} z_i^*\left(\sigma\left(\mathbf{X}\mathbf{W}^{(i,1)} + \mathbf{1} \cdot \mathbf{b}^{(i,1)}\right)\mathbf{W}^{(i,2)} + \mathbf{1} \cdot \mathbf{b}^{(i,2)} + \mathbf{1}\right) =$$

$$\sigma\left(\mathbf{X}\underbrace{\left[\mathbf{W}^{(1,1)}\cdots\mathbf{W}^{(m_3^*,1)}\right]}_{\left(\mathbf{W}^{(1,1)},\cdots,\mathbf{W}^{(m,1)}\right)\in\mathbb{R}^{1\times m}} + \mathbf{1}\cdot\underbrace{\left[\mathbf{b}^{(1,1)},\cdots,\mathbf{b}^{(m_3^*,1)}\right]}_{\left(\mathbf{b}^{(1,1)},\cdots,\mathbf{b}^{(m,1)}\right)\in\mathbb{R}^{1\times m}}\right)\underbrace{\begin{bmatrix} z_1\mathbf{W}^{(1,2)} \\ \vdots \\ z_{m_3^*}\mathbf{W}^{(m_3^*,2)} \end{bmatrix}}_{\boldsymbol{\alpha}^{(2)}} + \mathbf{1}\underbrace{\sum_{i=1}^{m_3^*} z_i\left(1 + \mathbf{b}^{(i,2)}\right)}_{\xi},$$

which is $f_2\left(\mathbf{X};\theta^{(2)}\right)$ with $m$ neurons. And $\|\boldsymbol{\alpha}^{(2)}\|_1 \leq m_1\|\mathbf{z}^*\|_1 = m_1\|\boldsymbol{\alpha}^{(3)}\|_1$. □

*Proof of Corollary 3.14 .* By Lemma E.23 , the dictionary matrix for the 2-layer net is full rank for sign activations. The dictionary matrices for deeper nets with sign activation are also full rank by Remark 3.18. Let $\mathbf{u} = \mathbf{A}^T(\mathbf{A}\mathbf{z}^* - \mathbf{y})$. By Remark E.22, as $\beta \to 0$, we have $\mathbf{u} \to \mathbf{0}$, so $\mathbf{A}\mathbf{z} - \mathbf{y} = (\mathbf{A}\mathbf{A}^T)^{-1}\mathbf{A}\mathbf{u} \to \mathbf{0}$. So as $\beta \to 0$, the optimal Lasso objective approaches 0, and by Lemma 3.2, so does the training problem. So $f_L\left(\mathbf{X};\theta\right) \xrightarrow{\beta\to 0} \mathbf{y}$. □

### E.5 Proofs of results for 2-D data

*Proof of Theorem 3.16.* Lemma E.7 holds for any dimension $d$, so the Lasso formulation for Lemma 3.2 similarly holds for $d > 1$ but with a different dictionary $A_{L,par}$ and matrix $\mathbf{A}$.

Let $x_n' = \angle\mathbf{x}^{(n)}$ and order the data so that $x_1' > x_2' > \ldots x_N'$. Let $\mathbf{X}' = (x_1'\cdots x_N') \in \mathbb{R}^N$. Let $\mathbf{w} \in \mathbb{R}^2$ with $\angle\mathbf{w} \in \left[\frac{\pi}{2},\frac{3\pi}{2}\right]$. Let $w' = \angle\mathbf{w}$. Note $\mathbf{w}\mathbf{x}^{(n)} \geq 0$ if and only if $x_n' \in \left[w' - \frac{\pi}{2}, w' + \frac{\pi}{2}\right]$. Since $x_n' < \pi$ for all $n \in [N]$, this condition is equivalent to $x_n' \geq w' - \frac{\pi}{2}$, ie $n \leq \max\left\{n \in [N] : x_n' \geq w' - \frac{\pi}{2}\right\}$. So $\left\{\sigma(\mathbf{X}'\mathbf{w}) : w' \in \left[\frac{\pi}{2},\frac{3\pi}{2}\right]\right\} = \mathbf{H}^{(1)} \cup \{-\mathbf{1}\}$. Similarly $\left\{\sigma(\mathbf{X}'\mathbf{w}) : w' \in \left[-\frac{\pi}{2},\frac{\pi}{2}\right]\right\}$ is the "negation" of this set. Therefore, the $L = 2$ training problem is equivalent to the Lasso problem with dictionary $\mathbf{H}^{(1)}$. Proposition E.18 holds analogously for this training data, so for $L > 2$, the dictionary is $\mathbf{H}^{(m_1)}$. □

*Proof of Lemma 3.17.* Observe $n \leq i$ if and only if $\mathbf{x}^{(n)}\mathbf{W}^{(i,1)} \geq 0$ and so $\sigma\left(\mathbf{X}\mathbf{W}^{(i,1)}\right)\left[\mathbf{1}_i^T, -\mathbf{1}_{N-i}^T\right] = \mathbf{A}_i$. Thus $f_{L=2}(\mathbf{X};\theta) = \sum_i \alpha_i\sigma\left(\mathbf{X}\mathbf{W}^{(i,1)}\right) = \mathbf{A}\mathbf{z}^*$ so $\theta$ achieves the same objective in the rescaled training problem

(10), as the optimal value of Lasso (1). Performing unscaling (Definition E.20) optimal parameters of the rescaled problem makes them optimal in the training problem. $\qquad\square$

## Solution sets of Lasso under minimal regularization

The next remark is from (31).

**Remark E.22** ((31)). *For a neural net with $L = 2$ layers and sign activation, the Lasso problem has an objective function $f(\mathbf{z}) = \frac{1}{2}\|\mathbf{Az} - \mathbf{y}\|_2^2 + \beta\|\mathbf{z}\|_1$ where $\mathbf{A} \in \mathbb{R}^{N \times N}$. By Lemma E.23, $\mathbf{A}$ is full rank, which makes $f$ strongly convex. Therefore the Lasso problem has a unique solution $\mathbf{z}^*$ (54). Moreover, for any Lasso problem, $\mathbf{z}^*$ satisfies the subgradient condition $\mathbf{0} \in \delta f(\mathbf{z}) = \mathbf{A}^T(\mathbf{Az}^* - \mathbf{y}) + \beta\partial\|\mathbf{z}^*\|_1$. Equivalently,*

$$\frac{1}{\beta}\mathbf{A}_n^T(\mathbf{Az}^* - \mathbf{y}) \in \begin{cases} \{-\text{sign}(\mathbf{z}_n^*)\} & \text{if } \mathbf{z}_n^* \neq 0 \\ [-1, 1] & \text{if } \mathbf{z}_n^* = 0 \end{cases}, \quad n \in [N].$$

The next result, Lemma E.23, is from (31).

**Lemma E.23** ((31)). *The 2-layer dictionary matrix for $\sigma(x)=\text{sign}(x)$ has inverse*

$$\mathbf{A}^{-1} = \frac{1}{2}\begin{pmatrix} & \Delta & & \begin{matrix} 0 \\ \vdots \\ 0 \\ -1 \end{matrix} \\ \hline 1 & 0 & \cdots & 0 & 1 \end{pmatrix} \quad \text{where } \Delta = \begin{pmatrix} 1 & -1 & 0 & \cdots & 0 & 0 \\ 0 & 1 & -1 & \cdots & 0 & 0 \\ 0 & 0 & 1 & \cdots & 0 & 0 \\ \vdots & \vdots & \vdots & \ddots & \vdots & \vdots \\ 0 & 0 & 0 & \cdots & 1 & -1 \\ 0 & 0 & 0 & \cdots & 0 & 1 \end{pmatrix} \in \mathbb{R}^{N-1 \times N-1}.$$

*Proof.* Can be verified that multiplying the two matrices gives the identity. $\qquad\square$

## Solution path for sign activation and 1-D data with binary, periodic labels

In this section we assume the neural net uses sign activation, and $d = 1$. Recall $\mathbf{e}^{(n)}$ as the $n^{\text{th}}$ canonical basis vector. Note that in Figures 1 , 24, nd 25, $\mathbf{y} = \mathbf{h}^{(T)}$, $N = 40, T = 10$, and vectors $\mathbf{v} = (v_1, \cdots, v_N)$ are depicted by plotting $(n, v_n)$ as a dot.

**Remark E.24.** *A neural net with all weights being $0$ achieves the same objective in the training problem as the optimal Lasso value and is therefore optimal.*

In this section, we will find the critical value $\beta_c$ defined in Section 4. Then for $\beta < \beta_c$, we use the subgradient condition from Remark E.22 to solve the Lasso problem (1). Note when $L = 2$, $(\mathbf{A}_n)^T = (\mathbf{1}_{1:n}, -\mathbf{1}_{n+1:N})$ switches at $n + 1$.

### E.6 $\quad L = 2$

Assume the network has 2 layers.

**Lemma E.25.** *The elements of $\mathbf{A}^T\mathbf{A} \in \mathbb{R}^{N \times N}$ are $(\mathbf{A}^T\mathbf{A})_{i,j} = N - 2|i - j|$.*

*Proof.* If $1 \leq i \leq j \leq N$ then $(\mathbf{A}^T\mathbf{A})_{i,j} = \sum_{k=1}^{i-1}\mathbf{A}_{i,k}\mathbf{A}_{j,k} + \sum_{k=i}^{j-1}\mathbf{A}_{i,k}\mathbf{A}_{j,k} + \sum_{k=j}^{N}\mathbf{A}_{i,k}\mathbf{A}_{j,k}$
$= \sum_{k=1}^{i-1}(1)(1) + \sum_{k=i}^{j-1}(-1)(1) + \sum_{k=j}^{N}(-1)(-1) = (i - 1) - (j-1-i+1) + (N-j+1) = N+2(i-j)$
$= N-2|i-j|$. Since $\mathbf{A}^T\mathbf{A}$ is symmetric, if $1 \leq j \leq i \leq N$ then $(\mathbf{A}^T\mathbf{A})_{i,j} = (\mathbf{A}^T\mathbf{A})_{j,i} = N+2(j-i) = N-2|i-j|$.
So for any $i, j \in [N]$, $(\mathbf{A}^T\mathbf{A})_{i,j} = N - 2|i - j|$. $\qquad\square$

**Remark E.26.** *By Lemma E.25, $\mathbf{A}^T\mathbf{A}$ is of the form*

$$\mathbf{A}^T\mathbf{A} = \begin{pmatrix} N & N-2 & N-4 & \cdots & 2 & 0 & -2 & \cdots & 6-N & 4-N & 2-N \\ N-2 & N & N-2 & \cdots & 4 & 2 & 0 & \cdots & 8-N & 6-N & 4-N \\ N-4 & N-2 & N & \cdots & 6 & 4 & 2 & \cdots & 10-N & 8-N & 6-N \\ \vdots & \vdots & \vdots & \ddots & \vdots & \vdots & \vdots & \ddots & \vdots & \vdots & \vdots \\ 6-N & 8-N & 10-N & \cdots & 2 & 4 & 6 & \cdots & 10-N & 8-N & 6-N \\ 4-N & 6-N & 8-N & \cdots & 0 & 2 & 4 & \cdots & 8-N & 6-N & 4-N \\ 2-N & 4-N & 6-N & \cdots & -2 & 0 & 2 & \cdots & N-4 & N-2 & N \end{pmatrix}. \tag{22}$$

An example of a column of $\mathbf{A}^T\mathbf{A}$ is plotted in the left plot of Figure 24.

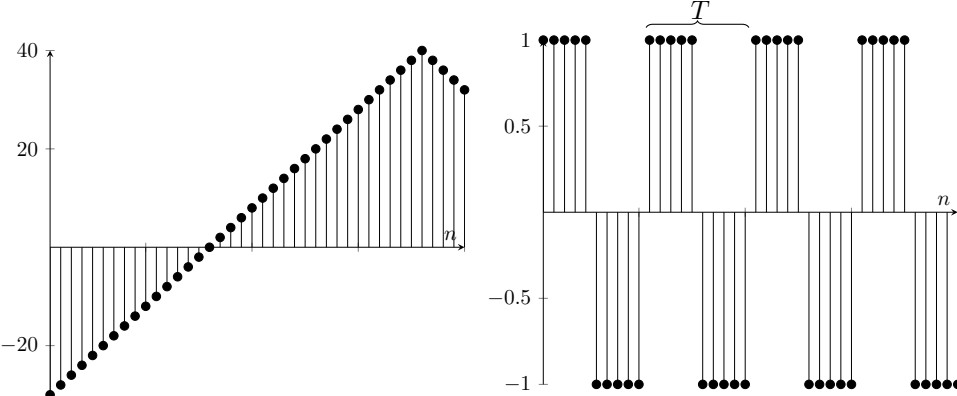

Figure 24: Left: Column 10 of $\mathbf{A}^T\mathbf{A}$ for $N = 40$. Right: Vector $\mathbf{h}^{(T)}$ for $T = 10$.

**Remark E.27.** *For $n \in [N]$, $(\mathbf{A}^T\mathbf{y})_n = \sum_{i=1}^{n} \mathbf{y}_i - \sum_{i=n+1}^{N} \mathbf{y}_i$.*

**Definition E.28.** *For $a, b \in \mathbb{Z}$, let $\mathrm{Quot}(a,b) \in \mathbb{Z}$ and $\mathrm{Rem}(a,b) \in \{0, \cdots, b-1\}$ be the quotient and remainder, respectively, when $a$ is divided by $b$. The* modified remainder *is*

$$\mathrm{rem}(a,b) = \begin{cases} \mathrm{Rem}(a,b) & \text{if } \mathrm{Rem}(a,b) > 0 \\ b & \text{if } \mathrm{Rem}(a,b) = 0 \end{cases} \in [b].$$

*The* modified quotient *is*

$$\mathrm{quot}(a,b) = \frac{a - \mathrm{rem}(a,b)}{b} = \begin{cases} \mathrm{Quot}(a,b) & \text{if } \mathrm{Rem}(a,b) > 0 \\ \mathrm{Quot}(a,b) - 1 & \text{if } \mathrm{Rem}(a,b) = 0. \end{cases}$$

The quotient and remainder are modified to handle vector indices starting at 1 instead of being zero-indexed.

**Remark E.29.** *The square wave has elements $\mathbf{h}_n^{(T)} = \begin{cases} -1 & \text{if } \mathrm{rem}(T,n) \leq T/2 \\ 1 & \text{else.} \end{cases}$*

**Remark E.30.** *Since $\mathbf{h}^{(T)}$ is periodic and zero mean, for $i, n \geq 0$, $\sum_{j=iT+1}^{nT} \mathbf{h}^{(T)}_j = 0$.*

**Lemma E.31.** *The vector $\mathbf{A}^T\mathbf{h}^{(T)}$ is periodic with period $T$. For $n \in [T]$,*

$$\left(\mathbf{A}^T\mathbf{h}^{(T)}\right)_n = 2 \begin{cases} n & \text{if } n \leq \frac{T}{2} \\ T-n & \text{else} \end{cases} \in [0, T].$$

*Proof.* By Remark E.27, Remark E.30, and periodicity of $\mathbf{h}^{(T)}$, for $n \in [T], j \in [2k-1]$,

$$\left(\mathbf{A}^T\mathbf{h}^{(T)}\right)_{n+jT} = \sum_{i=jT+1}^{jT+n} \mathbf{h}^{(T)}{}_i - \sum_{i=n+jT+1}^{(j+1)T} \mathbf{h}^{(T)}{}_i = \begin{cases} \sum_{i=1}^{n} 1 - \sum_{i=n+1}^{T/2} 1 + \sum_{i=1+T/2}^{T} 1 & \text{if } n \le \frac{T}{2} \\ \sum_{i=1}^{T/2} 1 - \sum_{i=\frac{T}{2}+1}^{n} 1 + \sum_{i=n+1}^{T} 1 & \text{else.} \end{cases}$$

Simplifying gives the result. $\qquad\square$

**Lemma E.32.** *Let* $q_n = \text{quot}\left(n, \frac{T}{2}\right) \in \{0, \cdots, 2k-1\}$. *Then*

$$\left(\mathbf{A}^T\mathbf{h}^{(T)}\right)_n = 2(-1)^{q_n+1}\text{rem}\left(n, \frac{T}{2}\right) - \mathbf{1}\{q_n \text{ odd}\}T. \tag{23}$$

*Proof.* Follows from Lemma E.31. $\qquad\square$

**Corollary E.33.** *Suppose* $\mathbf{z} = \mathbf{e}^{(\frac{T}{2})} + \mathbf{e}^{(N-\frac{T}{2})}$. *Then for* $n \le \frac{T}{2}$ *and* $n \ge N - \frac{T}{2}$, $\frac{1}{2}(\mathbf{A}^T\mathbf{A}\mathbf{z})_n = \left(\mathbf{A}^T\mathbf{h}^{(T)}\right)_n$. *And if* $\frac{T}{2} \le n \le N - \frac{T}{2}$, *then* $(\mathbf{A}^T\mathbf{A}\mathbf{z})_n = 2T$.

*Proof.* By Lemma E.25, for $n \in [N]$, $(\mathbf{A}^T\mathbf{A}\mathbf{z})_n = 2\left(N - \left|n - \frac{T}{2}\right| - \left|n - N + \frac{T}{2}\right|\right)$.
Simplifying and applying Lemma E.31 gives the result. $\qquad\square$

**Lemma E.34.** *If* $\mathbf{y} = \mathbf{h}^{(T)}$, *then the critical* $\beta$ *(defined in Section 4) is* $\beta_c = T$ .

*Proof.* By Remark E.24, $\beta_c = \max_{n \in [N]} |\mathbf{A}_n^T\mathbf{y}| = \max_{n \in [N]} \left|\left(\mathbf{A}^T\mathbf{h}^{(T)}\right)_n\right| = T$. $\qquad\square$

**Lemma E.35.** *Let* $\mathbf{y} = \mathbf{h}^{(T)}$. *If* $\beta_T \ge \frac{1}{2}$ *then the solution to the Lasso problem (1) is* $\mathbf{z}^* = \frac{1}{2}(1 - \beta_T)_+ \left(\mathbf{e}^{(\frac{T}{2})} + \mathbf{e}^{(N-\frac{T}{2})}\right)$.

*Proof.* By Lemma E.34, $\beta_c = T$. By Lemma E.34, if $\beta_T \ge 1$ then $\mathbf{z}^* = \mathbf{0}$ as desired. Now suppose $\frac{1}{2} \le \beta_T \le 1$. Let $\delta = 1 - \beta_T, \mathbf{g} = \mathbf{A}_n^T(\mathbf{A}\mathbf{z}^* - \mathbf{y})$. By Corollary E.33 and Lemma E.31,

$$\mathbf{g} = \begin{cases} (\delta - 1)\left(\mathbf{A}^T\mathbf{h}^{(T)}\right)_n = -2\beta_T n & \in [-\beta, 0] & \text{if } n \le \frac{T}{2} \\ \left(\delta T - (\mathbf{A}^T\mathbf{h}_{\mathbf{k},\mathbf{T}})_n\right) = \left(\beta_c - \beta - (\mathbf{A}^T\mathbf{h}^{(T)})_n\right) & \in [-\beta, \beta] & \text{if } \frac{T}{2} \le n \le N - \frac{T}{2} \\ (\delta - 1)\left(\mathbf{A}^T\mathbf{h}^{(T)}\right)_n = -2\beta_T(N - n) & \in [-\beta, 0] & \text{if } N - \frac{T}{2} \le n \end{cases},$$

where the second set inclusion follows from $\left(\mathbf{A}^T\mathbf{h}^{(T)}\right)_n \in [0, \beta_c]$ by Lemma E.31 so that $-\beta \le \mathbf{g} \le \beta_c - \beta \le \beta$. Therefore, $|\mathbf{A}_n^T(\mathbf{A}\mathbf{z}^* - \mathbf{y})| \le \beta$, and at $n \in \{n : \mathbf{z}_n^* \ne 0\} = \{\frac{T}{2}, N - \frac{T}{2}\}$, we have $\mathbf{A}_n^T(\mathbf{A}\mathbf{z}^* - \mathbf{y}) = -\beta_T\frac{T}{2} = -\beta = -\beta\text{sign}(\mathbf{z}_n^*)$. By Remark E.22, $\mathbf{z}^*$ is optimal. $\qquad\square$

**Lemma E.36.** *Let* $a, b, c, d \in \mathbf{Z}_+, d \in \mathbb{R}, r = 1 - \text{rem}(b - a, 2)$. *Then*

$$\sum_{j=a}^{b}(-1)^j(c - jd) = (-1)^a\left((c - ad)r + (-1)^r\frac{(b - (a+r)+1)d}{2}\right) = (-1)^a\begin{cases} \frac{(b-a+1)d}{2} & \text{if } r=0 \\ c - ad - \frac{(b-a)d}{2} & \text{else.} \end{cases} \tag{24}$$

*Proof.* We have

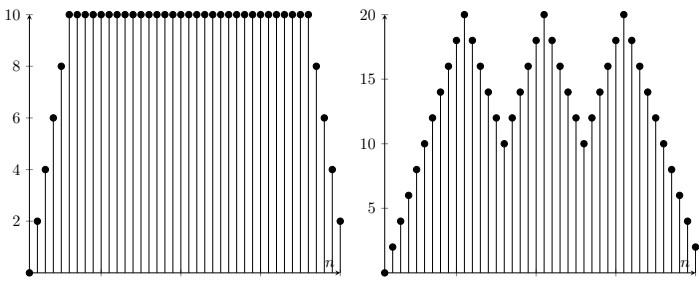

Figure 25: Examples of $\frac{1}{2w_{bdry}}\mathbf{A}^T\mathbf{A}\mathbf{z_{bdry}}$ (left) and $\frac{1}{w_{cycle}}\mathbf{A}^T\mathbf{A}\mathbf{z_{cycle}}$ (right).

$$\sum_{j=a}^{b}(-1)^j\left(c-jd\right) = (-1)^a(c-ad)r + \sum_{j=a+r}^{b}(-1)^j\left(c-jd\right)$$

$$= (-1)^a(c-ad)r + (-1)^{a+r}\sum_{\substack{a+r\le j\le b-1 \\ j-(a+r)\text{ is even}}}(c-jd)-(c-(j+1)d)$$

Simplifying gives equation 24. $\qquad\square$

**Lemma E.37.** *Let $L=2$, $\mathbf{y}=\mathbf{h}^{(T)}$, $0<\beta<\frac{\beta_c}{2}$. Let $w_{\mathrm{bdry}}=1-\frac{3}{2}\beta_T$, $w_{\mathrm{cycle}}=2\beta_T-1$. Let $\mathbf{z}_{\mathrm{bdry}}= w_{\mathrm{bdry}}\left(\mathbf{e}^{\left(\frac{T}{2}\right)}+\mathbf{e}^{\left(N-\frac{T}{2}\right)}\right), \mathbf{z}_{\mathrm{cycle}}=w_{\mathrm{cycle}}\sum_{i=2}^{2k-2}(-1)^i\mathbf{e}^{\left(\frac{T}{2}i\right)}$. Then $\mathbf{z}^*=\mathbf{z}_{bdry}+\mathbf{z}_{cycle}$ solves the Lasso problem 1.*

*Proof.* We show $\mathbf{z}^*$ is optimal using the subgradient condition in Remark E.22. By Corollary E.33,

$$\frac{1}{2w_{\mathrm{bdry}}}(\mathbf{A}^T\mathbf{A}\mathbf{z}_{\mathrm{bdry}})_n = \begin{cases}\left(\mathbf{A}^T\mathbf{h}^{(T)}\right)_n & \text{if } n\le\frac{T}{2}\text{ or }n\ge N-\frac{T}{2} \\ T & \text{if }\frac{T}{2}\le n\le N-\frac{T}{2}\end{cases}, \quad n\in[N]. \qquad (25)$$

Next,

$$\frac{1}{w_{\mathrm{cycle}}}(\mathbf{A}^T\mathbf{A}\mathbf{z}_{\mathrm{cycle}})_n = \sum_{j=2}^{2k-2}(-1)^j\left(N-2\left|n-j\frac{T}{2}\right|\right). \qquad (26)$$

See Figure 25. If $n\le\frac{T}{2}$ or $n\ge N-\frac{T}{2}$ then there is $s\in\{-1,1\}$ such that for all $2\le j\le 2k-2$, $n-j\frac{T}{2}=s(n-j\frac{T}{2})$. Applying Lemma E.36 to equation 26 and simplifying gives $(\mathbf{A}^T\mathbf{A}\mathbf{z_{cycle}})_n= w_{\mathrm{cycle}}\left(N-skT+2sn\right)$. Comparing with Lemma E.31 gives

$$\frac{1}{w_{cycle}}(\mathbf{A}^T\mathbf{A}\mathbf{z}_{\mathrm{cycle}})_n = 2\begin{cases}n & \text{if }n\le\frac{T}{2} \\ N-n & \text{if }n\ge N-\frac{T}{2}\end{cases} = (\mathbf{A}^T\mathbf{h}^{(T)})_n. \qquad (27)$$

Next suppose $\frac{T}{2}\le n\le N-\frac{T}{2}$. Let $q_n=\mathrm{quot}\left(n,\frac{T}{2}\right), r_n=\mathrm{rem}\left(n,\frac{T}{2}\right)$. Then

$$\frac{1}{w_{cycle}}(\mathbf{A}^T\mathbf{A}\mathbf{z_{cycle}})_n = \sum_{j=2}^{q_n}(-1)^j\left(N-2\left(n-j\frac{T}{2}\right)\right) + \sum_{j=q_n+1}^{2k-2}(-1)^j\left(N+2\left(n-j\frac{T}{2}\right)\right). \qquad (28)$$

Applying Lemma E.36 to equation 28 and simplifying gives

$$\frac{1}{w_{cycle}}\left(\mathbf{A}^T\mathbf{A}\mathbf{z}_{\mathrm{cycle}}\right)_n = \begin{cases}N-2n+2T+\frac{q_n-2}{2}T-\frac{2k-2-q_n}{2}T=2\left(T-\mathrm{rem}\left(n,\frac{T}{2}\right)\right) & \text{if }q_n\text{ is even} \\ \frac{1-q_n}{2}T+N+2n-(1+q_n)T-\frac{2k-3-q_n}{2}T=T+2\mathrm{rem}\left(n,\frac{T}{2}\right) & \text{if }q_n\text{ is odd}\end{cases}$$

$$=(2-\mathbf{1}\{q_n\text{ odd}\})T+2(-1)^{q_n+1}\mathrm{rem}\left(n,\frac{T}{2}\right)=2T-\left(\mathbf{A}^T\mathbf{h_{k,T}}\right)_n,$$

where the last equality follows from Lemma E.32. Combining with equation 27 gives

$$(\mathbf{A}^T\mathbf{A}\mathbf{z_{cycle}})_n = w_{cycle} \cdot \begin{cases} \left(\mathbf{A}^T\mathbf{h}^{(T)}\right)_n & \text{if } n \leq \frac{T}{2} \text{ or } n \geq N - \frac{T}{2} \\ 2T - \left(\mathbf{A}^T\mathbf{h}^{(T)}\right)_n & \text{if } \frac{T}{2} \leq n \leq N - \frac{N}{2}. \end{cases} \tag{29}$$

Add equation 25 and equation 29 and plug in $\mathbf{y} = \mathbf{h}^{(T)}$ to get

$$(\mathbf{A}^T\mathbf{A}\mathbf{z})_n = \begin{cases} (w_{cycle}+2w_{bdry})(\mathbf{A}^T\mathbf{y})_n = (1-\beta_T)\,(\mathbf{A}^T\mathbf{y})_n & \text{if } n \leq \frac{T}{2} \text{ or } n \geq N - \frac{T}{2} \\ (w_{bdry}+w_{cycle})2T - w_{cycle}(\mathbf{A}^T\mathbf{y})_n = \beta + (1-2\beta_T)\,(\mathbf{A}^T\mathbf{y})_n & \text{if } \frac{T}{2} \leq n \leq N - \frac{N}{2}. \end{cases} \tag{30}$$

Therefore,

$$-\frac{1}{\beta}\mathbf{A}^T(\mathbf{A}\mathbf{z} - \mathbf{y})_n = \begin{cases} \frac{1}{\beta_c}(\mathbf{A}^T\mathbf{y})_n & \text{if } n \leq \frac{T}{2} \text{ or } n \geq N - \frac{T}{2} \\ 1 + \frac{2}{\beta_c}(\mathbf{A}^T\mathbf{y})_n & \text{if } \frac{T}{2} \leq n \leq N - \frac{N}{2}. \end{cases} \tag{31}$$

By Lemma E.34, $\beta_c = T$. By Lemma E.31, $0 \leq \frac{1}{\beta_c}(\mathbf{A}^T\mathbf{h}^{(T)})_n = \frac{1}{\beta_c}(\mathbf{A}^T\mathbf{y})_n \leq 1$, and $\frac{1}{\beta_c}(\mathbf{A}^T\mathbf{y})_n = 1$ when $n$ is an odd multiple of $\frac{T}{2}$. Since $0 < \beta < \frac{\beta_c}{2}$, we have $w_{bdry} > 0$ and $w_{cycle} < 0$. Therefore $\frac{1}{\beta}\mathbf{A}^T(\mathbf{A}\mathbf{z} - \mathbf{y})_n = -\text{sign}(\mathbf{z}_n^*)$ when $\mathbf{z}_n^* \neq 0$, ie $n$ is an integer multiple of $\frac{T}{2}$. And for all $n \in [N]$, $\left|\frac{1}{\beta}\mathbf{A}^T(\mathbf{A}\mathbf{z} - \mathbf{y})_n\right| \leq 1$. By Remark E.22, $\mathbf{z}^*$ is optimal. $\qquad\square$

The nonzero indices of $\mathbf{z_{bdry}}$ and $\mathbf{z_{cycle}}$ partition those that are multiples of $\frac{T}{2}$.

*Proof of Theorem 4.1.* By Lemma E.35 and Lemma E.37,
$$\mathbf{z}^* = \begin{cases} \frac{1}{2}(1-\beta_T)_+ \left(\mathbf{e}^{\left(\frac{T}{2}\right)} + \mathbf{e}^{\left(N-\frac{T}{2}\right)}\right) & \text{if } \beta_T \geq \frac{1}{2} \\ \mathbf{z_{bdry}} + \mathbf{z_{cycle}} & \text{if } 0 < \beta_T \leq \frac{1}{2}. \end{cases}$$
It is shown in (31) (restated in Remark E.22) that the Lasso problem for the 2-layer sign network with 1-D inputs and general training data $\mathbf{X}, \mathbf{y}$ admits a unique solution $\mathbf{z}^* \in \mathbb{R}^N$. $\qquad\square$

*Proof of Corollary 4.2.* Note that unscaling (Definition E.20) does not change the neural network as a function. The reconstructed neural net (31) before unscaling is $f_2(x;\theta) = \sum_{i=1}^N z_i^* \sigma(x - x_i)$. For $\frac{1}{2} \leq \beta_T \leq 1$, $f_2(x;\theta) = \frac{1}{2}(1-\beta_T)_+ \left(\sigma\left(x - x_{\frac{T}{2}}\right) + \sigma\left(x - x_{N-\frac{T}{2}}\right)\right)$. We can compute $f_2(x;\theta)$ similarly for $\beta_T \leq \frac{1}{2}$. $\qquad\square$

### E.7 $L = 3$ layer nets

*Theorem 4.3.* Since $\mathbf{y} = \mathbf{h}^{(T)}$ switches $2k-1$ times, by Lemma 3.2, $\mathbf{A}_i = \mathbf{h}^{(T)}$ for some $i$. Since $\mathbf{y} = -\mathbf{A}_i$, and for all $n \in [N], \|\mathbf{A}_n\|_2 = N$, we have $i \in \text{argmax}_{n \in N}|\mathbf{A}_n^T\mathbf{y}|$. By Remark E.24, $\beta_c = \max_{n \in N}|\mathbf{A}_n^T\mathbf{y}| = \mathbf{y}^T\mathbf{A}_i = N$. So if $\beta > \beta_c$ then $\mathbf{z}^* = 0$, consistent with $z_i = -(1-\beta_T)_+$.

Next, $\mathbf{z}^*$ satisfies the subgradient condition in Remark E.22, since for $n \in [N]$, $\left|\mathbf{A}_n^T(\mathbf{A}\mathbf{z}^* - \mathbf{y})\right|$

$= \left|\mathbf{A}_n^T(z_i\mathbf{A}_i - \mathbf{A}_i)\right| = (z_i - 1)\left|\mathbf{A}_n^T\mathbf{A}_i\right| = \left|\frac{\beta}{\beta_c}\mathbf{A}_n^T\mathbf{y}\right| \leq \frac{\beta}{\beta_c}\text{argmax}_{n \in N}\left|\mathbf{A}_n^T\mathbf{y}\right| = \leq \beta$. Since $\mathbf{z}_i^* < 0$, when $i = n$, $\mathbf{A}_i^T(\mathbf{A}\mathbf{z}^* - \mathbf{y}) = \beta = -\beta\text{sign}(\mathbf{z}_i^*)$. By Remark E.22, $\mathbf{z}^*$ is optimal.

By (34), the fit $\mathbf{A}\mathbf{z}^*$ is unique, so the residual $\mathbf{r} = \mathbf{y} - \mathbf{A}\mathbf{z}^* = (1 - z_i^*)\mathbf{y}$ is unique. Since $\|\mathbf{A}_n\|_2 = N$ for all $n$, and $\mathbf{A}_i$ is the only feature that is colinear to $\mathbf{y}$, for all $n \neq i$ we have $|\mathbf{A}_n^T\mathbf{r}| = |1 - z_i^*||\mathbf{A}_n^T\mathbf{y}| < |1 - z_i^*||\mathbf{A}_i^T\mathbf{y}| = \beta$, so $z_n^*$ must be zero by the subgradient condition. Therefore the solution is unique. $\qquad\square$

*Corollary 4.4.* Follows from the reconstruction in Lemma 3.12. $\qquad\square$

