# OpenReview forum: "Deep Neural Nets in Low Dimensions with Sign Activations are Convex Lasso Models"
_TMLR — Accepted by TMLR_

### Review · Reviewer_8uab · 2026-04-14

**Summary Of Contributions:**

The paper studies the connection between deep sign networks and lasso problems. Specifically, the paper first starts with the observation that training 2 layer sign networks with 1-d inputs is equivalent to solving a lasso problem where the labels are training labels and features are binary columns with a single sign switch. The paper then generalizes this result to deeper networks, both "rectangular" and "tree-structured". An extension is also shown for 2d data. Experiments also validate that that the Lasso problem is nearly equivalent to training a neural network.

**Audience:**

Yes

**Audience Explanation:**

Yes: connections between neural network optimization (which is notoriously hard to study and interpret) and convex optimization (which has a much deeper set of theoretical tools for analysis) are of great interest to the fields of optimization and deep learning generally. This is especially true since this paper studies deep networks (with more than 2 layers) and arbitrary width, the most empirically relevant setting. The lasso method is also fairly intuitive and well-explored in the literature, which is great for potential future work in this area.

**Broader Impact Concerns:**

No broader impact concerns.

**Claims And Evidence:**

No

**Claims Explanation:**

Overall, the theoretical results appear correct. However, there are some issues that need to be resolved: first, the results largely build off of the results in (53), specifically Lemma 2.1. Unfortunately, this lemma is not sufficiently explained and intuition is not provided. This makes the remaining results difficult to follow. This is especially true because section 3 largely reads like a list of results with a few sentences of intuition connecting them. It would be better if there was an overarching message, perhaps my structuring the section further.

Another potential issue is that the main results could be more rigorously stated. To be specific, several results state an equivalence between a neural network optimization problem and a lasso problem, entailing that the result of the lasso optimization can be used to reconstruct the neural network parameters. However, this reconstruction is not provided in the main text.

Finally, the experiments are conducted in a way that makes it difficult to verify whether they indeed support the claims of the theory. First of all, it is not made clear when an exact equivalence between a lasso problem and the neural network problem holds (should we expect an exact equivalence to hold under all experimental settings?). Is the gap between the neural network loss and the lasso loss purely due to imperfect optimization of the non-convex problem, or is there another factor? Finally, how should we interpret the results from Figure 2: they seem to show that the convex loss lower bounds the non-convex loss, but there is a visible gap that disappears with depth. Does the reduction with depth align with the theory? Is the fact that the non-convex loss is greater than the convex loss sufficient confirmation of the theory? It would be great to have more interpretation of the results here.

**Requested Changes:**

**Critical**
- Add intuition for key theoretical results
- Clearly explain in the main paper how to reconstruct network parameters from the lasso solution (perhaps with an illustrative example)
- Clarify experimental interpretation (see above)

**Would strengthen**
- Restructure section 3 to have a clear message/story
- Figure 2 is too small, has unnecessary colors, and lacks error bars
- Figure 3 has an unused legend in the top right
- Neither Figures 2 nor 3 are centered

---

> ### Author Response · Authors · 2026-06-04
> **Response to Reviewer**
>
> 1) Lemma 2.1 not sufficiently explained:
>
> Added intuition immediately after Lemma 2.1.
>
> 2) suggestion for overarching message/structuring:
>
> We revised the opening of Section 3 to more clearly explain how the subsequent results are related in exploring sign networks as binary Lasso problems. We added further in tuition after Lemma 2.1, Lemma 2.1, Lemma 3.2, Theorem 3.4 (previously Lemma 3.3), Theorem 3.5. (previously Theorem 3.4), and Theorem 4.1 to clarify their interpretation and role in the development of the theory.
>
> 3) reconstruction is not provided in the main text:
>
> While the manuscript already contained an explicit reconstruction result in Lemma 3.10 for 2- and 3-layer parallel networks (which addresses Lemma 2.1 and Lemma 3.2), and an extension in Remark 3.11 to deeper rectangular networks (which addresses Theorem 3.3), the revised manuscript adds a detailed worked example immediately after Lemma 3.10. The example explains the reconstruction of the first-layer bias parameters in terms of runs, defined before the example, which provide another perspective to clarify the intuition in our g(i, n) notation. We also clarified the discussion of reconstruction for tree architectures (addressing Theorem 3.4).
>
> 4) ...unclear when an equivalence between a lasso and the neural network problem holds in
> experiments:
>
> We have expanded the discussion in Section 5.1 to clarify when equivalence is expected. Exact equivalence is predicted by our theory only when comparing the Lasso formulation with neural networks of the same architecture and depth, e.g. the first row of
> Figure 2.
>
> 5) what is the gap between the network and lasso loss due to?
>
> The gap is not necessarily due to a single factor. We have added discussion in Section 5.1 clarifying that several factors may contribute to this gap, including (i) convergence of the non-convex optimization to suboptimal solutions, (ii) the difficulty of optimizing networks with discontinuous sign activations, whose gradients are zero almost everywhere, and (iii) the use of a straight-through estimator (STE) to approximate backpropagation through the sign activation. We also note that finite hyperparameter tuning and optimizer choices may further affect the non-convex results.
>
> 6) Figure 2 visible gap: does the reduction with depth align with the theory?
>
> The reduction of the gap with depth is not a direct prediction of our theory, which characterizes exact equivalence between the Lasso formulation and 2-layer networks of matching architecture. In Figure 2, the convex curve corresponds to the globally optimal solution of the 2-layer Lasso model, while the deeper networks in the second and third rows correspond to non-convex models. The observed decrease in the gap with depth is more a consequence of increased expressivity in deeper architectures. This has been clarified in Section 5.1.
>
> 7) Is the fact that the non-convex loss is greater than the convex loss sufficient confirmation of the theory?
>
> Since the non-convex training is suboptimal, the gap between the convex and non-convex training itself does not definitively prove that the Lasso model finds a globally optimal solution, only that it finds a better solution. However, since β = 0 the optimal training loss is known to be 0, which the Lasso model appears to closely approach. This suggests the Lasso model may globally optimize the training loss, which supports our theory. We have expanded discussion in Section 5.1 on this point.
>
> Requested changes
>
> Critical
>
> - Add intuition for key theoretical results
>
> Added further intuition after Lemma 2.1, Lemma 3.2, Theorem 3.3 (previously Lemma 3.3), Theorem 3.4., and Theorem 4.1.
>
> - reconstruct network parameters from the lasso solution (perhaps with an illustrative example)
>
> Added an example to our existing reconstruction (please see response to Question 3 above.)
>
> - Clarify experimental interpretation (please see above.)
>
> Added clarifications of the experiments (please see responses to Questions 4-7 above.)
>
> Would strengthen
>
> - Restructure section 3 to have a clear message/story
>
> Please see our response to Question 2 above.
>
> - Figure 2 is too small, has unnecessary colors, and lacks error bars
>
> We have increased the size of Figure 2 to improve readability within the page margin
> constraints.
> Use of colors: we keep the individual loss trajectories for all random initializations to
> show the full variability of the non-convex optimization behavior, rather than aggre-
> gating them into a single summary curve.
> Error bars: since we plot all individual training runs (20 initializations) explicitly,
> the figure already displays the full distribution of outcomes and therefore conveys the
> variability directly. We have clarified this in the caption for improved readability.
>
> - Figure 3 has an unused legend in the top right
>
> Good catch, fixed.
>
> - Neither Figures 2 nor 3 are centered
>
> Thanks, they have been centered.

---

### Review · Reviewer_Fq1Q · 2026-04-24

**Summary Of Contributions:**

The paper studies mathematically sign-activation neural networks and shows that, for 1-D input data, training some classes of such networks with weight regularization is equivalent to solving a convex Lasso problem with an explicit binary dictionary matrix. It also mentions that 2-layer networks correspond to a switching-set dictionary, 3-layer rectangular/parallel networks expand this, and deeper rectangular networks do not further enrich the dictionary.

Strengths:
- The paper contains an interesting theoretical contribution. In particular, the explicit convex Lasso reformulation for deep sign networks on 1-D data, together with the characterization of features by switching complexity, is commendable and gives an interpretable view.
- Another strength is that the paper provides explicit reconstruction formulas, e.g., Lemma 3.10 for 3 layers, which makes the convex equivalence much more concrete and potentially more useful than formulations relying only on implicit feature descriptions.

Weaknesses
- The main limitation is the theory's narrow scope to 1-D data, plus a rather special 2-D extension that requires bias-free networks and points with unique angles in (0,\pi). The conclusion acknowledges these limitations, but also states that this analysis can serve as a basis to understand more complex networks and data. This however is not clear from the manuscript — how can this analysis be extended to more complex, practical settings?
- Another concern is that the experimental section is rather limited: the experiments focus on synthetic 1-D data, periodic square wave targets, and 2-D spiral examples. More broad experiments are missing, for example, investigating the depth-saturation.
- The positioning of the paper with respect to prior convexification papers needs to be strengthened. The related work discusses the closest papers, especially the threshold-activation and 2-layer sign-activation literature, but it is important to clearly state what is fundamentally new about sign activations and what is inherited from the broader convex-training line of work.
- Last but not least, the paper is difficult to read, it is densely written and can benefit from a more structured and streamlined presentation

**Audience:**

Yes

**Audience Explanation:**

I believe the paper would be of interest to parts of the TMLR audience working on the theory of neural networks, convex formulations of training, quantized or sign-activation models, etc.

**Broader Impact Concerns:**

I do not see major, immediate, broader-impact concerns.

**Claims And Evidence:**

Yes

**Claims Explanation:**

The core theoretical direction is interesting and supported by propositions, lemmas, and theorems. However, the evidence for some of the broader claims is less convincing. In particular, the claims around better generalization for 3-layer networks are not really established beyond simple data.

**Requested Changes:**

- I would suggest clarifying the technical consistency between the main text and the appendix, in particular the condition in Lemma 3.3 versus the reconstruction requirement in Lemma E.8, and the indexing mismatch in the 2-D extension.
- It would improve the paper if the experimental section were strengthened, for example, by adding at least one direct experiment for the tree architecture. I would also suggest clarifying the empirical setup and framing the comparisons to non-convex training more carefully, especially when comparing deeper standard networks against the 2-layer convex formulation.
- The paper would also benefit from a stronger positioning relative to prior work, by stating more explicitly what is the contribution and novelty.
- Finally, I would suggest improving the structure of the paper and the presentation so that the paper becomes easier to follow and understand the main take-home messages of the analysis.

---

> ### Author Response · Authors · 2026-06-04
> **Response to Reviewer**
>
> Weaknesses
>
> 1) narrow scope to 1-D; how can analysis be extended?
>
> While our explicit characterization of the Lasso dictionary as a switching set is presented for univariate data, we have added discussion in Section B.5 on how the Lasso equivalence extends to multivariate settings, where the dictionary is given by all parallel unit outputs evaluated on the data. Dictionaries can therefore be approximated in practice by sampling parameters and generating the corresponding parallel units as features. This perspective is further developed in prior work on threshold networks [1].
>
> We have added an experiment (Figure 14) in Section B.6 that demonstrates this in a multivariate time-series setting (ECG prediction).
> [1]: T. Ergen, H. I. Gulluk, J. Lacotte, and M. Pilanci, Globally optimal training of neural networks with threshold activation functions, arXiv:2303.03382, (2023)
>
> 2) experiments focus on synthetic 1-D data: More broad experiments are missing, for example, on depth-saturation
>
> We added Section B.6 and Section B.7 providing experiments on real ECG data, and investigating effect of noise, data dimension, width variation, and feature sampling size. In our framework, increasing depth changes the induced feature dictionary; however,
> in practice we approximate this by sampling a fixed number of features. (We have added Section B.5 discussing feature sampling.) As a result, in our experiments with deeper networks depth changes are only partially reflected within the feature subset, since the same number of sampled features is used across settings to ensure tractability and comparability.
>
> Rather than attempting to directly measure saturation in depth (which could require substantially increasing the sampled dictionary size with depth), we explored the other properties described above.
>
> 3) The positioning of the paper with respect to prior convexification papers needs to be strengthened
>
> We have expanded discussion in the Related Work section to more clearly distinguish novel versus inherited contributions. We also expanded discussion immediately after Lemma 2.1 explaining this prior result in more detail.
>
> 4) paper is difficult to read and dense, can benefit from structuring/streamlining
>
> We revised the opening of Section 3 to more clearly explain how the subsequent results fit together in developing the view of sign networks as binary Lasso models.
>
> We added explanatory discussion and intuition following Lemma 2.1, Lemma 3.2, Theorem 3.4 (previously Lemma 3.3), Theorem 3.5, and Theorem 4.1 to clarify their interpretation and role in the overall development of the theory. We also added a concrete example for the reconstruction given in Lemma 3.10.
>
> We broke several dense paragraphs into shorter units for readability and promoted key, previously inline, statements to dedicated remarks and definitions such as Remark 3.6 and Definition 3.3. We reclassified the previous Lemma 3.3 as a theorem to better reflect its central role in the paper as a main result.
>
> Requested changes:
>
> - clarify the technical consistency between the main text and the appendix
>
> We really appreciate this observation as you caught some important typos.
>
> Lemma 3.3 (now Theorem 3.4): m1 should be mL as consistent with Lemma E.8. This has been fixed.
>
> For the 2-D extension, Theorem 3.16 (previously Theorem 3.14):  $\mathbf{H}^{(m_{L-2})}$ should be $\mathbf{H}^{(1)}$ for $L=2$ and $\mathbf{H}^{(m_{1})}$ for $L>2$. Its proof in Section E.5 also had a typo that $\mathbf{H}^{(m_{L-2})}$ should be $\mathbf{H}^{(m_{1})}$. This has been fixed.
>
> - adding one direct experiment for the tree architecture
>
> We have added Section B.7, which provides Lasso experiments for 4-layer tree networks on 1-D sinusoidal data. This section evaluates the tree network found through the convex formulation on training and testing data, as well as exploring the regularization path, the performance distribution under feature sampling, and effects of structured and unstructured noise. The results are plotted in Figures 15-19.
>
> - clarify the empirical setup and frame the comparisons to non-convex training more carefully
>
> We have revised Section 5.1 to clarify the interpretation of Figure 2 and the scope of our theory. We now explicitly distinguish between the exact equivalence between convex Lasso training and global optimization of 2-layer sign networks, and comparisons involving deeper (3- and 4-layer) non-convex networks, which are not covered by this equivalence and are therefore only qualitative comparisons of expressivity rather than theory-preserving baselines.
>
> - stronger positioning relative to prior work
>
> Please see our response to Question 3 above.
>
> - improving the structure of the paper and the presentation
>
> Please see our response to Question 4 above.

---

### Review · Reviewer_xtdS · 2026-05-06

**Summary Of Contributions:**

This paper establishes mathematical equivalences between training sign-activated neural networks on low-dimensional data and solving convex Lasso problems with explicitly characterized dictionary matrices. Binary networks offer hardware efficiency advantages, but are difficult to train due to the vanishing gradient problem introduced by the non-differentiable sign activation. This work sidesteps that challenge by reformulating the training problem as an equivalent convex Lasso problem, guaranteeing globally optimal solutions. Theoretical results are developed for rectangular and tree network architectures with sign activation across 2 and 3 layers, covering both 1-D and 2-D input data. Empirical simulations confirms the findings.

**Audience:**

Yes

**Audience Explanation:**

This work provides rigorous theoretical analysis of sign-activated neural networks, establishing exact and equivalent reformulations as convex Lasso problems. In doing so, it directly addresses a fundamental training challenge for binary neural networks, broadening its appeal beyond the theoretical machine learning community to hardware efficiency and edge/loT researchers potentially for deployment.

**Broader Impact Concerns:**

N/A.

**Claims And Evidence:**

Yes

**Claims Explanation:**

The paper is well-written and technically rigorous. The theoretical results are supported by numerical simulations. The appendix provides comprehensive supplementary proofs, analysis and additional experiments that further corroborate the main findings.

**Requested Changes:**

1. The theory is developed for both rectangular and tree networks. Could the authors provide additional numerical analysis of solution paths, generalization performance, or reconstruction quality specifically for tree architectures, beyond the rectangular case?
2. How robust is the proposed method to noise in the input data, including both random noise and structured perturbations on the underlying low-dimensional manifold?
3. How does the proposed LASSO-based solver compare to its (non-binary) MLP counterpart in terms of prediction accuracy?
4. Experiments are limited to synthetic 1-D and a 2-D spiral data. There are no experiments on real-world low-dimensional datasets (e.g., audio signals, time-series, MRI data as mentioned in the introduction for motivation). I understand that the primary focus is on theoretical analysis, it would be valuable to see how the approach performs on more complex, real-world data. In particular, how easily can the method be extended beyond 2D to slightly higher-dimensional settings in practical scenarios?

---

> ### Author Response · Authors · 2026-06-04
> **Response to Reviewer**
>
> Requested Changes:
>
> 1) provide additional numerical analysis of solution paths, generalization performance, or reconstruction quality specifically for tree architectures:
>
> We have added Section B.7, which provides Lasso experiments for 4-layer tree networks on 1-D sinusoidal data. This section evaluates reconstruction quality and generalization performance by evaluating the tree network reconstructed from a Lasso solution on testing (as well as training) data. The results are plotted in Figures 15-19. The experiments include analyzing network fits (solution paths) and loss (regularization paths) (Figure 16) of tree networks as the regularization parameter changes.
>
> 2) robustness to random and unstructured noise:
>
> In general, as studied in prior literature, the ℓ1 regularization of Lasso improves robustness to noise by selecting a sparse, parsimonious set of features to represent the data that helps avoid overfitting. A larger regularization parameter β selects fewer features, which can increase robustness to noise. While specific theoretical analysis of robustness to noise for the neural network-induced Lasso is a direction for future work, we added experiments in Section B.7 investigating this numerically for the tree architecture. We perform experiments where the underlying data is a sinusoid. In one experiment, random Gaussian noise is added on top of the sinusoid (Figure 17). In another experiment, structured perturbations are introduced through a small sinusoidal phase shift applied to the underlying sinusoid (Figure 18). In these experiments, we vary the noise strength, quantified by the variance for Gaussian noise and by the amplitude for the sinusoidal phase-shift, and evaluate the network. The network’s error increases as expected as the noise strength increases, but the fit remains reasonable for a moderate amount of noise, as seen in the network fits (right panels) and error (left panels) of Figures 17, 18. In particular, the test error for the phase shift noise appears convex as a function of the noise amplitude, suggesting robustness at low noise levels. A theoretical analysis of this phenomenon is a direction for future work.
>
> 3) How does the proposed LASSO-based solver compare to its (non-binary) MLP counterpart in terms of prediction accuracy?
>
> Thank you for the question. We added Section B.8 providing a comparison between convex Lasso for 2-layer sign networks and standard non-convex training of 2-layer ReLU (non-binary) networks on ECG data. The non-convex loss over epochs is shown in Figure 22. Figure 20 shows a box plot of the training and test loss. While the convex sign network achieves lower training loss, its test loss is higher than that of the standard MLP. Reasons include a different activation function between the two networks, as well as optimality guarantee differences between convex versus non-convex training. While the sign network may have higher test error over ReLU, the 2-layer convex sign training still outperforms the 2-layer non-convex sign training in Section 5.1. While ReLU may enjoy somewhat better test performance over sign activation, their performance in this experiment is comparable, as can be seen visually in the overlapping fits in Figure 21. Moreover, sign activation has benefits such as being compatible with quantization
> and efficient computation. The best choice of activation therefore depends on the computational and performance needs of the setting.
>
> 4) how easily can the method be extended beyond 2D to slightly higher-dimensional settings in practical scenarios?
>
> While our explicit characterization of the Lasso dictionary as a switching set is for univariate data, we have added discussion in Section B.5 on how the Lasso equivalence extends to multivariate inputs, where the dictionary is given by all parallel unit outputs
> evaluated on the data. Dictionaries can therefore be approximated in practice by sampling parameters and generating the corresponding parallel units as features. This perspective is further developed in prior work on threshold networks [1]. We have added an experiment (Figure 14) in Section B.6 that demonstrates this in a multivariate time-series setting (ECG prediction)
>
> [1]: T. Ergen, H. I. Gulluk, J. Lacotte, and M. Pilanci, Globally optimal training of neural networks with threshold activation functions, arXiv:2303.03382, (2023)

---

### Author Response · Authors · 2026-06-04
**Response to all Reviewers**

Dear Reviewers,

Thank you for the careful evaluation of the manuscript as well as the comments and ideas beyond the presented work. Changes in the revised manuscript are indicated by red font of the uploaded revised manuscript.
Please find in the response boxes below the answers to your questions and addressing your requested changes.

Thanks very much for your time,

The Authors

---

### Decision · Action_Editor_gJ8S · 2026-06-30

**Recommendation:** Accept with minor revision

**Additional Comments:**

Before publication, I recommend a thorough editorial revision, including:
* Where feasible, improving the overall exposition and readability throughout the paper.
* Fixing LaTeX issues (e.g., the quotation marks around "vanishing gradient").
* Removing blank pages (p14) and addressing formatting inconsistencies.
* Correcting typographical errors (e.g., "parallsign", "ffects", and similar issues).

**Audience:**

Yes

**Audience Explanation:**

The connections between neural networks and convex optimization are an established and active line of theoretical machine learning research, and the explicit characterization developed in this paper is likely to be of interest to readers working on optimization, learning theory, and neural network analysis.

**Claims And Evidence:**

Yes

**Claims Explanation:**

The paper presents several theoretical results on sign-activated neural networks, establishing explicit connections between their training problems and convex Lasso formulations, and analyzing how these equivalences evolve with network depth.

The work builds on prior literature on convex reformulations of threshold and sign-activated neural networks, but the reviewers broadly agree that the paper contains novel theoretical contributions. While the paper is primarily theoretical, relying on simplified data settings and without demonstrating computational efficiency improvements in practice, there was also general agreement that this is appropriate for a contribution of this type.

Several concerns raised during the review process were adequately addressed in the rebuttal, including a clearer discussion of the relationship to prior work and the addition of further experimental evidence. The main remaining concern relates to presentation. One reviewer suggested that the paper would benefit from substantial rewriting for clarity, but did not provide specific requests and did not engage further during the discussion period. Given that the authors’ responses were reasonable, no reviewer raised substantive concerns about the correctness of the theoretical results, and no additional experiments appear necessary, I support acceptance subject to improvements in clarity and presentation.

---

> ### Author Response · Authors · 2026-07-15
>
> We thank the Action Editor for the careful assessment of our manuscript and for recommending acceptance subject to minor revision. We have carefully revised the camera-ready version to address the editorial suggestions. Specifically, we performed an additional editorial pass throughout the manuscript to improve exposition and readability, including minor wording, grammar, and clarity improvements, as well as shortening long sentences where appropriate. We corrected the LaTeX formatting issue involving the quotation marks around "vanishing gradient." We carefully reviewed the manuscript for formatting consistency. The submitted manuscript does not contain a blank page on page 14. To improve the layout, we increased the spacing between Figures 2 and 3 on page 14 and centered Figures 7, 24, and 25 in the appendix. We also corrected typographical errors, including "parallsign" and "ffects."

---

> > ### Comment · Action_Editor_gJ8S · 2026-07-20
> >
> > Please add a link to the review page for the paper, as requested by the instructions.

---

> > > ### Author Response · Authors · 2026-07-20
> > >
> > > Thanks for the catch. We have added the link to the OpenReview page.